# RNA-mediated condensation of TFE3 oncofusions facilitates transcriptional hub formation to promote translocation renal cell carcinoma

Lei Guo [1,2,4,5] ✉, Rongjie Zhao[1,4], Yi-Tsang Lee [1,4], Junhua Huang[1], James Wengler [1], Logan Rivera [1], Tingting Hong [1], Tianlu Wang[1], Kunjal Rathod[1], Ashley Suris[1], Yitian Wu[1], Xiaoli Cai[1], Rui Wang[1], Yubin Zhou [1,2,5] ✉ & Yun Huang [1,2,3,5] ✉

Transcription factor E3 (TFE3) oncofusions are frequently detected in the Microphthalmia transcription factor (MiT) family translocation renal cell carcinoma (tRCC), a rare pediatric renal cancer with limited treatment options. The mechanisms by which TFE3 oncofusions promote tRCC malignancy remain inadequately defined. Here, we demonstrate that the RNA-binding capability conferred by TFE3 fusion partners drives the formation of TFE3 condensates. This further enables TFE3 oncofusions to co-condensate with RNA polymerase II (RNAPII) and other RNA-binding proteins, such as para-speckle component 1 (PSPC1), ultimately driving the formation of transcriptional hubs to promote pro-oncogenic transcription. Dissolution of oncofusion condensates through nanobody-based chemogenetic manipulation effectively curtails tRCC cell growth both in vitro and in vivo, suggesting the therapeutic potential for targeting oncofusion condensation in tRCC. Collectively, our study establishes the causal role of RNA and RNA-binding proteins in facilitating oncofusion condensation to promote renal cancer progression.

Oncofusion proteins arising from chromosomal translocation are regarded as common drivers for malignant transformation[1–4]. These oncofusions often induce aberrant protein kinase activity and reshape transcriptional outputs to drive oncogenesis[2]. These oncofusion events are notably prevalent in pediatric cancers with poor prognosis[1,5]. Oncofusion events involving the microphthalmia transcription factor (MiT) family are prominent oncogenic drivers in a subset of translocation renal cell carcinoma (tRCC)[6–10]. MiT belongs to a family of helix-loop-helix leucine zipper proteins that play critical roles in the development and differentiation of various cell types[6–10]. The MiT family tRCC is characterized by chromosomal rearrangements that fuse the transcription factor E3 (TFE3) or EB (TFEB) (on chromosomal loci Xp11.2 and 6p21, respectively) with various partners[6–10]. Over 80% of these fusion events involve TFE3, pairing with partners such as Non-POU domain-containing octamer-binding (NONO), RNA-binding motif protein 10 (RBM10), splicing factor proline and glutamine rich (SFPQ), alveolar soft part sarcoma chromosome region 1 (ASPSCR1), proline rich mitotic checkpoint control

[1]Institute of Biosciences and Technology, Texas A&M University, Houston, TX, USA. [2]Department of Translational Medical Sciences, School of Medicine, Texas A&M University, Houston, TX, USA. [3]Department of Biomedical Engineering, College of Engineering, Texas A&M University, College Station, TX, USA. [4]These authors contributed equally: Lei Guo, Rongjie Zhao, Yi-Tsang Lee. [5]These authors jointly supervised this work: Lei Guo, Yubin Zhou, Yun Huang. ✉e-mail: guoleijay@tamu.edu; yubinzhou@tamu.edu; yun.huang@tamu.edu

factor (PRCC), and mediator complex subunit 15 (MED15)[10,11]. Representing ~30% of renal cell carcinoma cases in children and adolescents, tRCC often has a poor prognosis and lacks effective treatment strategies[12], highlighting the need for a detailed mechanistic understanding of its molecular etiology. Interestingly, our analysis using published RNA-seq data[13,14] reveals that tRCC cases exhibit remarkably similar transcriptomes regardless of fusion partners, which are distinctly different from those of non- tRCC cases, suggesting a shared transcriptional regulatory mechanism among these fusion proteins to support the pathogenesis of tRCCs (Supplementary Fig. 1a).

Oncofusion protein condensation has been increasingly recognized as an underappreciated mechanism to support oncogenic signaling and transcriptional remodeling during tumorigenesis[2,5,15,16]. In tRCC, the intrinsically disordered coiled-coil domain (CCD) of NONO within the NONO-TFE3 fusion protein has been reported to facilitate its liquid-like condensate formation[17]. While the disordered regions within various proteins have been reported to support the multivalency condensate formation in many cases[2,5], other biomolecules, such as nucleic acids (DNA and RNA), also play crucial roles in establishing transcriptional condensates[18,19]. Abnormal RNA production can lead to re-localization of transcriptional condensates and contribute to cancer occurrence and progression[20]. Additionally, RNA binding proteins (RBPs) could interact with RNA polymerase II (RNAPII) to enhance the polymerase engagement and transcriptional activity[21]. Among all the analyzed fusion partners of TFE3, 57% (8 out of 14) are found to be RBPs (Supplementary Fig. 1b, c), suggesting a potential shared role of RNA and/or RBPs in regulating oncogenic transcription in the MiT family tRCC.

In this study, using the NONO-TFE3 fusion protein as a model, we have identified that the RNA binding capability of the TFE3 fusion partner plays a critical role in supporting condensate formation and pro-oncogenic transcription, a common feature shared by several other TFE3 oncofusions found in tRCC patients. Employing a dTAG-based inducible degradation system alongside integrative omics studies, as well as CRISPR-based functional genomic screening, we have uncovered that the RNA binding ability conferred by the TFE3 fusion partners, such as NONO, could facilitate its co-condensation with RNAPII, thereby promoting the formation of transcriptional hubs. Moreover, we have found that paraspeckle component 1 (PSPC1), an RNA binding protein[22], enhances the formation of TFE3 oncofusion condensates to boost gene transcription. Our findings highlight the role of RNA-mediated multivalency in fostering the formation of liquid-like condensates by oncofusion proteins, thereby promoting pro-oncogenic transcription and renal cancer cell growth. By leveraging a nanobody fused with maltose-binding protein (MBP), we have exploited a chemogenetic approach to effectively disrupt TFE3 oncofusion-mediated transcriptional condensate formation, resulting in the suppression of tRCC cell growth in vitro and in vivo. Together, our study establishes the rationale for targeting oncogenic condensates as a promising therapeutic strategy for renal cancer intervention.

## Results

### Identifying DNA and RNA binding targets of NONO-TFE3

To identify the direct targets of the TFE3 fusion protein in tRCC, we used a CRISPR-based knock-in approach to introduce a GFP-dTAG-Flag cassette[23] at the C-terminus of the endogenous *NONO-TFE3* oncofusion gene in a patient-derived UOK109 tRCC cell line[24,25] (a gift from Dr. Marston Linehan's laboratory) (Fig. 1a, Supplementary Fig. 1d–f). The knockin of GFP-dTAG-Flag was confirmed by Sanger's sequencing (Supplementary Fig. 1e) and immunoblotting (Supplementary Fig. 1f), with the resultant cell line designated UOK109-KI. Using this inducible degradation system, we observed pronounced depletion of NONO-TFE3 starting at 6 h (h) after dTAG-13 treatment. Importantly, the reversible nature of the dTAG system allowed for restoration of NONO-TFE3 fusion protein expression at 24–48 h after dTAG-13 withdrawal

(Fig. 1b). Functionally, we observed a reduction in colony formation in dTAG-13 treated cells, which was reversible upon dTAG-13 washout for 48 h. Such changes were not observed in the UOK109 parental cells lacking the dTAG system (Fig. 1c). Similar scenarios were seen in UOK145, a patient-derived tRCC cell line that bears SFPQ-TFE3 oncofusion (Supplementary Fig. 1g). Collectively, these findings unequivocally establish the critical role of TFE3 fusion proteins in supporting tRCC cell growth in vitro.

Since NONO-TFE3 contains both DNA and RNA binding motifs[25], we performed cleavage under targets and tagmentation (CUT&Tag)[26]and RNA immunoprecipitation followed by sequencing (RIP-seq)[27]to systematically map its potential DNA and RNA binding sites, respectively, across the mammalian genome. CUT&Tag and RIP-seq analyses confirmed strong DNA and RNA binding of NONO-TFE3 in UOK109-KI cells prior to dTAG-13 treatment (Fig. 1d, and Supplementary Fig. 2a, b, Supplementary Data 1). Upon acute dTAG-13-induced NONO-TFE3 degradation, we observed a pronounced reduction in both DNA and RNA binding (Fig. 1d–f, and Supplementary Fig. 2a, b). The disappearing or weakening of CUT&Tag peaks at specific regions could be restored upon dTAG-13 washout (Fig. 1d–e), indicating the specificity of identified DNA targets of NONO-TFE3. The genomic distribution analysis revealed that NONO-TFE3 binding regions identified by both assays were enriched at promoters (50–59%) and introns (25–27%) (Fig. 1g). Further analysis using the Genomic Regions Enrichment of Annotations Tool (GREAT)[28] revealed that CUT&Tag peaks were most enriched at genes associated with cell death, cytoskeleton organization, GTPase signaling, and metabolic regulation (Supplementary Fig. 2c, and Supplementary Data 1). RIP-seq data also revealed the enrichment of similar gene sets (Supplementary Fig. 2d, and Supplementary Data 1). Motif analysis based on CUT&Tag data showed enrichment of DNA binding motifs for TFE3, MITF, and AP-1 family transcription factors, which are known regulators of cell proliferation and metabolism[29–31] (Supplementary Fig. 2e). To further clarify the types of RNAs that bind to NONO-TFE3, we analyzed the RIP-seq data and found that protein-coding mRNAs are the predominant RNA species interacting with NONO-TFE3, followed by lncRNAs and miRNAs (Supplementary Fig. 2f). We cannot rule out the presence of enhancer RNAs (eRNAs), as the RIP-seq conditions may not be optimal for detecting these short-lived and low-abundance RNA species. Comparative analysis of genes located at the identified DNA or RNA binding sites revealed that ~40% (1557 out of 3666) of NONO-TFE3 RNA-binding sites overlapped with their DNA binding regions (Fig. 1h, i). RNA-seq analysis revealed that the overlapping genes identified from RIP-seq and CUT&Tag (Fig. 1h, $n = 1557$) exhibited higher expression levels and more pronounced changes in expression following dTAG-13 treatment compared to non-overlapping genes (Supplementary Fig. 2g). Moreover, genes exhibiting higher RNA association with NONO-TFE3 also showed enhanced DNA binding (Fig. 1i), suggesting a potential *cis*-regulatory mechanism between RNA and DNA binding of NONO-TFE3 in tRCC cells.

The transcription factor TFE3 can bind genomic DNA independently of fusion partners[32]. Fusion with its partner NONO, a well-established RNA-binding protein, might alter the genomic distribution of TFE3. To explore this, we compared the DNA binding profiles of TFE3 in the presence and absence of NONO fusion. Considering that wild type (WT) TFE3 predominantly resides in the cytosol under non-stressed conditions[33,34], we utilized the reported S321A mutation fused with a nuclear localization sequence (NLS) (NLS-TFE3-S321A) to enforce nuclear localization for genomic binding[33,34]. To exclude the possibility that endogenous NONO-TFE3 might interfere with the binding of overexpressed TFE3-S321A, we transfected dTAG-13-treated UOK109 KI cells with NLS-TFE3-S321A and confirmed the comparable expression levels of NLS-TFE3-S321A and NONO-TFE3 (Supplementary Fig. 2h). We performed similar CUT&Tag experiments and identified over 88% identical genomic binding sites between the two groups (Fig. 1j, and

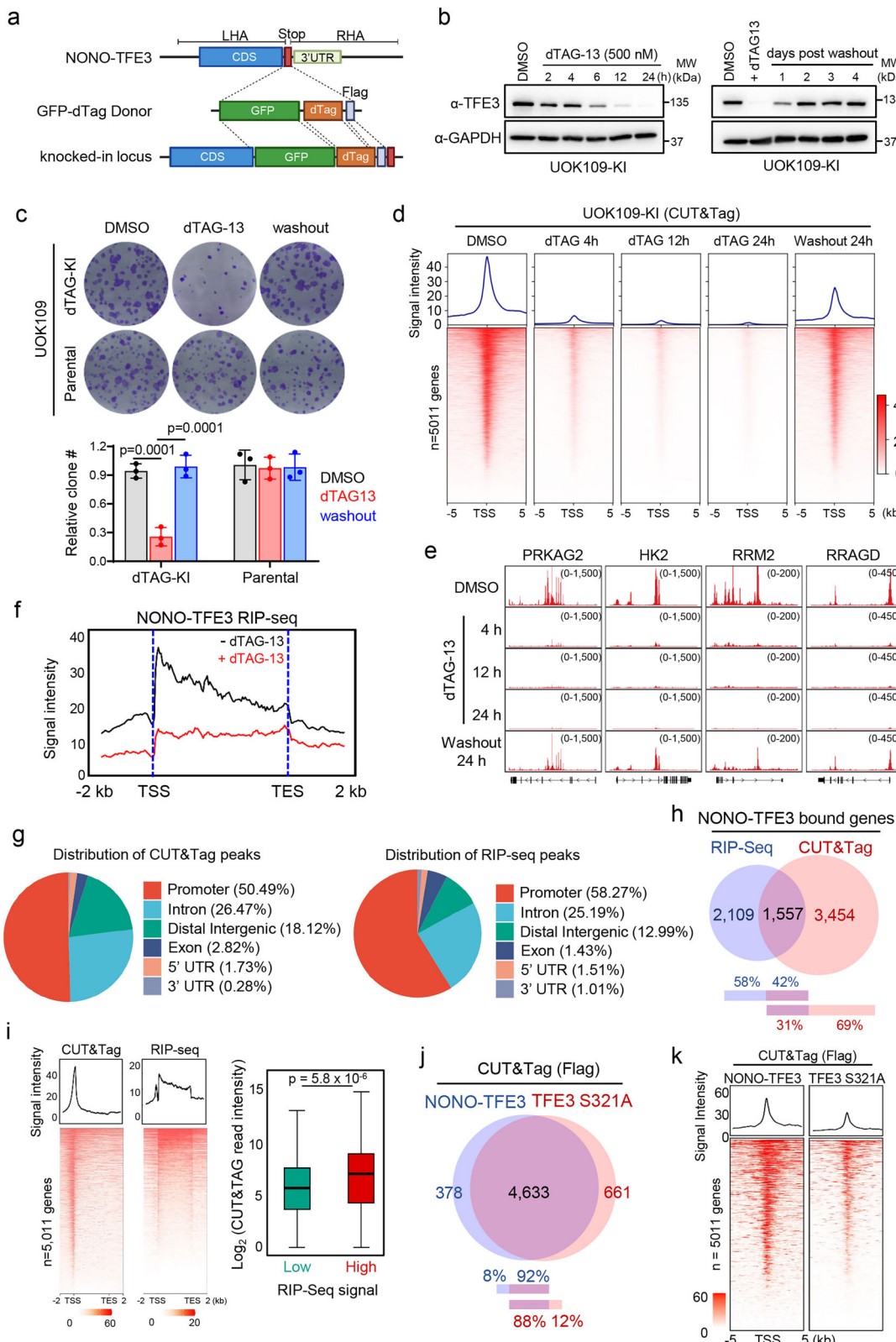

Supplementary Data 1). Interestingly, NONO-TFE3 showed enhanced genomic binding at original TFE3 binding sites (Fig. 1k). This observation was further supported by chromatin association assays under various salt concentrations to strip a protein of interest from chromatin. We observed that NONO-TFE3 showed a notably stronger chromatin association than NLS-TFE3-S321A once the NaCl concentrations reached above 150 mM (Supplementary Fig. 2h). In summary, these findings suggest that TFE3 fusion proteins, such as NONO-TFE3 with dual DNA and RNA binding capabilities, may participate in cis-regulation involving interplays between DNA and RNA binding. While the fusion partner NONO enhances chromatin association, it does not lead to massive alterations in the genomic distribution of TFE3.

**Fig. 1 | Identification of DNA and RNA binding targets of NONO-TFE3.**
**a** Schematic depicting the CRISPR-homology directed degron (dTAG) knocking in (KI) strategy. GFP and Flag tags were also knocked in together with dTAG. LHA or RHA: left or right homology arm. **b** Representative immunoblotting showing time-dependent degradation of the endogenous NONO-TFE3 with dTAG13 (left) (500 nM) and reversible expression (right) after dTAG13 washing out at the indicated time points in UOK109 KI cells. $n = 3$ independent biological replicates. **c** Representative images (top) and quantification (bottom) of clone formation assay using UOK109 parental or KI cells with indicated conditions. $n = 3$ independent biological replicates, one-way ANOVA with Tukey's post-hoc test. Data were shown as mean ± SD. **d, e** CUT&Tag heatmap signals of NONO-TFE3 DNA binding sites around the transcriptional start site (TSS, ± 5 kb) (**d**) and genome browser views of representative loci (**e**) in UOK109 KI cells treated with dTAG13 (500 nM) at indicated time points. $n = 2$ independent biological replicates. **f** Normalized RIP-seq intensities of NONO-TFE3 binding genes from −2 kb TSS to +2 kb. transcriptional

end site (TES) with or without dTAG13 (500 nM) for 24 h. $n = 2$ independent biological replicates. **g** Genomic annotation of the NONO-TFE3 CUT&Tag peaks (left) and RIP-seq peaks (right) in UOK109 KI cells. **h** Venn diagram showing the overlapping bound genes of NONO-TFE3 identified from CUT&Tag and RIP-seq. **i** The binding profiles (left) and correlation analysis (right) between NONO-TFE3 CUT&Tag and RIP-seq signals. Heatmaps were sorted by NONO-TFE3 CUT&Tag and presented the signals of the bound genes from −2 kb TSS to +2 kb TES. The box plots indicated the median (center line), the third and first quartiles (box limits) and 1.5x interquartile range (IQR) above and below the box (whiskers). ($n = 2$ independent biological replicates; two-sided Wilcoxon test). **j, k** Venn diagram (**j**) and heatmap binding profiles of NONO-TFE3 or TFE3-S321A CUT&Tag signals (**k**) in dTAG-13 treated UOK109 KI cells transfected with Flag-tagged NLS-TFE3 S321A. $n = 2$ independent biological replicates. Source data are provided as a Source Data file.

## Identifying the primary transcriptional targets of NONO-TFE3 in tRCC

To identify the primary transcriptional targets of NONO-TFE3, we performed SLAM-seq[35], a robust method designed to identify nascent transcripts by selectively labeling newly transcribed genes, in UOK109-KI cells. SLAM-seq was conducted under five conditions: DMSO treatment, dTAG-13 treatment for 4, 12, and 24 h, and 24 h post dTAG-13 washout (Fig. 2a, and Supplementary Fig. 3a). We observed dynamic changes in nascent transcription upon NONO-TFE3 depletion (Fig. 2b, c, and Supplementary Fig. 3b-c), indicating direct involvement of NONO-TFE3 in transcriptional regulation. Differential gene expression analysis revealed that the majority genes (278 out of 297, ~94%) were down-regulated following dTAG-13-induced fusion protein degradation (Fig. 2c, d, and Supplementary Data 2). Importantly, the transcriptional levels of these genes were restored upon dTAG-13 washout for 24 h, confirming the specificity of these genes as NONO-TFE3 targets (Fig. 2c). Gene Ontology (GO) analysis revealed that NONO-TFE3 targets are primarily involved in cellular and lipid metabolism, MAPK cascade, and homeostatic processes (Fig. 2d, e, and Supplementary Data 2), which are known downstream pathways regulated by WT TFE3 and TFE3 oncofusion proteins[32,36–38].

To further explore the relationship between genomic binding of NONO-TFE3 and its transcriptional targets, we compared the differentially expressed genes (DEGs) identified from SLAM-seq with the genomic regions enriched by CUT&Tag between the DMSO (solvent control) and 4-h dTAG-treatment groups (Fig. 2f). Among the 169 DEGs identified from SLAM-seq, 68% (115 out of 169 genes) were found to be occupied by NONO-TFE3 (Fig. 2f, and Supplementary Data 2). However, only a very small fraction of genes (2%, 115 out of 5011) within the differential NONO-TFE3 bound genomic regions, as revealed by CUT&Tag analysis described above (Fig. 1d), showed overlaps with SLAM-seq identified DEGs (Fig. 2f). To investigate this further, we compared the DNA (CUT&Tag) and RNA (RIP) binding profiles of NONO-TFE3 at these primary ($n = 115$; defined as those showing overlaps in the SLAM-seq and CUT&Tag assays) and non-primary ($n = 4896$) target genes. Interestingly, we observed significant enrichment of both CUT&Tag and RIP-seq peaks at these primary targets of NONO-TFE3, but to a much lesser extent at non-primary targets (Fig. 2g). Such enrichment was markedly reduced after acute depletion of NONO-TFE3 (Fig. 2g), suggesting the direct involvement of NONO-TFE3 in mediating this event. Furthermore, ~64% of NONO-TFE3 targeted genes revealed by SLAM-seq overlapped with peaks from RIP-seq (Fig. 2h), suggesting a potential *cis*-regulatory feedback loop in which RNA binding to its target loci may influence the transcriptional activity of NONO-TFE3. To explore this possibility, we used a customized reporter system by incorporating a TFE3 DNA-binding sequence (GTCACGTGAC, 6x) upstream of the luciferase reporter gene[21]. In this assay, luciferase activity served as a readout for NONO-TFE3 transcriptional activity. To manipulate the RNA levels surrounding the

locus, we cloned DNA sequences encoding RRM2 RNAs (200 nucleotides from 5′ of RRM2 cDNA, which was identified from RIP-seq) adjacent to the luciferase gene under the control of doxycycline (Dox)-inducible promoters (Fig. 2i, top panel). Following co-transfection of this reporter construct with NONO-TFE3 into HEK293T cells, we applied increasing concentrations of Dox to induce varying levels of RNA expression in the vicinity of TFE3 binding sites (Supplementary Fig. 3d). As anticipated, moderate levels of feedback RNA enhanced NONO-TFE3 transcriptional activity, while excessive RNA expression led to suppression (Fig. 2i, bottom panel). This finding is consistent with previously reported effects observed for the Mediator complex component mediator 1 (MED1)[21]. These results collectively support a model in which transcription is governed by a non-equilibrium, RNA-mediated feedback mechanism (Fig. 2j). In this model, low levels of short RNAs produced during transcription initiation enhance target gene expression, whereas high levels of RNAs generated during elongation exert inhibitory effects[21,39,40].

In addition, these primary NONO-TFE3 targeted genes identified in UOK109 cells are highly expressed not only in patients bearing NONO-TFE3 fusions but also across various fusion types in tRCC patients with various MiT oncofusion events (Supplementary Fig. 3e)[41]. These results suggest the clinical relevance of our data collected from UOK109 cells. At 12- and 24-h post-dTAG treatment, we noted a gradual increase in the numbers of DEGs identified from SLAM-seq analysis (Supplementary Fig. 3f, g, Supplementary Data 2). Many of these DEGs are involved in metabolic processes, cell cycle, and stress response (Supplementary Fig. 3h, and Supplementary Data 2). Notably, 56% (144 out of 256) and 27% (189 out of 700) DEGs overlapped with CUT&Tag data collected at the matched time points (Supplementary Fig. 3i). This overlap suggests potential secondary effects involved in NONO-TFE3 mediated transcriptional regulation. Collectively, these results strongly suggest that the transcriptional activity of NONO-TFE3 is linked to its genomic binding and RNA interactions.

## RNA-dependent NONO-TFE3 condensation supports tRCC growth

Given that NONO alone can form liquid-like condensates[42], we set out to examine whether NONO-TFE3 would exhibit similar behavior. Confocal microscopy revealed robust condensate formation of GFP-NONO-TFE3, but not the nucleus-localized TFE3 variant (NLS-TFE3-S321A), when expressed in UOK109 cells (Fig. 3a). Using fluorescence recovery after photobleaching (FRAP), we observed dynamic fusion events mediated by NONO-TFE3 condensates, both in the cellular contexts and with purified recombinant proteins (Fig. 3b, c, Supplementary Fig. 4a, b), confirming its liquid-like behavior. Similar condensation patterns were also observed with at least five additional tRCC-related TFE3 fusion proteins, including U2AF2-TFE3, KHSRP-TFE3, MATR3-TFE3, RBM10-TFE3 and SPFQ-TFE3, where the fusion partners themselves also demonstrated intrinsic droplet-forming

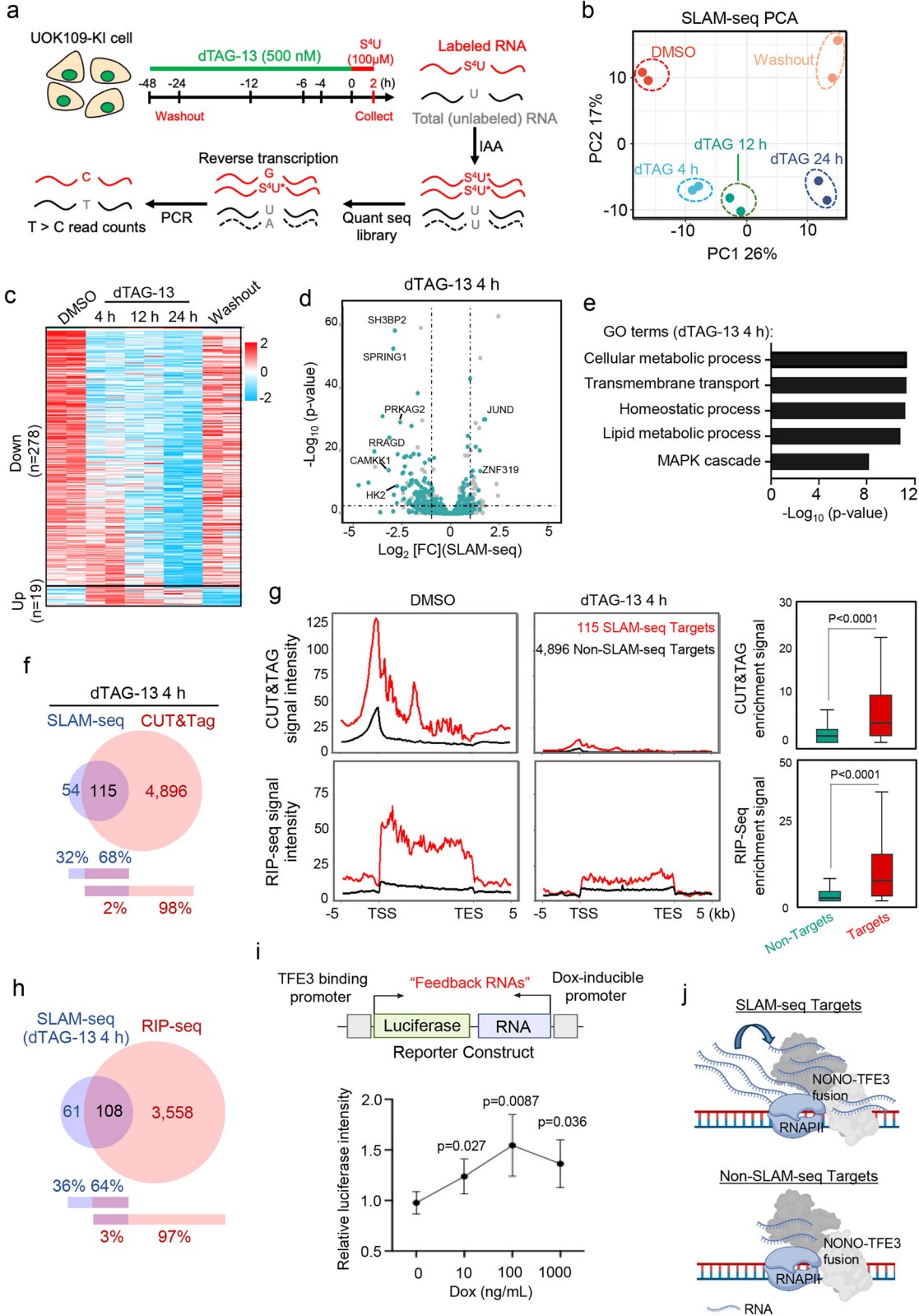

properties (Supplementary Fig. 4c). These findings converge to suggest that the fusion partners are one of the important components contributing to liquid-like condensate formation of TFE3 oncofusions, indicating a common molecular feature among TFE3 oncofusions found in tRCC patients.

In the NONO-TFE3 fusion protein, the NONO part (exons 1–9) contains a prion-like domain (PLD), RNA recognition motifs 1 and 2

(RRM1/2), and a coiled-coil domain (CCD)[43] (Fig. 3d). To identify domain(s) that are essential for supporting biomolecular condensation, we generated several NONO-TFE3 variants by deleting selected domains within NONO and expressed them individually in UOK109 cells at comparable levels (Supplementary Fig. 4d, top panel). The nuclear locations of these variants were confirmed (Fig. 3e). Deletion of CCD or RRM1/2, but not PLD, abolished condensate formation

**Fig. 2 | Identifying the primary transcriptional targets of NONO-TFE3 in tRCC.**
**a** Scheme of the experimental design for SLAM-seq in UOK109 KI cells. See details in
the "Methods" section. **b** Principal component analysis (PCA) of SLAM-seq results at
the indicated timepoints. $n = 2$ independent biological replicates. **c** Heatmap of
differentially expressed genes (DEGs) at the indicated timepoints after dTAG-13
(500 nM) treatment as measured by SLAM-seq in UOK109 KI cells. **d** Volcano plot
showing the genes differentially expressed after 4-h dTAG13 treatment (two-sided,
unpaired Student's t-test). **e** The Gene Ontology (GO) analysis of DEGs identified
from the 4-h dTAG13 treatment group (Fisher's exact test). **f** Venn diagram showing
the overlapping genes identified from SLAM-seq and CUT&Tag analysis (compar-
ison between before and 4-h after dTAG13 treatment). **g** Average metagene plots
(left) and quantification (right) of CUT&Tag or RIP-seq signal intensities of defined
NONO-TFE3 targeted ($n = 115$) or non-targeted genes ($n = 4896$) from −5 kb TSS to
+5 kb TES. The box plots indicated the median (center line), the third and first

quartiles (box limits) and 1.5x interquartile range (IQR) above and below the box
(whiskers). ($n = 2$ independent biological replicates; two-sided Wilcoxon test).
**h** Venn diagram showing the overlapping genes identified from SLAM-seq (4-h
dTAG 13 treatment) and RIP-seq assays. **i** Scheme depicting the reporter system
(top) where local RNA expression near a luciferase reporter gene can be induced by
doxycycline (Dox). Transcriptional activity of NONO-TFE3 was monitored by luci-
ferase intensity (normalized to 0 ng/mL Dox, bottom). $n = 6$ independent biological
replicates, one-way ANOVA with Tukey's post-hoc test. Data are shown as mean ±
SD. **j** A tentative model showing how NONO-TFE3 regulates target gene expression.
High promoter binding and *cis* binding of newly synthesized mRNAs at the same
loci are both required to sustain the expression of NONO-TFE3 target genes. Source
data are provided as a Source Data file. Created in BioRender. Suris, A. (https://
BioRender.com/j0aoaxy).

---

(Fig. 3e). PONDR analysis indicates that CCD is highly disordered
(Fig. 3d), and its role in supporting NONO-TFE3 condensation aligns
with previous findings that highlight the role of disordered regions in
facilitating condensate formation[17]. In contrast, RRM1/2 are structu-
rally ordered according to PONDR analysis (Fig. 3d), suggesting that
they may employ a distinct mechanism to aid NONO-TFE3 condensate
formation.

RRM1/2 are RNA recognition motifs within NONO to facilitate its
RNA binding[43]. Nucleic acids, such as RNA, are known to support
multivalent interactions within condensates[18,19]. To demonstrate that
RNA is an integral component of NONO-TFE3 condensates, we
employed an RNA labeling approach that enables temporal and spatial
visualization of newly synthesized RNA within cells[44]. This strategy
involves the incorporation of 5-ethynyl uridine (EU), an alkyne-
modified nucleoside, into nascent RNA, followed by "click chemistry"
with an azide-conjugated fluorescent dye. We detected a robust
colocalization between nascent RNA and NONO-TFE3 condensates,
confirming the ability of NONO-TFE3 to associate with newly tran-
scribed RNA (Supplementary Fig. 4e). Given this, we hypothesized that
the RNA binding capability of RRM1/2 is crucial for the condensation of
NONO-TFE3. To test this, we took a two-pronged approach: (i) utilizing
a 4A mutant (F113A, F115A, K192A and I194 A) that has been reported
to disrupt RNA binding of RRM1/2[45]; and (ii) fusing RNase A at the
N-terminus of NONO-TFE3 to degrade the surrounding RNA
molecules. Both the 4A mutant and the RNase A fused NONO-TFE3
chimera abolished condensate formation in UOK109 cells (Fig. 3e). A
catalytically inactive RNase A mutant (H12A, H119A) fused to NONO-
TFE3 retained its ability to form condensates (Supplementary
Fig. 4f, g). These results clearly demonstrated the critical role of
RNA binding in facilitating nuclear condensate formation of NONO-
TFE3. Furthermore, deletion of RNA binding domains in other
fusion partners, e.g., U2AF2, KHSRP, MATR3, RBM10 and SFPQ,
impaired their nuclear condensates formation (Supplementary
Fig. 4c), suggesting that RNA binding mediated biomolecular con-
densation formation is a general feature shared by some of TFE3
oncofusion proteins. Next, we performed CUT&Tag and RIP-seq in
cells expressing NONO-TFE3 and 4A mutation to determine if RNA-
binding influences genomic occupancy of NONO-TFE3. As shown in
Supplementary Fig. 4h, the 4A mutant exhibited markedly reduced
RNA-binding capacity along with diminished genomic binding,
suggesting that RNA binding is a critical determinant mediating the
genomic binding of NONO-TFE3.

To further ascertain that RNA binding is crucial for supporting
NONO-TFE3 condensation on chromatin, we introduced NONO-TFE3
variants into an engineered U2OS cell line capable of synthetic LacO
array analysis[46] (Fig. 3f). The LacO array enables visualization of
interactions between a protein of interest and its genomic binding loci
at local high-concentration hubs[46]. Using this assay, we observed that
both WT NONO-TFE3 and its ΔPLD variant formed bright LacO-
associated hubs (Fig. 3g). In contrast, the ΔRRM1/2 variant and the 4A

mutant, both devoid of RNA binding, failed to form such hubs (Fig. 3g).
These results indicate that RNA binding is required for NONO-TFE3 to
create multivalent hubs on the chromatin.

Next, we moved on to evaluate the impact of NONO-TFE3 con-
densation on tRCC cell growth using an in vitro colony formation assay
and xenograft mouse models. We introduced NONO-TFE3 and its
fusion variants, as well as NLS-TFE3-S321A (as nuclear control), into
UOK109-KI cells pretreated with dTAG-13 for 2 days to avoid compi-
lations from endogenous NONO-TFE3. As shown in Supplementary
Fig. 4i, NLS-TFE3-S321A and NONO-TFE3 variants that failed to form
condensates, including ΔCCD, ΔRRM1/2, and 4A, did not support tRCC
cell growth. Consistently, similar results were obtained in mouse
xenograft models inoculated with UOK109 cells, where shRNA tar-
geting endogenous *NONO-TFE3* fusion gene was introduced alongside
the corresponding shRNA-resistant variants (Supplementary Fig. 4d,
bottom panel; Fig. 3h–i). We noticed that UOK109 cells stably
expressing NLS-TFE3-S321A showed reduced protein levels compared
to those expressing NONO-TFE3 (Supplementary Fig. 4d, middle
panel; Supplementary Fig. 4j). A previous study identified an evolu-
tionarily conserved phosphorylation-dependent degron sequence
(E46-D52) within the N-terminus of TFE3, which is essential for its
ubiquitination and subsequent proteasomal degradation by the E3
ligase CUL1[β-TrCP1/2 34]. Mutation or deletion of this degron sequence
increased TFE3 protein stability[34]. In the case of NONO-TFE3, the
N-terminal degron is replaced by the NONO sequence, thereby
bypassing E3 ligase-mediated degradation. Consequently, NONO-TFE3
exhibits enhanced protein stability and higher levels compared to NLS-
TFE3-S321A, which retains the degron sequence at the N-terminus. To
rule out the possibility that the compromised oncogenic potential of
NLS-TFE3-S321A is due to its lower expression level, we generated
UOK109 KI cells expressing the C-terminal portion of TFE3 (TFE3-C, the
region fused to NONO in the chimeric protein). After dTAG-13 treat-
ment, these cells expressed NONO-TFE3 and TFE3-C at comparable
levels (Supplementary Fig. 4j). Furthermore, TFE3-C was exclusively
localized to the nucleus (Supplementary Fig. 4k). Similar as NLS-TFE3-
S321A, TFE3-C failed to rescue the oncogenic signatures observed in
UOK109 KI cells expressing the intact NONO-TFE3 (Supplementary
Fig. 4i). Moreover, RNA-seq was performed on dTAG-13−treated cells
expressing various TFE3 variants to compare their molecular features.
As shown in Supplementary Fig. 4l, m, dTAG-13 treated cells expressing
full-length NONO-TFE3 exhibited transcriptional profiles similar to
those of DMSO-treated cells. In contrast, NONO-TFE3 truncations
deficient in RNA binding (4A) or condensation formation (4A or ΔCC),
along with TFE3 mutants (NLS-TFE3-S321A and TFE3-C), displayed
transcriptional landscapes similar to those of cells lacking full-length
NONO-TFE3 (dTAG-13−treated group) and failed to restore the tran-
scriptomic signatures of UOK109 KI cells prior to dTAG-13 treatment.
Collectively, findings from both in vitro and in vivo studies underscore
the requirement of condensation for supporting the oncogenic func-
tion of NONO-TFE3 in tRCC. The disordered CCD and the RNA-binding

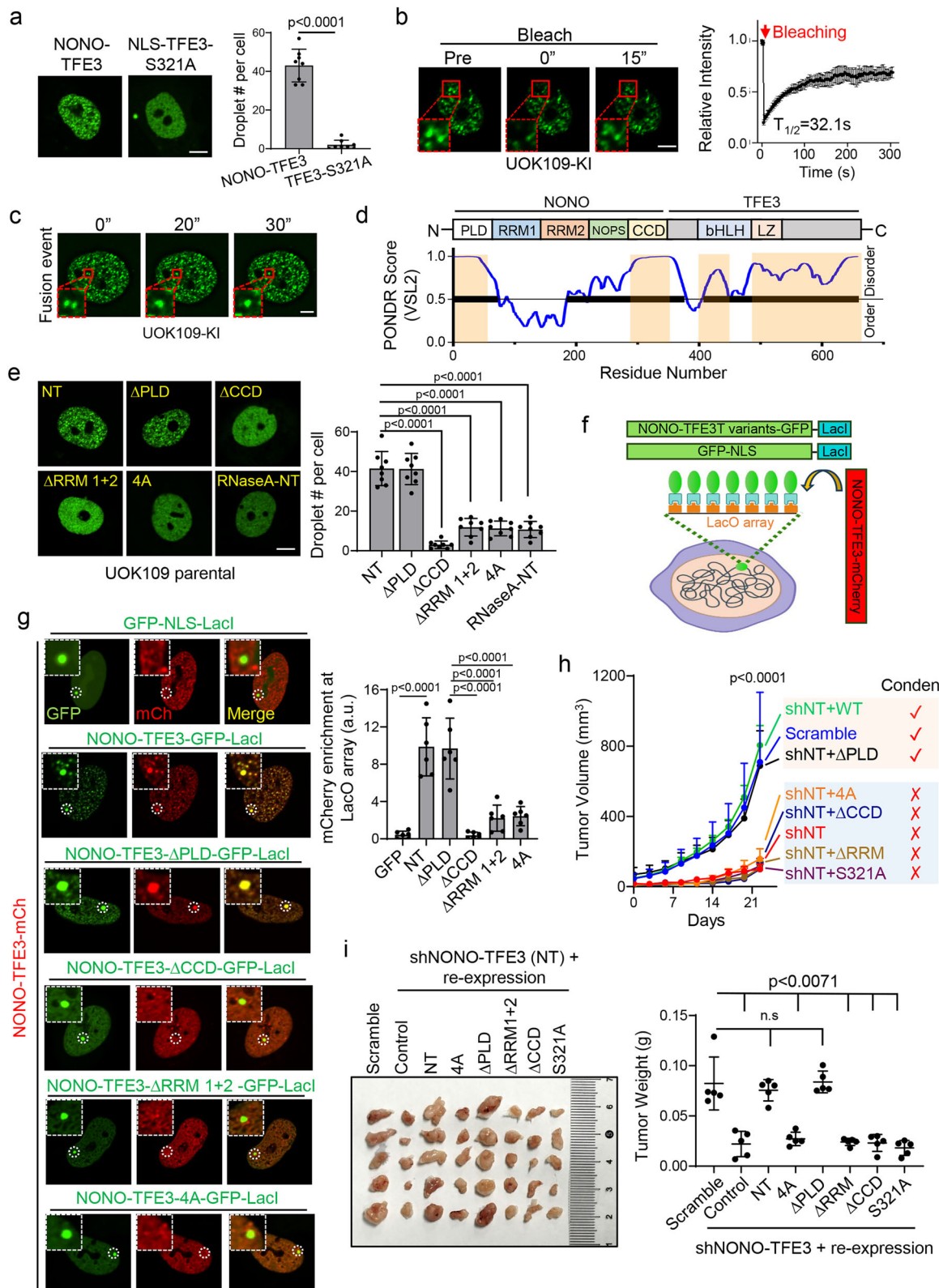

capability of RRM1/2 in NONO both contribute to the condensate formation and oncogenic potential of NONO-TFE3.

## NONO-TFE3 co-condensates with PSPC1 to promote tRCC malignancy

To examine how RNA binding mediates condensation to support the pro-oncogenic transcriptional activity of NONO-TFE3, we explored the

potential molecular composition within NONO-TFE3 condensates by using TurboID-based proximity proteomics[47] (Fig. 4a, and Supplementary Fig. 5a, b). TurboID is an engineered biotin ligase that can capture weak and transient interactions with a protein of interest in the presence of biotin via biotinylation of surrounding proteins within a radius of 10 nm. The NONO-TFE4-4A mutant incapable of RNA binding was used in the assay as a control (Fig. 4b, and Supplementary Fig. 5a,

**Fig. 3 | RNA-dependent condensation of NONO-TFE3 supports tRCC growth.**
**a** Representative images (left) and quantification (right) of droplet formation in
UOK109 cells transfected with NONO-TFE3-GFP or NLS-TFE3 S321A. NLS:
nuclear localization signal. ($n = 8$ cells from 3 independent biological replicates;
two-sided unpaired Student's t-test). Scale bar, 10 μm. **b** Representative
images (left) and normalized fluorescence signals (right) of endogenous NONO-
TFE3-GFP during FRAP assay in UOK109 KI cells ($n = 3$ cells). The red arrow: photo-
bleaching starting time point. Scale bar, 10 μm. **c** Representative live-cell imaging
showing fusion events of endogenous NONO-TFE3-GFP over time (red box). $n = 4$
independent biological replicates. Scale bar, 5 μm. **d** Domain structure (top) and
intrinsically disordered tendency (bottom) of NONO-TFE3 calculated by Predictor
of Natural Disordered Regions (PONDR). A score of over 0.5 indicates predicted
intrinsically disordered regions (IDRs). **e** Representative images (left) and quanti-
fication (right) of droplet formation in UOK109 parental cells transfected with
GFP-fused NONO-TFE3 truncation variants. ($n = 8$ cells from 3 independent bio-
logical replicates; one-way ANOVA with Tukey's post-hoc test). Scale bar, 10 μm. **f** The LacI-LacO array for
multivalent homotypic interaction detection between NONO-TFE3 and its trunca-
tions. NONO-TFE3 truncates-LacI are labeled with GFP, and NONO-TFE3 WT is
labeled with mCherry. If homotypic interaction exists, both GFP- and mCherry-
labeled proteins will form puncta at endogenous LacO arrays. **g** Representative
images (left) and quantification (right) of puncta formation in U2OS cells con-
taining LacI-LacO array transfected with the indicated NONO-TFE3 truncation var-
iants. The LacO array locus is circled and images representing the zoom-in views of
LacO array loci are shown (white box). ($n = 6$ cells from 3 independent biological
replicates; one-way ANOVA with Tukey's post-hoc test). Scale bar, 10 μm. **h** Tumor
growth curves of UOK109 KI xenografts transduced with shRNA against NONO-
TFE3 and rescued with the indicated constructs. ($n = 10$ mice; one-way ANOVA with
Tukey's post-hoc test). **i** Representative images of isolated tumors (left) and
quantification (right) of tumor weights at day 24 of in vivo models mentioned in **h**.
($n = 10$ mice; one-way ANOVA with Tukey's post-hoc test). Data are shown as
mean ± SD. Source data are provided as a Source Data file.

b) to identify potential candidates that co-condensed with NONO-TFE3
in an RNA-dependent manner. We identified a total of 546 proteins in
the proximity of NONO-TFE3 but not NONO-TFE4-4A (Fig. 4b, and
Supplementary Data 3). GO analysis unveiled enrichment of proteins
involved in RNA binding, splicing, epigenetic regulation, and DNA
replication (Fig. 4c). To independently validate these interactions, we
performed co-immunoprecipitation (Co-IP) experiments for selected
candidates such as SFPQ and PSPC1 (Fig. 4d). Interestingly,
many known TFE3 fusion partners, including ZC3H4, SFPQ, RBM10, MATR3,
and KHSRP, were also enriched as NONO-TFE3 binding proteins, but
not with the NONO-TFE4-4A mutant (Fig. 4b, and Supplementary
Data 3). Again, these findings strongly suggest a shared regulatory
mechanism among these TFE3 fusion proteins.

Given the enrichment of RNA binding proteins and epigenetic
regulators in the TurboID experiment, we performed a CRISPR-
mediated functional screen using customized sgRNA libraries that
target RNA binding proteins and epigenetic regulators[48], seeking to
evaluate their impact on tRCC cell survival (Fig. 4e, and Supplementary
Fig. 5c, d). To identify candidates specific for tRCC cell (UOK109)
survival, we included another RCC cell line, 786-O, which lacks TFE3
fusion protein. Our screen identified 147 candidates that were essential
for both UOK109 and 786-O survival, as well as 48 and 43 candidates
specifically required for UOK109 or 786-O cell survival, respectively
(Fig. 4f, g, Supplementary Data 3). TFE3 fusion partners, such as
MATR3 and SETD1B, were also identified as key regulators supporting
UOK109 cell survival (Fig. 4g), which suggests that they may utilize
similar RNA binding features to support oncogenic transcription.

Among the candidates identified from both TurboID and CRISPR
screens, PSPC1 emerged as a prominent candidate (Fig. 4b, g). By
plotting sgRNA counts identified from UOK109 and 786 O cells, PSPC1
stood out as the top candidate, exhibiting the most differential
enrichment of sgRNA between these two cell lines. (Fig. 4g, Supple-
mentary Fig. 5e). PSPC1 is an RNA binding protein implicated in tran-
scriptional regulation, RNA metabolism, and cellular stress
response[22,43]. We performed co-condensation experiments by expres-
sing mCherry-PSPC1 in UOK109-KI cells bearing a GFP tag at the
endogenous loci of NONO-TFE3. We detected strong co-condensation
between PSPC1 and NONO-TFE3, but not between PSPC1 and NLS-
mCherry (as a nuclear protein control) (Fig. 4h). This co-condensation
feature was further confirmed using an in vitro co-condensation assay
with recombinant protein (Fig. 4i). To further validate the role of PSPC1
in supporting tRCC cell growth, we depleted PSPC1 using shRNA-
mediated knockdown (Supplementary Fig. 5f) followed by a colony
formation assay in both UOK109 (with NONO-TFE3 fusion) and 786-O
(without fusion) cells. As a control for cell type specificity, we also
knocked down SFPQ (Supplementary Fig. 4f), identified as crucial for
both UOK109 and 786-O cell survival in our functional genomic study.
Depletion of PSPC1 selectively suppressed UOK109 cell growth, while

SFPQ knockdown suppressed both UOK109 and 786-O cell growth
(Fig. 4j). Taken together, these findings indicate that PSPC1 is a potent
candidate that could co-condensate with NONO-TFE3 and selectively
facilitate tRCC cell growth.

## PSPC1 and NONO-TFE3 co-condensation regulates genomic
## binding of RNAPII
To further evaluate whether co-condensation between PSPC1 and
NONO-TFE3 supports oncogenic transcription in tRCC, we performed
CUT&Tag experiments using antibodies against various RNA poly-
merase II (RNAPII) forms, including total RNAPII, and serine 2 (S2) or
serine 5 (S5) phosphorylated RNAPII (Fig. 5a, and Supplementary
Data 4). Phosphorylation at S2 and S5 of the C-terminal domain (CTD)
of RNAPII is known to regulate transcriptional elongation and initia-
tion, respectively[49]. In parallel, CUT&Tag using antibodies against
histone 3 lysine 27 acetylation (H3K27ac) and histone 3 lysine 4 tri-
methylation (H3K4me3) were used to pinpoint the actively transcribed
genomic regions (Fig. 5a). We compared the genomic enrichment of
NONO-TFE3 with RNAPII in UOK109 cells and found that 65%, 82%, and
56% of NONO-TFE3 binding sites were co-occupied by total RNAPII,
RNAPII-S5, and RNAPII-S2, respectively (Supplementary Fig. 6a). The
high occupancy of RNAPII-S5 at NONO-TFE3 binding sites suggests
that NONO-TFE3 might be involved in active transcription. This is
further confirmed by comparing SLAM-seq data with RNAPII CUT&Tag
data (Supplementary Fig. 6b).

Since PSPC1 is identified as a key component in NONO-TFE3
condensates, we evaluated the genomic distribution of PSPC1 using
the CUT&Tag method and compared it with NONO-TFE3 genomic
distribution. More than 90% (4567 out of 5011) of NONO-TFE3 binding
sites were also occupied by PSPC1 (Fig. 5b). At the transcriptional
level, ~64% (500 out of 786) of genes regulated by NONO-TFE3 showed dif-
ferential expression upon PSPC1 knockdown in UOK109 cells (Fig. 5b,
and Supplementary Fig. 6c, d, Supplementary Data 4). Among the 500
overlapping DEGs, 403 genes are downregulated in NONO-TFE3 or
PSPC1-depleted cells. The expression of these NONO-TFE3 and PSPC1
co-downregulated genes ($n = 403$) was also significantly upregulated in
tRCC patients, as indicated by previously published data[41] (Fig. 5c). To
further confirm the pro-oncogenic function of NONO-TFE3 and PSPC1
targeted genes, we selected *hexokinase 2 (HK2)* and *ribonucleotide
reductase M2 (RRM2)* as examples (Fig. 5d, e). Both HK2 and RRM2 are
recognized as potential therapeutic targets in various cancer types[50-52],
with inhibitors developed for cancer intervention[53,54]. Indeed, CHO29
(an RRM2 inhibitor)[55] or 3-BP (an HK2 inhibitor)[56] significantly sup-
pressed colony formation of UOK109 cells (Fig. 5d), suggesting the role
of NONO-TFE3 and PSPC1 co-regulated genes in supporting pro-
oncogenic transcription in tRCC.

Next, we assessed the impact of NONO-TFE3 or PSPC1 depletion
on RNAPII occupancy at 115 NONO-TFE3 primary target genes

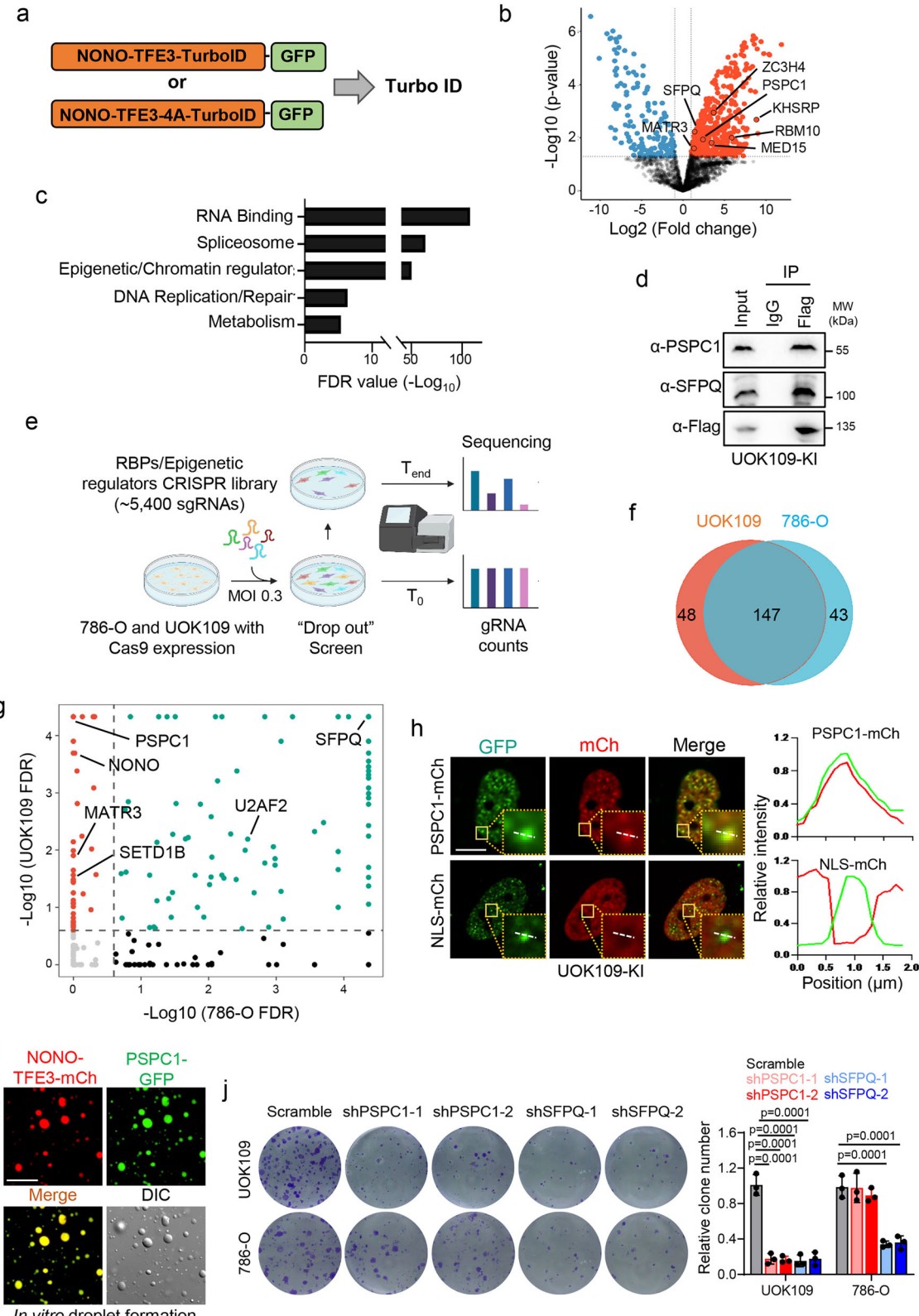

identified from Fig. 2f. Compared to the control group, PSPC1 depletion led to the upregulation of 1,609 genes and downregulation of 1064 genes, respectively (Supplementary Fig. 6c, Supplementary Data 4). Many of these DEGs are involved in DNA replication, cell cycle and metabolic pathways (Supplementary Fig. 6d). Upon depletion of NONO-TFE3 (using dTAG13 treatment) or PSPC1 (via shRNA), we observed a significant reduction in total RNAPII, RNAPII-S2, and

RNAPII-S5 occupancy at these NONO-TFE3 primary target genes (Fig. 5e–h). By contrast, genes not primarily regulated by NONO-TFE3 showed no changes in RNAPII binding upon either NONO-TFE3 (dTAG-13 treatment) or PSPC1 (via shRNA) depletion (Fig. 5f–h). These results suggest that NONO-TFE3 and PSPC1 play important roles in maintaining RNAPII genomic binding, thereby facilitating pro-oncogenic transcription to support tRCC cell growth.

**Fig. 4 | Co-condensation of NONO-TFE3 and PSPC1 is essential for tRCC growth. a** Schematic showing TurboID-based proximity biotin labeling to identify potential interacting proteins for NONO-TFE3 or NONO-TFE3-4A mutant. **b** Volcano plot showing proteins preferentially interacting with NONO-TFE3 over NONO-TFE3-4A (logFC > 1 or < −1 and $p < 0.05$, two-sided, unpaired Student's t-test). **c** The Gene Ontology (GO) analysis on the proteins that interact with NONO-TFE3, but not NONO-TFEF3-4A mutant (red dots in Fig. 4b). Top 5 enriched categories are listed. **d** Validation of NONO-TFE3 interaction with PSPC1 or SFPQ by co-immunoprecipitation (Co-IP) in UOK109 KI cells. ($n = 3$ independent biological replicates). **e** Schematic representation of the CRISPR screening process. Created in BioRender. Suris, A. (https://BioRender.com/zonrc7c). **f** Venn diagram showing the candidates number identified from the CRISPR screening with UOK109 and 786-O

cell lines. **g** Plot showing the UOK109 cell specific (red dots), 786-O specific (black dots), and shared (green dots) candidates identified from the CRISPR screening. **h** Representative images (left) and line profile analysis (right) of endogenous NONO-TFE3 (GFP labeled) and mCherry (mCh) labeled PSPC1 co-condensation droplet in UOK109 KI cells. mCherry (mCh) alone was used as control. $n = 3$ independent biological replicates; Scale bar, 10 μm. **i** In vitro co-condensation assay using recombinant proteins of mCherry-tagged NONO-TFE3 and GFP-tagged PSPC1. Scale bar, 10 μm. $n = 4$ independent biological replicates. **j** Representative images (left) and quantification (right) of clone formation assay results for UOK109 or 786-O cells expressing the indicated shRNAs. ($n = 3$ independent biological replicates; one-way ANOVA with Tukey's post-hoc test). Data are shown as mean ± SD. Source data are provided as a Source Data file.

## PSPC1, NONO-TFE3 and RNA synergize to enable RNAPII to form condensates on chromatin

Our data suggests that NONO-TFE3 and PSPC1 form co-condensates on chromatin, and deletion of either protein reduces RNAPII chromatin association. To explore whether the condensation of NONO-TFE3 enables RNAPII partitioning to form multivalent transcriptional hubs on chromatin, we utilized an engineered U2OS cell line with LacO arrays as described in Fig. 3. We observed that NONO-TFE3 and its ΔPLD variant enabled the formation of bright LacO-associated hubs containing RNAPII or PSPC1. By contrast, variants lacking CCD (ΔCCD), RRM2 (ΔRRM2), or with disrupted RNA binding (4A mutant) failed to form such hubs (Fig. 6a, b). Furthermore, in vitro co-condensation assays using purified RNAPII-CTD and NONO-TFE3 proteins indicated that NONO-TFE3 is essential for RNAPII-CTD condensation. Recombinant RNAPII CTD alone failed to form condensates but was efficiently incorporated into NONO-TFE3 condensates (Fig. 6c, and Supplementary Fig. 7a). The addition of PSPC1 further enhanced co-condensation, resulting in larger condensates than those formed by NONO-TFE3 and RNAPII CTD alone (Fig. 6c, d).

To evaluate the role of RNA in modulating condensate formation, we titrated increasing concentrations of RNA (10, 100, 1000 ng/μL) into these condensates (Fig. 6c, d). At low RNA concentration (10 ng/μL), we observed a moderate increase in condensate size and the co-condensation efficiency of NONO-TFE3 and RNAPII CTD, whereas a more substantial enhancement was seen in condensates containing PSPC1, NONO-TFE3, and RNAPII CTD. At an intermediate concentration (100 ng/μL), RNA disrupted the condensates formed by NONO-TFE3 and RNAPII CTD, but had a minor impact on those containing PSPC1, NONO-TFE3, and RNAPII CTD. At high RNA levels (1000 ng/μL), condensates containing PSPC1, NONO-TFE3, and RNAPII CTD were disrupted (Fig. 6c, d). Of note, both sense and antisense RNAs produced similar effects, suggesting that charge balance, rather than RNA sequence specificity, governs the condensate formation of the NONO-TFE3 complex (Supplementary Fig. 7b). These data align with prior studies suggesting that charge balance between RNA and protein is crucial for the equilibrium behavior of transcriptional condensates[21,39,40]. PSPC1 could counteract excessive RNA-induced disruption of condensates containing RNAPII CTD and NONO-TFE3, suggesting that PSPC1 stabilizes the condensates by neutralizing negatively charged RNAs. To further validate these findings, we performed droplet sedimentation assay[57] (Fig. 6e, top), which confirmed that RNAPII CTD was more effectively retained in condensates containing PSPC1/NONO-TFE3/RNAPII CTD compared to those lacking PSPC1. Moreover, consistent with co-condensation results, low RNA concentration (10 ng/μL) enhanced RNAPII CTD incorporation efficiency into these condensates (Fig. 6e, bottom).

Hyperphosphorylation of the RNAPII CTD is required for its transcriptional activity in mammalian cells. To explore the synergistic roles of RNA, PSPC1 and NONO-TFE3 in promoting CTD phosphorylation, we performed an in vitro kinase assay[58]. The results revealed that the presence of either PSPC1 or RNA markedly enhanced CTD phosphorylation relative to control conditions lacking these

components. Notably, the combination of both PSPC1 and RNA resulted in the highest level of CTD phosphorylation (Fig. 7a). Together with the above co-condensation assays, these findings strongly indicate that enhanced molecular interactions within phase-separated condensates facilitate more efficient biochemical reactions by locally concentrating key components. Next, we investigated the impact of PSPC1 and RNA on transcription using an in vitro transcription system by using a DNA template containing a heteroduplex bubble, which is widely used as nucleic acid scaffold[59]. This template allows RNAPII to directly engage the single-stranded DNA within the bubble without requiring general transcription factors[59]. NONO-TFE3 was added to the reaction in the presence or absence of PSPC1 together with RNAPII, with the transcriptional activity measured by qRT-PCR (Fig. 7b). Under transcription-permissive buffer conditions, mixing all these components led to the formation of droplets containing the DNA template (Fig. 7c). NONO-TFE3 enhanced transcriptional output, while PSPC1 and NONO-TFE3 further boosted transcription by over 3-fold compared to the control (Fig. 7d). This enhancement was abolished when using the NONO-TFE3-4A mutant, regardless of PSPC1 presence (Fig. 7d). These data support our model that low levels of RNA produced in the reaction may work together with NONO-TFE3 and PSPC1 to support RNAPII-mediated transcription. At last, we adopted a Gal4DBD-mediated luciferase reporter assay to assess transcriptional activity in a cellular context[60]. In this assay, luciferase expression is driven by the interaction between the DNA binding domain (DBD) of the yeast transcription factor GAL4 and multiple GAL4-responsive upstream activating sequences (UAS) (Fig. 7e). We observed higher luciferase activities in cells expressing Gal4DBD-NONO-TFE3 or its ΔPLD variant compared to other NONO-TFE3 variants, including ΔCCD, ΔRRM2 and 4A variants, suggesting that NONO-TFE3 condensation is critical for supporting robust transcription (Fig. 7f, and Supplementary Fig. 7c). Interesting, we also noted enhanced luciferase activity in cells co-expressing PSPC1 with GAL4DBD-NONO-TFE3 or its ΔPLD variant; whereas such enhancement was not observed in other NONO-TFE3 variants that failed to form condensates (Fig. 7f). In summary, these data strongly suggest that RNA, owing to its high negative charge density, serves as a potent regulator of condensates formed through electrostatic interactions. PSPC1 not only neutralizes the charge repulsion exerted by excessive RNA, which could otherwise expel RNAPII CTD, but also makes use of RNAs to facilitate condensates formation conducive to RNAPII engagement. This, in turn, efficiently compartmentalizes RNAPII CTD, enhancing its phosphorylation by CDKs and ultimately boosting transcriptional output. Therefore, the cooperative interplay between PSPC1 and RNA in promoting RNAPII-containing condensates represents a crucial regulatory mechanism governing the transcriptional activity of NONO-TFE3 (Fig. 7g).

## Dissolution of TFE3 oncofusion condensates curtails tRCC

Given the pivotal role of TFE3 oncofusion condensates in driving pro-oncogenic transcription, we hypothesized that disrupting this process

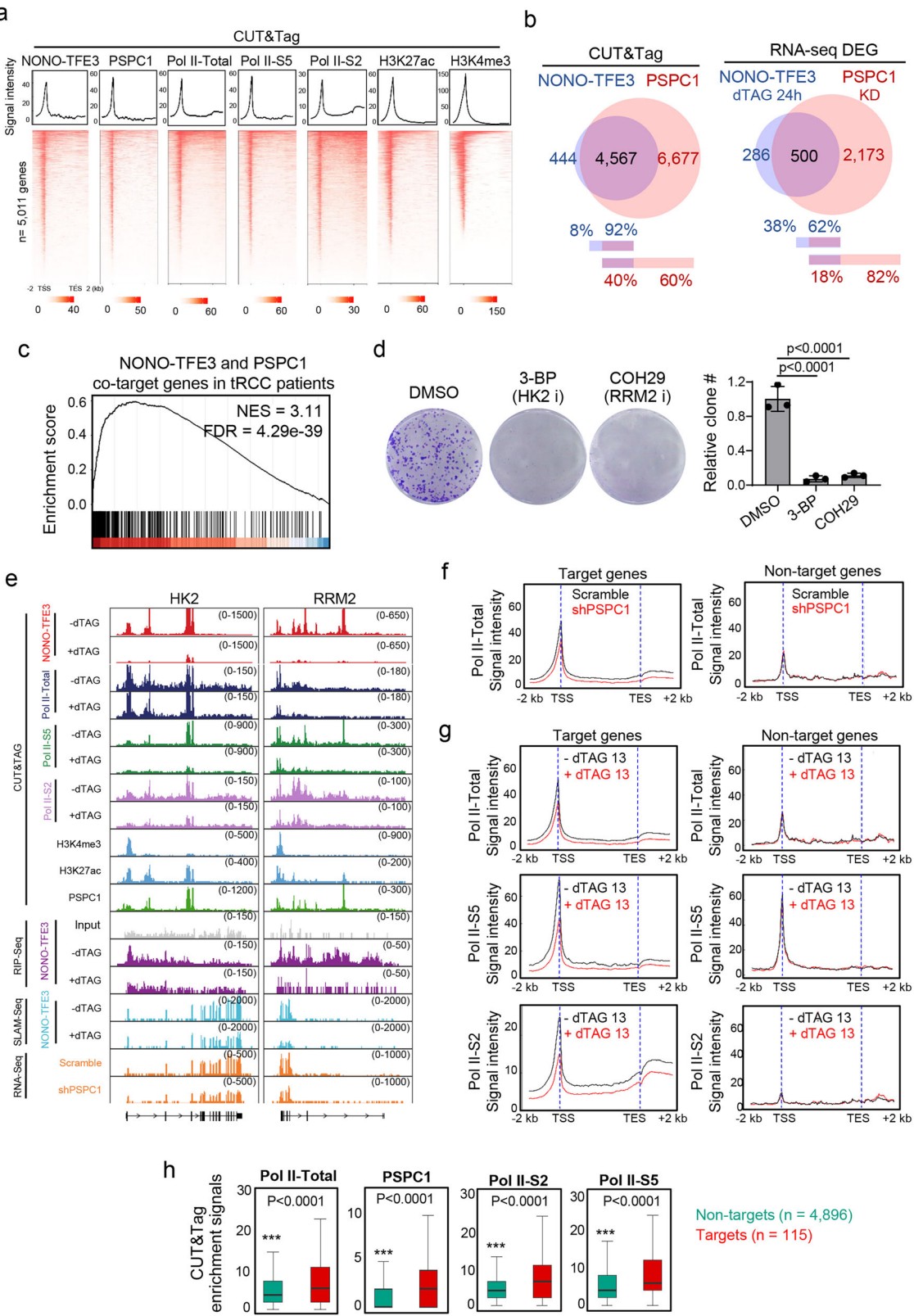

might suppress tRCC cell growth. To test this hypothesis, we utilized maltose-binding protein (MBP) to enhance the solubility of target proteins[61]. The efficacy of MBP in dissolving biomolecular condensates has been validated using FUS-IDR as a proof-of-concept[62]. To target endogenous GFP-tagged NONO-TFE3 in UOK109-KI cells (Fig. 8a, and Supplementary Fig. 8a), we fused mCherry-MBP to a LaG nanobody that specifically recognizes GFP[63]. With this design, we found that

mCherry-LaG-MBP, but not mCherry or mCherryl-LaG, could effectively prevent GFP-NONO-TFE3 condensate formation (Supplementary Fig. 8b, c) and substantially suppressed UOK109 cell growth (Supplementary Fig. 8d). Given that the constitutive expression of LaG-MPB potently affected cell growth, we decided to carry out a similar experiment using a chemically inducible split nanobody (Chessbody) approach reported previously[64]. In this strategy, LaG was split into two

**Fig. 5 | PSPC1 co-condensates with NONO-TFE3 to recruit RNAPII. a** Heatmaps showing the binding of NONO-TFE3, PSPC1, RNAPII-total, RNAPII-Ser5p (Pol II-S5), RNAPII-Ser2p (Pol II-S2), H3K27ac and H3K4me3 from −2 kb TSS to +2 kb TES at 5,011 genes occupied by NONO-TFE3. **b** Venn diagram showing the overlapping genes occupied by NONO-TFE3 and PSPC1 in UOK109 KI cells (left) and overlapping differentially expressed genes (DEGs) identified from RNA-seq in UOK109 KI cells treated with DMSO or dTAG-13 for 24 h (NONO-TFE3 regulated genes) or UOK109 KI cell with and without PSPC1 knocking down (KD) (PSPC1 regulated genes) (right). **c** Gene set enrichment analysis of overlapped DEGs (*n* = 500) identified from (**b**) in tRCC patient sample cohorts[41]. **d** Representative images (left) and quantification (right) of clone formation assay for UOK109 KI cells treated with DMSO or inhibitors against HK2 (HK2 i) or RRM2 (RRM2 i). *n* = 3 independent biological replicates; one-way ANOVA with Tukey's post-hoc test. Data are shown as mean ± SD. **e** Genome browser views of the indicated sequencing results at representative loci

in UOK109 KI cells upon treatments. *n* = 2 independent biological replicates. **f** Metagene binding profiles of RNAPII-total in control (shScramble, black line) and PSPC1KD (shPSPC1, red line) UOK109 KI cells at NONO-TFE3 regulated (left, *n* = 500) or non-regulated genes (right, *n* = 10,113) (from −2 kb TSS to +2 kb TES) identified from RNA-seq. **g** Metagene binding profiles of RNAPII-total (Pol II-total), RNAPII-Ser5p (Pol II-S5), RNAPII-Ser2p (Pol II-S2) in DMSO (black line) or dTAG-13 treated (500 nM, 4 h, red line) UOK109 KI cells at NONO-TFE3 regulated (left, *n* = 500) or non-regulated genes (right, *n* = 10,113) (from −2 kb TSS to +2 kb TES) identified from RNA-seq. **h** Binding enrichment of PSPC1, RNAPII-total, RNAPII-Ser5p (Pol II-S5), RNAPII-Ser2p (Pol II-S2) at NONO-TFE3 target (*n* = 115) or non-target genes (*n* = 4896) identified from SLAM-seq. The box plots indicated the median (center line), the third and first quartiles (box limits) and 1.5x interquartile range (IQR) above and below the box (whiskers). (*n* = 2 independent biological replicates; two-sided Wilcoxon test). Source data are provided as a Source Data file.

parts: its N-terminal domain tagged by cpFRB and the C-terminal domain fused with FKBP and MBP (Fig. 8b, and Supplementary Fig. 8e). Rapamycin-induced FKBP-cpFRB heterodimerization brought the split LaG parts into close proximity to form an intact functional LaG nanobody, thereby restoring its interaction with GFP-tagged proteins. This chemogenetic tool, termed CB-LaG-MBP, allowed us to temporally control MBP-target interactions for inducible condensate dissolution in both UOK109 (NONO-TFE3 fusion) and UOK145 (SFPQ-TFE3 fusion) cells (Fig. 8c, and Supplementary Fig. 8f). Rapamycin-inducible disruption of TFE3 oncofusion condensation was accompanied by suppression of tRCC cell growth, as demonstrated by both in vitro colony formation assays (Fig. 8d, Supplementary Fig. 8g) and mouse xenograft models in vivo (Fig. 8e, f). We further performed RNA-seq analysis on UOK109 cells expressing CB-LaG-MBP following rapamycin treatment. Disruption of NONO-TFE3 condensation by rapamycin induced transcriptomic changes that closely mirrored those observed upon dTAG-13 treatment (Fig. 8g–i, and Supplementary Fig. 8h). These results provide additional support for the conclusion that the condensate-forming ability of NONO-TFE3 is essential for its transcriptional activity. Together, by taking a chemogenetic approach, we have demonstrated the feasibility of manipulating the dissolution of TFE3 oncofusion condensates to effectively curtail tRCC cell growth both in vitro and in vivo.

## Discussion

Condensation mediated by oncofusion proteins has been increasingly documented in recent studies[2,5,15,16]. It is widely recognized that intrinsically disordered regions (IDRs) within oncofusion proteins promote the formation of liquid-like condensates to facilitate oncogenic events[65]. However, liquid-like condensates are characterized by multivalency and may contain diverse biomolecules, such as RNA. Several recent studies have reported the role of RNA binding proteins (RBPs) and RNA in both the formation and stabilization of biomolecular condensates. Dysregulation of RBPs and abnormal RNA production can lead to the re-localization of biomolecular condensates and contribute to disease progression[20,21].

Here, we have discovered that RNA binding capability conferred by the fusion partners of TFE3 supports the formation of oncofusion condensates, which ultimately drives pro-oncogenic transcription to promote malignancy. Through detailed domain mapping, we revealed that both the disordered coiled-coil domain (CCD) and the highly ordered RNA binding motifs (RRM) within the TFE3 fusion partner NONO contribute to condensate formation. Using site-directed mutagenesis and fusing TFE3 oncofusions with RNase A, we disrupted the RNA binding capabilities of TFE3 oncofusion proteins, resulting in the abolishment of condensates formation. Furthermore, using a reporter system, we uncovered a regulatory feedback loop in which newly transcribed RNAs influence the transcriptional activity of TFE3 oncofusions. However, we also recognized the limitations of this

system, as it does not accurately replicate how RNA production regulates NONO-TFE3 transcriptional activity at the endogenous locus. Given current technical challenges, developing a more physiologically relevant in situ model will be necessary to thoroughly investigate this potential feedback mechanism. These findings converge to support the notion that, beyond IDRs, the biochemical properties of structurally ordered regions such as RNA binding motifs could play a pivotal role in biomolecular condensate formation.

In the current study, we also compared the genomic binding profiles between the TFE3 oncofusion protein, specifically NONO-TFE3, and nuclear-localized TFE3 (TFE3-S321A mutant). Interestingly, our data showed that NONO-TFE3 and TFE3-S321A share the majority of genomic binding sites, with NONO-TFE3 exhibiting stronger enrichment than TFE3-S321A. This suggests that the fusion partner likely reinforces TFE3-mediated transcriptional activity through transcriptional condensate formation rather than by altering its genomic distribution. These findings align with the insufficient rescue phenotype observed in UOK109-KI cells expressing TFE3-S321A under dTAG-13 treatment. It is worth noting that this observation does not conflict with the established oncogenic role of nuclear TFEB/TFE3 activation in renal cancer[66]. Previous studies have reported that kidney-specific overexpression of TFEB in genetically modified mouse models leads to malignant transformation, supporting the oncogenic role of nuclear activation of TFE transcription factors. We compared published RNA-seq data from genetically modified mouse models with either TFE3 overexpression or ASPSCR1-TFE3 transgene expression[41,66] (Supplementary Fig. 9). Although both models developed renal cancer, they exhibited distinct transcriptional landscapes, with less than 15% overlap in differentially expressed genes between the two models. We further compared the expression levels of canonical TFEB/TFE3 target genes[41,67,68] using these RNA-seq datasets and found that their expression was notably higher in the tRCC mouse model compared to the TFEB transgenic model. This indicates a more strongly activated TFEB/TFE3-driven transcriptional program in tRCC, potentially reflecting the increased transcriptional activity conferred by the oncofusion proteins compared to their wild-type counterparts. Collectively, these data suggest that while nuclear activation of TFE family members has an oncogenic function in renal cancer generally, TFE3 oncofusion proteins are specifically required to mediate malignant transformation in tRCC.

We acknowledge the limitations of evaluating TFE3-S321A genomic binding in UOK109-KI cells treated with dTAG-13. The CUT&Tag experiments were performed in in vitro cultured UOK109 tRCC cell lines, where the chromatin environment may have been altered due to long-term passage and multi-generational culture. Because both fusion and non-fusion TFE3 proteins share the same DNA-binding domains, they may adapt similarly to such chromatin environments. Therefore, it remains unclear whether fusion partners can drive TFE3 binding toward distinct genomic regions during tumor initiation. Further

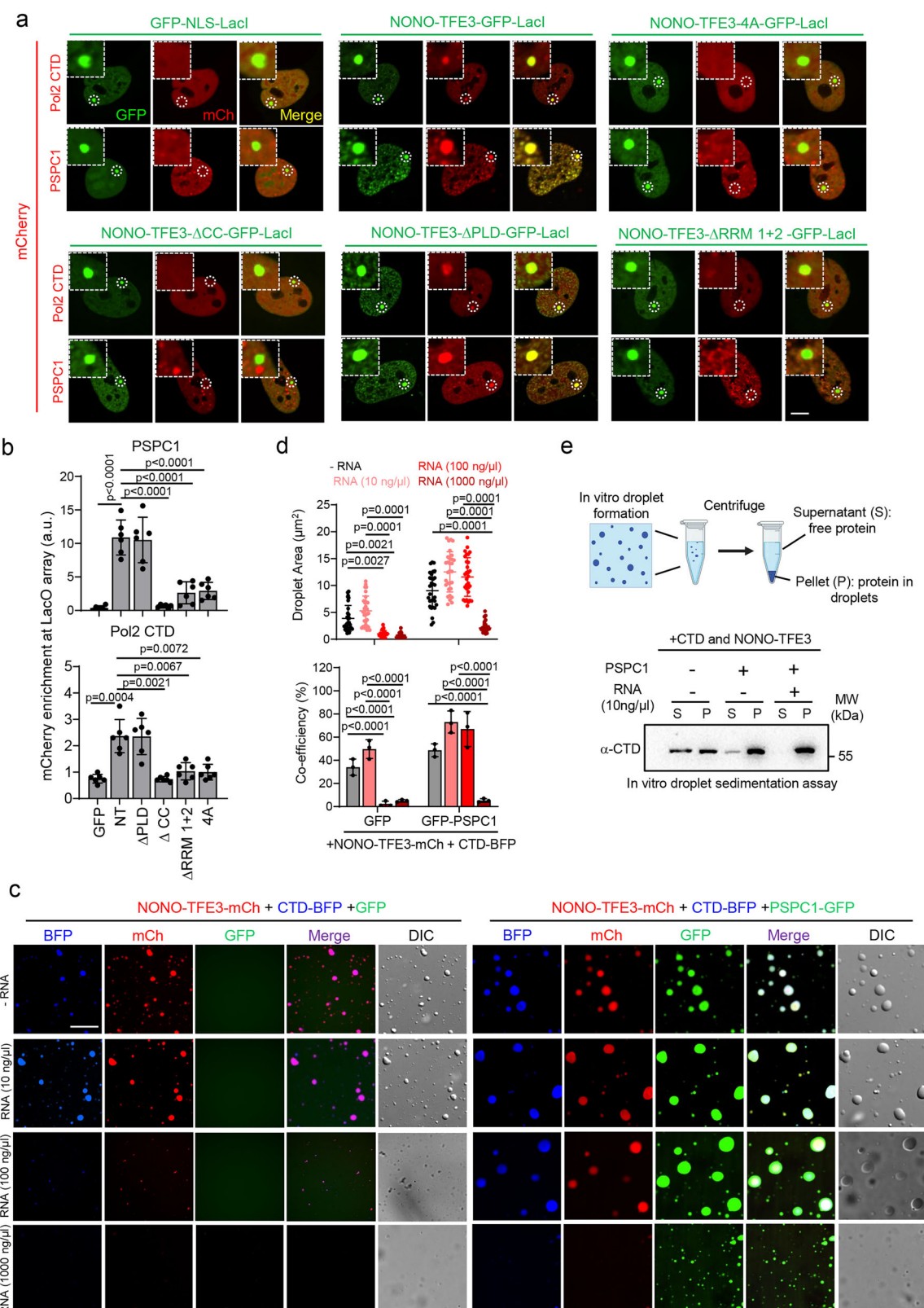

investigations will be required to address this caveat. In addition to RNA, our study has revealed that many other RNA binding proteins could interact with NONO-TFE3. Our functional genomic screen has demonstrated the indispensable roles of RNA binding proteins in enhancing condensate formation and supporting tRCC cell growth. We have focused on PSPC1 as an example to demonstrate its ability to co-condensate with the TFE3 fusion protein and RNAPII, thereby establishing transcriptional hubs to amplify transcriptional outputs. Notably, SFPQ, a close family member of NONO and PSPC1, has been identified as essential for the survival of both TFE3 fusion and non-fusion kidney cancer cells. This dependency could be attributed to its fundamental roles in RNA splicing, polyadenylation RNA stabilization, and DNA damage repair[22], which are important for the viability of many cell types and not readily compensated for by other proteins.

**Fig. 6 | PSPC1 co-condenses with NONO-TFE3 and RNAPII on chromatin.**
**a, b** The LacI-LacO array for multivalent heterotypic interaction detection between NONO-TFE3 truncations and PSPC1 or RNAPII C-terminal domain (Pol2 CTD) on genomic loci (**a**). NONO-TFE3 truncates-LacI are labeled with GFP, and PSPC1 or Pol2 CTD are labeled with mCherry. The LacO array locus is circled and zoom-in LacO array locus images are shown (white box). The quantifications of enrichment (**b**) were performed with 6 cells from 3 independent biological replicates. (one-way ANOVA with Tukey's post-hoc test). Scale bar, 10 μm. Data are shown as mean ± SD. **c** Representative images of in vitro droplets formed by mCherry-fused NONO-TFE3, BFP-fused RNAPII CTD, and GFP-fused PSPC1 with the indicated concentrations of RNA. GFP alone was used as control. Scale bar, 10 μm. **d** Quantifications of droplet area (top) and co-condensation efficiency (bottom) from in vitro droplet formation assay shown in (**c**). (n = 30 droplets from 3 independent biological replicates; two-sided unpaired Student's t-test). Data are shown as mean ± SD. **e** Schematic illustration of the droplet sedimentation assay (top) and immunoblotting of CTD incorporation in droplets (bottom). n = 3 independent biological replicates. Created in Created in BioRender. Suris, A. (https://BioRender.com/zwfnjs8). Source data are provided as a Source Data file.

Furthermore, our TurboID-based proximity proteomic studies have unveiled that other TFE3 fusion partners, such as SFPQ, MATR3, RBM10, ZC3H4, KHSRP1, and MED15, also stand out as prominent RNA-dependent binding partners of NONO-TFE3. These results reinforce the conclusion that RNA and RNA binding protein-mediated transcriptional co-condensation may represent a universal mechanism underpinning the pro-oncogenic function of TFE3 oncofusions found in tRCC patients. It is important to note that, given the existence of multiple fusion partners for TFE3, the mechanism we identified with NONO-TFE3 may represent only some of the key characteristics of these oncofusions. We cannot rule out the possibility of other mechanisms specific to different fusion types, which remain to be fully characterized and warrant further investigation. Despite the current lack of drugs and pharmacological approaches with high specificity to effectively perturb nuclear condensates in tRCC, our chemogenetic studies employing split nanobody-MBP as a prototype point to the exciting possibility of selective dissolution of oncogenic condensates for cancer intervention. In addition, targeting RNA binding proteins that facilitate nuclear condensate formation might offer another promising approach for tRCC treatment.

## Methods

### Cell culture and lentivirus generation
HEK293T cells (CRL-3216) and 786-O cells (CRL-1932) were purchased from ATCC and cultured in DMEM medium (Corning, Manassas, VA, USA) supplemented with 10% fetal bovine serum (Omega, Tarzana, CA, USA), 1% antibiotics (100 IU penicillin and 100 μg/mL streptomycin) and 2 mM L-glutamine. UOK109, UOK145 and UOK146 cells were kindly provided by Dr. W. Marston Linehan (National Cancer Institute). U2OS cells with Lac operators were kindly provided by Dr. David Spector (Cold Spring Harbor Laboratory). All these cells are cultured in DMEM medium supplemented with 10% fetal bovine serum. For virus production, $5 \times 10^6$ of HEK293T cells were plated in 10-cm plates and co-transfected with lentiviral plasmids encoding targeting genes or sgRNA together with pMD2.G and psPAX2 using iMFectin poly DNA transfection reagent (GenDEPOT) following the manufacturer's instructions. Supernatant containing viral particles was collected 48 h after transfection, filtered with 0.22 μm syringe filters, and subsequently concentrated by ultra-centrifugation. For infection, virus was added to cells at 70% confluency and fresh media were changed 24 h after infection. After 3-day puromycin selection or sorting by appropriate fluorescent protein markers (GFP or mCherry), cells were maintained for at least one additional day without drug for further experiments. For inducible expression systems, doxycycline was added to the culture medium at 1 μg/ml for 24 or 48 h as needed.

### Generation of knockin (KI) cell lines
M1 and M2 tagging oligos targeting the 5' and 3' homology arm of human TFE3 locus were designed using the Mammalian PCR tagging website (http://www.pcr-tagging.com) based on a previous study[69]. The pMaCTag-05 plasmid was purchased from Addgene (#119984). The GFP-FKBP12(F36V)-FLAG was introduced by replacing the eGFP fragment with GFP-FKBP12(F36V)-FLAG using the BamHI and SpeI restriction enzymes. The M1 and M2 tagging oligos were then used to amplify the GFP-dTAG-FLAG fragment using the pMaCTag-05 plasmid as the template to obtain a PCR cassette containing homology arms to the target TFE3 locus and a functional crRNA for gene cleavage. UOK109 cells were transfected with 500 ng of enCas12a (Addgene # 89351) along with 500 ng of purified PCR cassettes. Following transfection, GFP-positive UOK109 cells were selected by fluorescence-activated cell sorting (FACS) 72 h post-transfection. Single clone was selected and further confirmed by sequencing the genomic DNA region spanning the knockin junction site at the C-terminus of NONO-TFE3 and immunoblot with anti-Flag and anti-GFP antibodies.

### Colony formation assay
A total of 500 cells per well were seeded into a 6-well plate and treated with either DMSO (control) or 500 nM dTAG-13. For the washout groups, UOK109-KI cells were pre-treated with dTAG-13 for 24 h and then switched back to normal DMEM without dTAG-13 for another 24 h. Subsequently, cells were re-seeded with a total of 500 cells per well on the same day as seeded for the DMSO and dTAG-13 groups. Growth media containing dTAG-13 or DMSO was replenished daily for the following 14 days, with the culture assessed via crystal violet staining at the endpoint.

### RNA-seq and data analysis
Total RNA was extracted using the Qiagen RNeasy kit following the manufacturer's instructions. 1 μg total RNA was subjected to mRNA selection with Poly(A) mRNA Magnetic Isolation Module (NEB #E7490) and then subjected to library construction using the NEBNext Ultra Directional RNA Library Prep Kit (NEB #7760) by following the manufacturer's instructions. The quality of libraries was monitored by Agilent High Sensitivity DNA kit (Agilent Technologies) and then sequenced on an Illumina Nova-seq platform with a paired-end mode (150/8/8/150).

RNA-seq reads were aligned to the human genome (GRCh38) using HISAT2 (v2.2.1)[70] with "--rna-strandness RF". Gene expression was calculated using featureCounts (v2.0.3) with "-s 2". Differentially expressed genes (DEGs) were identified using DESeq2 package (v1.38.3)[71] with the criteria of |log2FC | >1 and FDR < 0.05. Raw bam files were converted to bigwig format using bamCoverage (v3.5.4)[72] with "--normalizeUsing RPKM" to visualize in Integrated Genome Viewer (IGV, v2.13.2)[73].

### CUT&Tag and data analysis
$1 \times 10^5$ cells were used as the starting material for the CUT&TAG library preparation using the Hyperactive Universal CUT&TAG Assay Kit for Illumina Pro (Vazyme, TD904) following the manufacturer's instructions. Briefly, cells were immobilized on ConA beads and subjected to digitonin permeabilization, enabling the binding of primary and secondary antibodies to specific genomic sites. After digestion of genomic DNA by the protein A-Tn5 (pA-Tn5) transposase pre-loaded with sequencing adapters, the captured DNA fragments were then purified by magnet beads and utilized for sequencing library amplification by PCR with a unique index. Prior to sequencing, the libraries were assessed for size distribution and concentration, and subsequently sequenced on NovaSeq to generate paired-end reads of $2 \times 150$ bp.

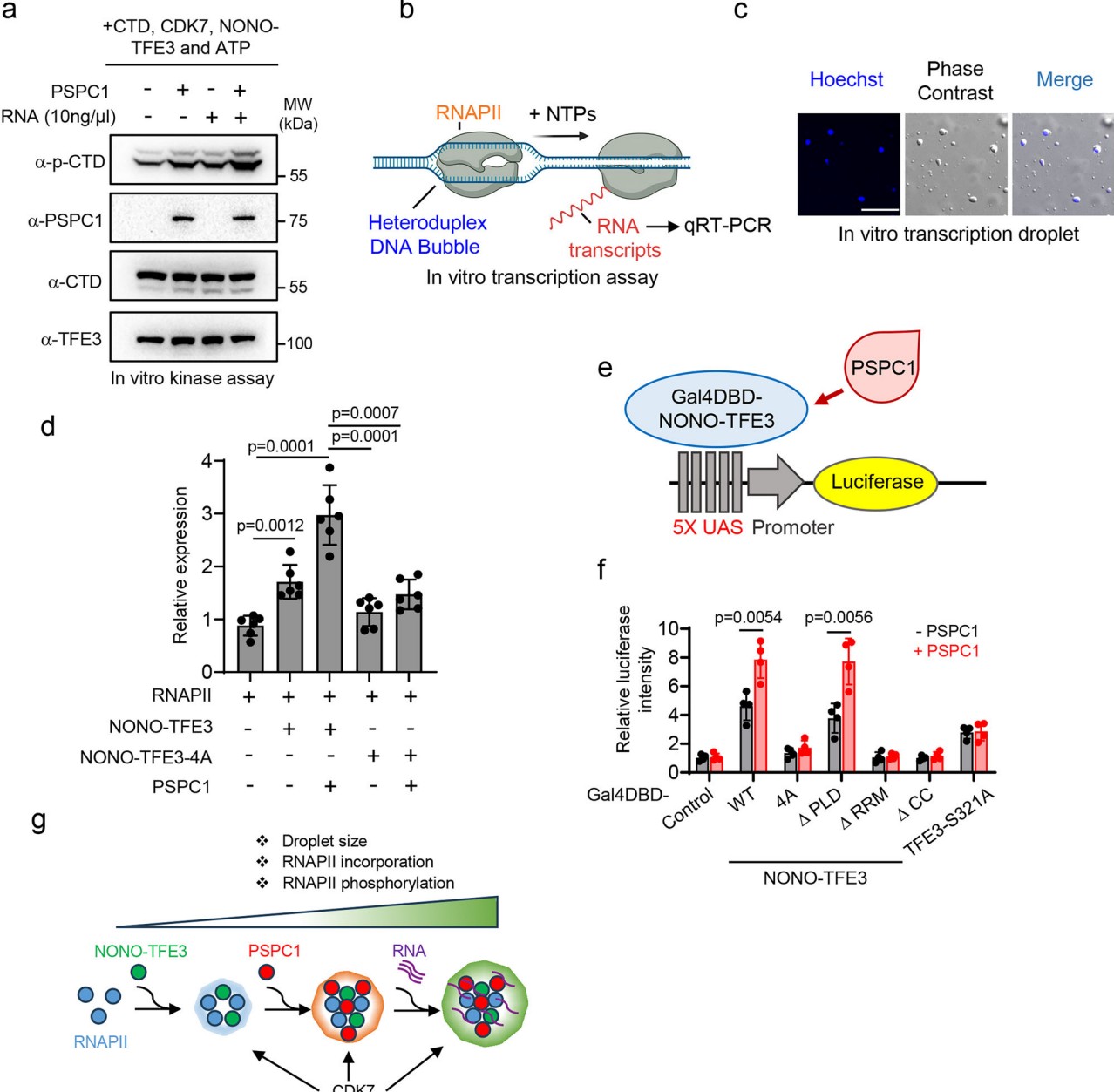

**Fig. 7 | Synergistic effects of PSPC1, RNA and NONO-TFE3 on transcription. a** In vitro kinase assay using recombinant proteins as indicated. Phosphorated CTD (p-CTD) and other components were determined by immunoblotting using their corresponding antibodies. $n = 3$ independent biological replicates. **b** Schematic illustration of the in vitro transcription assay. Created in BioRender. Suris, A. (https://BioRender.com/i1s8xt7). **c** Images of droplets formed during the in vitro transcription reaction. Hoechst was used to stain the DNA component within the droplets. Scale bar, 10 µm. $n = 3$ independent biological replicates. **d** Quantifications of in vitro transcription reaction activity (normalized to RNAPII only) with the indicated recombinant proteins. ($n = 6$ independent biological replicates; two-sided unpaired Student's t-test). Data are shown as mean ± SD. **e** Schematic illustration of the luciferase reporter assay for assessing gene transcription. Constructs encoding the truncated NONO-TFE3 variants fused with the Gal4 DNA binding domain (Gal4DBD) were co-transfected into HEK293T cells

bearing a luciferase reporter and containing 5 repeated GAL4-responsive upstream activating sequences (UAS) in the presence or absence of PSPC1. **f** Quantification of luciferase signals (normalized to the control) using NONO-TFE3 truncated variants or the TFE3 S321A mutant, with or without PSPC1. ($n = 4$ independent biological replicates; two-sided unpaired Student's t-test). Data are shown as mean ± SD. **g** Schematic diagram showing the interplay among NONO-TFE3, PSPC1 and RNA in promoting droplet formation and subsequent CTD incorporation and phosphorylation. RNAPII CTD alone exhibits limited ability to form condensates. Co-condensation with NONO-TFE3 and PSPC1 enhances droplet formation, resulting in larger condensates. RNA further cooperates with NONO-TFE3 and PSPC1 to markedly promote condensate assembly and CTD incorporation, thereby concentrating CTD and CDKs to facilitate more efficient phosphorylation. Source data are provided as a Source Data file.

CUT&Tag reads were aligned to the human genome (GRCh38) using bowtie2 (v 2.5.1)[74] with "--end-to-end --very-sensitive" mode, and sorted by samtools (v1.6)[75]. PCR duplicates were marked and removed using the picard pipeline (v2.27.4, https://broadinstitute.github.io/picard/). Peak calling was conducted using MACS2 (v2.2.9.1)[76] with default narrow or broad calling parameters (q-value < 0.001), and peaks located within a distance of 5 kb around transcription start site were defined as promoter peaks. Reads on the peaks were counted using bedtools (v2.31.1)[77] and converted to reads per kilobase per million mapped reads (RPKM). CUT&Tag binding signals were normalized

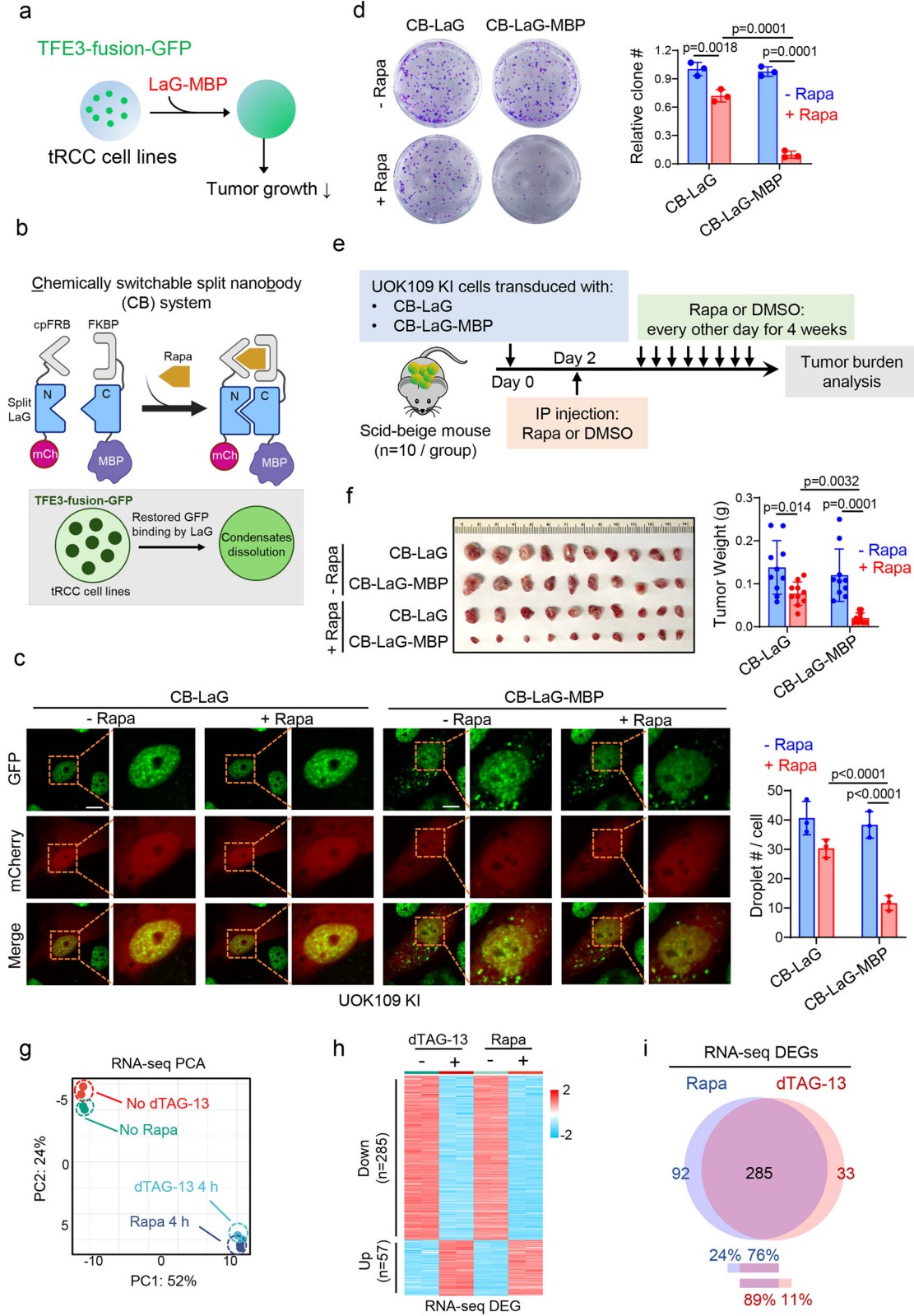

using deepTools bamCoverage (v3.5.4)[72] with murine spike-in supplied by kit for comparing samples, and visualized using deepTools computeMatrix (v3.5.4)[72] and Integrated Genome Viewer (IGV, v2.13.2)[73].

## SLAM-seq and data analysis

UOK109-KI cells were seeded in a 6-well plate and treated with 500 nM of dTAG-13 at the indicated time points. After dTAG treatment, 100 μM

of S4U was added to the medium for another 2 h in the absence of light, and samples were collected simultaneously. For washout group, dTAG-13 was added 2 days (~48 h) and removed by replacing with normal medium 1 day (~24 h) before sample collecting. Cell pellets were subjected to RNA extraction and iodoacetamide (IAA) treatment following the manufacturer's instructions (LEXOGEN, 061, SLAMseq Kinetics Kit - Anabolic Kinetics Module). In brief, total RNA was

**Fig. 8 | Disrupting condensation of TFE3 fusions inhibits tRCC growth.**
**a** Schematic depicting the strategy to employ MBP linked nanobody against GFP (LaG-MBP) to dissolve the condensates of TFE3 oncofusions. **b** The design of a chemically switchable split nanobody (CB) system. The GFP nanobody (LaG) is split to an N-terminus linked with cpFRB and a C-terminus linked with FKBP and MBP. Rapamycin (Rapa) promotes the interaction of cpFRB and FKBP to form functionally intact LaG-MBP and disrupt the condensation of GFP-labeled NONO-TFE3. Created in BioRender. Suris, A. (https://BioRender.com/jvewyzg). **c** Representative images (left) and quantification (right) of droplet formation for UOK109 KI cells transfected with the indicated constructs with or without rapamycin (Rapa) treatment. (*n* = 3 independent biological replicates; one-way ANOVA with Tukey's post-hoc test). Scale bar, 10 μm. Data are shown as mean ± SD. **d** Representative images (left) and quantification (right) of clone formation assay results for UOK109 KI cells

transduced with the indicated constructs with or without rapamycin (Rapa) treatment. (*n* = 3 independent biological replicates; one-way ANOVA with Tukey's post-hoc test). Data are shown as mean ± SD. **e** Experimental setup for monitoring tumor growth in vivo. **f** Representative images of isolated tumor masses (left) and quantification (right) of tumor weights at day 24 for the in vivo experiment shown in **e**. (*n* = 10 mice; one-way ANOVA with Tukey's post-hoc test). Data are shown as mean ± SD. **g, h** Principal component analysis (**g**) and heatmap (**h**) of RNA-seq data from UOK109 KI cells treated with dTAG-13 or rapamycin (Rapa). *n* = 2 independent biological replicates. **e** Venn diagram showing the overlap of differentially expressed genes (DEGs) identified from RNA-seq analysis of UOK109 KI cells treated with dTAG-13 or rapamycin (Rapa) for 4 h. Source data are provided as a Source Data file.

extracted using the TRIzol-Chloroform-Isopropanol method and treated with 100 μM IAA to alkylate the S4U-labeled RNA. Subsequently, 500 ng of IAA-treated RNA was utilized for library construction as per the product instructions (LEXOGEN, 192, QuantSeq 3′ mRNA-Seq Library Prep Kit V2 FWD with UDI 12 nt Set B1). The library concentration was further confirmed using the KAPA Library Quantification Kit (Roche, 07960336001). Sequencing of the library was performed by NovaSeq to generate paired-end reads of 2 × 150 bp.

SLAM-seq reads were aligned to the human genome (GRCh38) using STAR (v2.7.11a)[78] with default parameters except with "--alignEndsType EndToEnd --outSAMattributes nM MD NH". Quantification and differentially expressed analysis of nascent RNA were performed with GRAND-SLAM (v2.0.7b)[79] and grandR (v0.2.5) pipeline[80]. Genes with |log2FC| >1 and FDR < 0.05 were considered as statistically significant.

### RIP-seq and data analysis
Cell pellets collected from 5 million UOK109 cells were fixed by 1% formaldehyde for 10 min at room temperature. After fixation, the cell pellets were washed three times with ice-cold PBS and resuspended in 1× isolation buffer (3.2 M Sucrose, 10 Tris-HCl pH 7.5, 5 mM MgCl2, 1% Triton X-100) supplemented with 100 U/mL RNAseOUT (Invitrogen, 10777019), followed by incubation on ice for 20 min. The cells were then centrifuged at 2500 × *g* for 15 min at 4 °C, and the supernatant was discarded to isolate the cell nuclei. The pellet containing the cell nuclei was resuspended in RIP buffer (150 mM KCl, 25 mM Tris pH 7.4, 5 mM EDTA, 0.5 mM DTT, 0.5% NP40, 100 U/mL RNAse Inhibitor, 1× Protease Inhibitors), sonicated, and centrifuged at 10000 g for 10 min at 4 °C to remove debris. The resulting supernatant was subjected to immunoprecipitation using FLAG antibody (Sigma, F1804) and Pierce Protein A/G magnetic beads a 4 °C overnight. The beads were then washed three times with RIP buffer, and the bound RNA-protein complexes were eluted with 500 μL Elution buffer (10 mM Tris-HCl pH 7.5, 5 mM EDTA, 100 mM NaCl, 1% SDS) supplemented with proteinase K at a shaker incubator at 55 °C for 15 min at 1100 rpm. Subsequently, the eluate was subjected to TRIzol-chloroform-isopropanol RNA extraction, rRNA depletion and subjected to library preparation with the NEBNext Ultra II RNA Library Prep kit. The resulting library was validated for size and concentration using a bioanalyzer and then sequenced by NovaSeq to generate paired-end reads of 2 × 150 bp.

RIP-seq reads were aligned to the human genome (GRCh38) using HISAT2 (v2.2.1)[70] with "--rna-strandness RF", and sorted by samtools (v1.6)[75]. PCR duplicates were marked and removed using picard pipeline (v2.27.4, https://broadinstitute.github.io/picard/). Alignments on "+" and "-" strands were filtered separately to identify strand-specific peaks using MACS2 (v2.2.9.1)[76] with default parameters (q-value < 0.001), and peaks located within a distance of 5 kb around transcription start site were defined as promoter peaks. Reads on the peaks were counted using the summarizeOverlaps function from GenomicAlignments (v1.34.1)[81] package and converted to reads per kilobase per million mapped reads (RPKM). RIP binding signals were

RPKM-normalized using deepTools bamCoverage (v3.5.4)[72], and visualized using deepTools computeMatrix (v3.5.4)[72] and Integrated Genome Viewer (IGV, v2.13.2)[73].

### Recombinant protein expression and purification
The pET28-6×His-tagged constructs encoding NONO-TFE3, PSPC1 and RNAPII CTD were transformed into *Escherichia coli* BL21 component cells. Bacterial culture at OD600 of 0.6-0.8 were supplemented with 1 mM of isopropyl β-D-1-thiogalactopyranoside (IPTG) for 12 h at 16 °C for protein induction. For purifications of 6× His-tagged proteins, bacterial pellets from 1 L IPTG-induced culture were resuspended in 8 ml lysis buffer (50 mM Tris-HCl pH 7.3, 450 mM NaCl, 10 mM imidazole, 1× protease inhibitor cocktail; Roche, Catalog No. 11836170001) and lysed by sonication. After centrifugation at 11,000 g for 10 min to remove the debris, the clarified supernatant containing the protein of interest was loaded to a column packed with Ni-NTA beads (Thermo Scientific, 10038124), washed with the binding buffer, and eluted with a Ni-NTA elution buffer (50 mM Tris-Cl pH 7.9, 500 mM NaCl, and 500 mM imidazole). The eluted proteins were further concentrated and desalted by using a centrifugal filtration unit with a molecular weight cutoff of 30 kDa (Millipore, UFC803024). Protein concentration was determined by measuring the absorbance at 280 nm using a Nanodrop spectrophotometer.

### In vitro droplet formation assay
Recombinant mCherry, GFP or BFP fusion proteins were concentrated and desalted to appropriate salt concentrations using Amicon Ultra-4 centrifugal filters (Millipore, 30 K MWCO). Purified proteins were then diluted in a buffer containing 20 mM Tris-HCl (pH 7.5), 1 mM DTT, and 10% PEG-8000. Protein solution (5 μl) was loaded onto a glass slide, covered with a coverslip, and mounted to an A1R confocal microscope (Nikon) for imaging. Droplets sizes were determined using the Image J software.

### Fluorescence recovery after photobleaching (FRAP)
FRAP assay was performed with the built-in FRAP module using a Nikon Eclipse Ti-E microscope-based confocal imaging system. The NONO-TFE3 related droplets were photo-bleached using a 488-nm laser beam. Bleaching was focused on a circular region of interest (ROI) using 100% laser power and time-lapse images were immediately collected. Fluorescence intensity was measured using Image J. Background intensity was subtracted and the percent recovery rate was calculated as (I−Imin)/(I0−Imin) × 100%, where I, Imin and I0 represent the normalized intensity for each time point, minimum (0 s after bleaching), and initial (before bleaching) intensities, respectively.

### TurboID-mediated proximity biotin labeling
UOK109 cells stably expressing NONO-TFE3-TurboID-GFP or NLS-TurboID-GFP were incubated with freshly made biotin (100 μM) for 20 min. Cells without biotin treatment was used as control. Cells were then washed 6 times with phosphate-buffered saline (PBS) and lysed in

fresh RIPA lysis buffer (1 mM PMSF, 5 mM Trolox, 10 mM sodium azide and 10 mM sodium ascorbate, 2 ml for each 10-cm dish) by gentle pipetting on ice for 5 min and subjected to one −80 °C freeze-thaw cycle. Whole-cell lysates were centrifuged at 15,000 g for 10 min at 4 °C to remove the cellular debris. To pull down biotinylated proteins, streptavidin magnetic beads (500 μl for each sample) were washed 3 times with 1 ml of RIPA lysis buffer and cleared whole-cell lysates with 100 mg of proteins were incubated with streptavidin magnetic beads at 4 °C overnight. Then, beads were washed twice with RIPA lysis buffer, five times with 0.1 M $Na_2CO_3$ at 4 °C and resuspended in 100 μL 0.1 M $NaHCO_3$ for on-beads digestion and further analysis with the Taplin Biological Mass Spectrometry Facility at Harvard University.

## Mass spectometry

Beads were washed at least five times with 100 μL of 50 mM ammonium bicarbonate. Subsequently, 5 μL (200 ng/μL) of modified sequencing-grade trypsin (Promega, Madison, WI) was added, and samples were incubated overnight at 37 °C. Following digestion, samples were centrifuged or placed on a magnetic stand (for magnetic beads), and the supernatant was removed. Extracts were dried in a speed-vac (-1 h) and re-suspended in 50 μL of HPLC solvent A (2.5% acetonitrile, 0.1% formic acid) for desalting via STAGE tips. Final samples were reconstituted in 10 μL of solvent A. A nano-scale reverse-phase HPLC column was prepared by packing 2.6 μm C18 spherical silica beads into a fused silica capillary (100 μm inner diameter × ~30 cm length) with a flame-drawn tip. After column equilibration, samples were loaded via a Famos autosampler (LC Packings, San Francisco, CA). Peptides were eluted with a gradient of increasing solvent B (97.5% acetonitrile, 0.1% formic acid). Eluted peptides were ionized by electrospray and analyzed on a Velos Orbitrap Elite mass spectrometer (Thermo Fisher Scientific, Waltham, MA) operated in DIA mode with a spray voltage of 2.445 kV and an ion transfer tube temperature of 250 °C. Full MS scans (m/z 350–1650) were acquired with RF lens at 40%, AGC target set to custom, normalized AGC target at 300%, maximum injection time 20 ms, and resolution 30,000. MS/MS DIA scans were performed with stepped normalized HCD collision energies (25.5%, 27%, 30%), multiplex ions disabled, first mass 200 m/z, AGC target 3000%, and maximum injection time 55 ms. Data were acquired in profile mode for both MS and MS/MS scans. Peptide identification was performed using Sequest (Thermo Fisher Scientific) against a protein database containing reversed decoy sequences. Peptide-spectrum matches were filtered to achieve a 1–2% false discovery rate (FDR).

## Immunoblotting and chromatin-associated protein fractionation

For immunoblotting, cells were lysed on ice with a RIPA lysis buffer (50 mM Tris-HCl pH 7.4, 1% NP-40, 0.5% sodium deoxycholate, 0.1% SDS, 150 mM NaCl, 2 mM EDTA, and 50 Mm NaF) supplemented with protease inhibitors and phosphatase inhibitors cocktail (Gendeport, Barker, TX, USA). After centrifugation at 12,000 g for 5 min, supernatants were collected and subject to concentrations determination using the BCA protein assay kit (Thermo, Rockford, IL, USA) as instructed. After denaturing with 1X SDS loading buffer at 95 °C for 5 min, the same amount of each sample was loaded to SDS-PAGE, transferred to nitrocellulose membranes (Millipore, Billerica, MA, USA), and blocked with 5% BSA in Tris-buffered saline pH 7.6 containing 0.1% Tween-20 (TBS-T). The membranes were incubated with the corresponding primary antibodies overnight at 4 °C and then washed with TBS-T buffer 3 times. An anti-rabbit secondary antibody (1:5,000, sigma, Cat# 7074) or anti-mouse secondary antibody (1:3,000, Cell Signaling Technology, Cat# 7076) were applied for another 2-h incubation at room temperature. After 3 rounds of TBS-T washing, the antigen–antibody complexes were detected with West-Q Pico Dura ECL Solution (Gendeport, Barker, TX, USA) under the

ChemiDoc Imaging system (Bio-Rad). Detailed antibody information used in the study was listed in Supplementary Data 5.

For chromatin protein salt fractionation, HEK293T cells ($2 × 10^7$) were suspended in 1 mL buffer A (10 mM Tris·HCl pH 7.5, 10 mM NaCl, 3 mM MgCl2, 0.5% NP- 40, and 2.5 mM DTT), incubated on ice for 10 min to release the nuclei, and then centrifuged at 1,200 g for 10 min. The supernatants were discarded as cytosol fraction and the nuclear pellets were harvest and resuspended in 100 μl buffer A containing various NaCl concentration and rotated at 4 °C for 30 min. After centrifugation at 1,700 g at 4 °C for 10 min, the supernatants were collected and labeled as the soluble chromatin fraction, and the pellets were regarded as the insoluble chromatin-bound fraction. All fractions were denatured with 1% SDS loading buffer and subjected to immunoblotting as described above by using the indicated antibodies. Uncropped and unprocessed scans of gels and blots are provided in Source data.

## In vivo tumor xerograph assay and analysis

Animal studies were approved and performed in accordance with the protocol provided by the Institutional Animal Care Use Committee (IACUC) of the Institute of Biosciences and Technology, Texas A&M University. Mice were kept under light-dark cycles of 12 h:12 h in a climate-controlled environment.UOK109 cells that stably expressing the indicated constructs were mixed with Matrigel (Corning, #354234) at a 1:1 ratio (100 μl of $5 × 10^6$ cells) and then injected subcutaneously (s.c.) into the flank of SCID-beige mice (8–12 weeks old, female, TACONIC CBSCBG). Tumor volume was determined with the formula: length × width² × 0.52. The second day after tumor cell injection, rapamycin (dissolved in corn oil containing 5% (v/v) DMSO, 2 mg/kg) were subjected into mice via intraperitoneal (i.p.) injection every other day.

## Luciferase-based bioluminescence assay

NONO-TFE3 variants, including the full length (FL), ΔPLD, ΔCC, ΔRRM, 4 A, and TFE3-S321A (constitutively localized in the nucleus) were fused with the Gal4 DNA-binding domain (Gal4DBD) in the pcDNA3.1 vector. For the reporter plasmid, a 5x upstream activation sequence (UAS) was inserted upstream of the E1b minimal promoter in the vector psiCHECK-2. For each assay, 100 ng of psiCHECK-2, 200 ng of each pcDNA3.1-GAL4 construct, and 200 ng of pTriEX-mCh-PSPC1 were co-transfected into HEK293T cells in a 12-well plate. To exclude the dosage effect, we co-expressed pTriEX-mCh without PSPC1 (-PSPC1). For the negative control, pcDNA3.1-GAL4-GFP-NLS was co-transfected to serve as a spatial control, as the NONO-TFE3 variants and TFE3-S321A were both located within the nucleusw. The luciferase-catalyzed bioluminescence intensity was measured at 36 h post transfection using the Dual-Luciferase Reporter assay (Promega), with *Renilla* luciferase activity normalized to firefly luciferase activity.

## LacO array imaging assay

Human U2OS cells carrying a LacO array with ~50,000 LacO elements in the genome were a gift from Dr. David L. Spector (Cold Spring Harbor Laboratory). Cells were seeded onto a 4-well glass bottom plate at 70% confluency and transfected with the indicated plasmids. Imaging was performed 24 h post-transfection. Fluorescence images were acquired on a W1 Yokogawa Ti2 Nikon spinning disk confocal microscope with the pinhole size setting up to 1.00 AU. Protein-protein interactions were quantified from the two-color images as previously reported[46]. In brief, the z stack slice (#N) in the GFP channel that exhibited the highest fluorescence intensity associated with a LacO-associated punctum was identified first. Then the central pixel of the LacO array was located and radial profiles of the fluorescence intensity centered on this pixel in both the GFP and mCherry channels were obtained. Next, we extracted the radial profile of GFP intensity and determined the radius of the punctum by identifying the point at

which the derivative of fluorescence intensity to distance first dropped to zero from the central pixel. Subsequently, we measured the maximum and peripheral mCherry intensities of the punctum. To mitigate intensity noise at the single-pixel level, we applied convolution to image slice #N of the mCherry channel using a $5 \times 5$ convolution kernel J5 (an all-ones matrix). We then averaged the intensities of four convoluted pixels surrounding the peak intensity pixel to obtain Ipeak. and two values on the mCherry intensity radial profile immediately outside the punctum periphery to obtain Iperiphery. Finally, we calculated the intensity ratio (Ipeak/Iperiphery) as an indicator of the mCherry enrichment at the LacO array. A ratio exceeding 1 indicates potential protein-protein interactions.

## CRISPR screening

UOK109 and 786-O cell lines stably expressing Cas9 were transduced with lentiviruses encoding a customized sgRNA library targeting RBPs and epigenetic regulators (kindly provided by Dr. Eric Wang, The Jackson Laboratory) at a low MOI (-0.3). The MOI was determined by GFP-positive (sgRNA-positive) cells using FACS on day 4 after transduction (T0). A total of $1 \times 10^7$ cells (with -300-500X sgRNA coverage) was maintained through passaging until day 44 (T44) for UOK109 or day 30 (T30) for 786-O cells. Genomic DNA (gDNA) was isolated using the QIAquick gel extraction kit (QIAGEN, 28104) according to the manufacturer's protocol. For library construction, forward primer (5' TCTTGTGGAAAGGACGAAACACCG 3') and reverse primer (5' TCTACTATTCTTTCCCCTGCACTGT 3') flanking the gRNA sequence region was used for PCR. 200 ng of gDNA was amplified for 20 cycles using Phusion Master Mix and then purified by using the SPRI beads (Beckman Colter, B23318). The purified products were used for sequence library construction (NEBNext Ultra II DNA Library Prep Kit for Illumina, E7645S). For pooled CRISPR screen analysis, individual time points for all samples were normalized using the formula (sgRNA read count/total read count) × 100,000. Subsequently, the normalized reads were used to calculate the log2fold change (normalized read count T0 normalized read count T44 or T30).

## In vitro kinase assay

The kinase complex containing CDK7-CycH-MAT1 was purchased from Millipore. 200 ng purified recombinant RNAPII-CTD, NONO-TFE3 and PSPC1 proteins were incubated with 100 ng kinase complex in a 30 μl reaction buffer (50 mM HEPES pH 7.3, 50 mM NaCl, 1 mM DTT, 10 mM $MgCl_2$, and 0.1 mM ATP) at 30 °C for 30 min. The kinase reactions were stopped by adding 10 μl of the 4X SDS-PAGE sample-loading buffer. After heating at 95 °C for 10 min, the samples were subjected to SDS-PAGE and immunoblotting with the indicated antibodies.

## In vitro droplet sedimentation assay

In vitro purified protein samples were mixed as described in droplet formation and then centrifuged for 10 min at 14,000 g, 4 °C. The same fractions of supernatant and pellet were used for immunoblotting analysis.

## In vitro transcription assay

Template bubble DNA was generated by annealing the forward primer (AGGCAGGCCTTAGCTCCGTTCGCCGTGTCCTACCTATCCTCTC CTCACCACTCCCGGGGCCATTC) with the reverse primer (TGGC CCCGGGAGTGGTGAGGAGAGGATAGGTAATCAGTTACGCCCGGAGC TAAGGCCTGCCTAGT), followed by ligation to the upstream of the first 281 bp of the GFP sequence. To initiate transcription, template bubble DNA, RNAPII CTD, NONO-TFE3, and PSPC1 were mixed as indicated in an in vitro transcription reaction buffer (25 mM HEPES, pH 7.5, 50 mM KCl, 65 mM NaCl, 10% glycerol, 5 mM $MgCl_2$, 1 mM DTT, and 0.05 mg/ml BSA). A nucleotide mix containing 0.375 mM ATP, CTP, UTP, and GTP (Invitrogen), along with 0.01 U RNase inhibitor (Invitrogen) and 1.25% PEG-8000, was then added. The reactions

were incubated at 30 °C for 2 h and stopped by adding a STOP buffer (0.3 M NaAc, 5 mM EDTA, 0.1% SDS, and 1 mg/ml Proteinase K). The reaction products were then subjected to RNA isolation using the RNeasy kit (Qiagen) with a spike-in RNA control. To remove template DNA, purified RNAs were treated with DNase (Invitrogen) at 37 °C for 30 min. Reverse transcription was carried out using the PrimeScript First-Strand cDNA Synthesis Kit (Takara), followed by qPCR with Universal SYBR Green Fast qPCR Mix (ABclonal) to quantify transcriptional output. Ct values were normalized to the spike-in RNA control.

## Nascent RNA labeling and imaging

UOK109 cells transfected with NONO-TFE3-GFP were seeded onto coverslips and allowed to adhere overnight. The following day, cells were incubated with 1 mM EU for 1 h to label nascent RNA. Subsequent chemical modification was performed using the Click-iT RNA Alexa Fluor 594 Imaging Kit (ThermoFisher, # C10330) according to the manufacturer's protocol. Fluorescence imaging was taken using a Nikon Eclipse Ti-E microscope.

## RNA transcriptional feedback reporter assay

Vectors used in the reporter assay were modified as previously described[21] using the pTETRIS-cargo vector (#126033, Addgene). Briefly, 6X STOP codon sequence was inserted into the pTETRIS-cargo vector to generate pTETRIS-cargo-STOP. RNA sequences with different lengths were cloned downstream of the 6X STOP sequence to prevent translation of these feedback RNAs. The feedback gene and the reporter gene have their own polyA termination signals facing to each other to terminate transcription. The reporter gene is regulated by TFE3 response elements (GTCACGTGAC, 6 repeats) and the feedback gene is regulated by a tetracycline-controlled promoter.

For luciferase assays, $1 \times 10^5$ cells were plated for transfections and allowed to settle overnight. Cells were then treated with doxycycline (Sigma) with the indicated concentrations and harvested after 24 h to measure either luciferase activity or to purify RNA. Luciferase activity was measured using the Luciferase Assay System (Promega) according to manufacturer instructions. Luciferase signal was normalized to total protein content, measured by BCA protein assay kit (Invitrogen, #23227), and then normalized to a control not treated with doxycycline. To measure RNA expression, RNA was purified using the QIAGEN RNeasy Mini kit (QIAGEN) according to manufacturer instructions, cDNA was generated by PrimeScript First-Strand cDNA Synthesis Kit (Takara), followed by qPCR with Universal SYBR Green Fast qPCR Mix (ABclonal). Ct values were normalized to a housekeeping gene (actin) and a control condition with no doxycycline treatment.

## Data analyses

Data analyses were conducted using Prism 10 software (GraphPad Software, San Diego, CA). Statistical evaluations were carried out using either one-way ANOVA, two-tailed unpaired t-tests or two-sided Wilcoxon test, as appropriate. Quantitative results are presented as mean ± SD unless otherwise noted.

## Reporting summary

Further information on research design is available in the Nature Portfolio Reporting Summary linked to this article.

## Data availability

The sequencing datasets have been deposited into the NCBI BioProject with the accession number PRJNA1129016. The mass spectrometry proteomics data generated in this study have been deposited in the ProteomeXchange database with accession code PXD056977. Source data are provided with this paper.

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

## Acknowledgements

We are grateful for the epigenetics core and flow cytometry core at the Institute of Biosciences and Technology at Texas A&M University. This work was supported by grants from the National Institutes of Health (R21CA277257, R35HL166557, R01DK132286, and R01CA240258 to YH; R01GM144986 to Y.Z.), the Leukemia & Lymphoma Society (LLS-TRP 6680-24 to Y.Z.), the Welch Foundation (BE-1913-20220331 to Y.Z.), the American Cancer Society (RSG-18-043-01-LIB to Y.H.), and the Alkek Early Career Fellowship to LG.

## Author contributions

Y.H., Y.Z. and L.G. directed and oversaw the project. L.G. and Y.L. performed most biochemical, cell based, animal-related work, high-throughput sequencing library construction and imaging analysis. R.Z. performed the majority of bioinformatics analysis with help from L.R. and J.W. J.H., T.H., T.W., K.R., Y.W., A.S., X.C., and R.W. provided essential resources and key intellectual inputs to support this study. All the authors participated in the discussion, data interpretation, and manuscript preparation or discussion.

## Competing interests

The authors declare no competing interests.
