## [Transparent Peer Review file · Nature Communications]

RNA-mediated condensation of TFE3 oncofusions facilitates transcriptional hub formation to promote translocation renal cell carcinoma (tRCC)

Corresponding Author: Dr Yun Huang

Version 0:

Reviewer comments:

Reviewer #1

(Remarks to the Author)

Guo et al., presented a study on elucidating the mechanism by which TFE3 fusion proteins form the condensates to facilitate transcription and promote tRCC formation. This study has been very well designed complemented with convincing data throughout the manuscript. In addition, this study also provides some proof-of-principle strategies on how to target oncofusion condensates through nanobody-based approaches. Overall, this is an excellent study. A couple of comments below to further strengthen this paper.

1. For Fig.4, it was unclear why 786-O parental cells were used for screening. One would argue that another TFE3 fusion protein containing tRCC cell lines or 786-O expressing TFE3 fusion protein may serve better. Also, multiple shRNAs would need to be used to further solidify the conclusion. In addition, why was SFPQ required for the survival of 786-O, which lacks TFE3 fusion. This needs to be clarified.
2. For Fig.5, is there a gene list or specific pathway that NONO-TFE3 controls that may regulate tRCC tumorigenesis and progression? This needs to be discussed.

Reviewer #2

(Remarks to the Author)

In this paper, Zhou et al. revealed that RNA-binding capability of TFE3 fusion partners drives the formation of TFE3 condensates. And TFE3 oncofusions could co-condensate with RNAPII and PSPC1 to form the transcriptional hubs and support gene transcription, promoting the tRCC malignancy. Besides, Zhou et al. designed a chemically inducible split nanobody and observed a good anti-tumor effect both in vitro and in vivo. This paper demonstrated the mechanism of TFE3 fusion protein in the tRCC malignant and provided the treatment strategy in this field. However, I have several concerns about this manuscript.

- 1、 In this paper, the author mainly used UOK109 (NONO-TFE3) cell lines to perform experiments, thus it is not appropriate to reach the conclusion, as tRCC has many fusion types.
- 2、 On Page 4, for the dTAG-13 treatment, how long the protein of NONO-TFE3 would be depleted , 24h? It could also be compared with the shNONO or shTFE3.
- 3、 On page 4, RIP-seq data also revealed the enrichment of similar gene sets (Figure S1G, Table S1). Should this figure legend be Figure S1H?
- 4、 On page 5 , the reported TFE3-S321A was introduced in UOK109 and conducted the CUT&Tag experiments, does the endogenous NONO-TFE3 interfere the results?
Does the TFE3-S321A transfect the other cells without endogenous NONO-TFE3 such as dTAG-treated UOK109KI, or 786-O?

- 5、 On page 5, confirming comparable expression levels of TFE3-S321A and NONO-TFE3, what does this mean? Were the protein expression levels of TFE3 comparable?
- 6、 On page 6 , the author compared the SLAM-seq between the DMSO (solvent control) and 4-h dTAG-treatment groups. Did the author analyze the shorter dTAG-treatment groups (2h) , which may provide more sensitive genes?
- 7、 On page 6, for Figure 2J, are the enrichments of NONO-TFE3 regulated genes in different fusion of tRCC patients different, or mainly upregulated in the tRCC patients with NONO-TFE3 fusion?
- 8、 On Page 6, similar condensation patterns were also observed with at least five additional tRCC-related TFE3 fusion proteins. Is the condensation pattern observed in the ASPSCR1-TEF3? As the ASPSCR1-TFE3 was the most common and had a poor prognosis, this fusion is better to be validated.
- 9、 On Page 7, the author explored the domain of NONO partner that support condensation, The previous study (Nat Commun. 2021 Sep 6;12(1):5262) has reported two types of NONO-TFE3 fusion, the NONO part retains exons 1-9 or exons 1-11, whether the condensation of NONO-TFE3 type (retains exons 1-11) is similar to the type (retains exons 1-9)?
- 10、 On page 7, deletion of RNA binding domains in other fusion partners, e.g., U2AF2, KHSRP, MATR3, RBM10 and SFPQ. Similarly, whether the ASPSCR1-TFE3 was analyzed?
- 11、 On page 9, for the candidates identified from both TurboID and CRISPR screens, please provide detailed information.
- 12、 In page 9, did the author observe the combined effect of dTAG-13+shPSPC1 in the vivo and vitro?
- 13、 On page 10, why selected HK2 and RRM2 as examples?
- 14、 On page 12, does author observe the distribution or toxic effect (such as liver and kidney function) of nanobody in vivo? And compared with the antitumor effect dTAG?
- 15、 The figure legends could be clarified more clearly, such as the representative images in Figure 3 and S3.
- 16、 The tRCC was renamed as TFE3-rearranged RCC in 2022 WHO Renal Cell Carcinoma classification, it should be modified,

Reviewer #3

(Remarks to the Author)

Guo et al. explored the importance of NONO in the TFE3 oncofusions for tRCC malignancy and its interaction with PSPC1 and Pol II via several approaches like sequencing, cell imaging and animal studies. They reported a possible mechanism that is TFE3 oncofusions form condensates with RNAPII and other RNA-binding proteins to promote tumor cell proliferation. Furthermore, they propose a potential therapy method for tRCC based on the above mechanism to inhibit tumor growth by dissolving oncofusion condensates, which produces positive results at the cellular level and in mouse experiments.

While numerous studies have demonstrated that fusion oncoproteins can form condensates to recruit RNA Polymerase II and regulate gene transcription in vivo and in vitro, this manuscript uniquely highlights the roles of RNA and the RNA-binding protein PSPC1 in facilitating oncoprotein condensate formation. However, previous work has shown that PSPC1 broadly contributes to genome-wide transcription activation in ESCs (Shao, W., Bi, X., Pan, Y. et al. Phase separation of RNA-binding protein promotes polymerase binding and transcription. *Nat. Chem. Biol.* 18, 70–80 (2022)). The authors should clarify what is distinct in PSPC1's function or mode of action within this context compared to existing studies. Regarding the role of RNA, the evidence underscores the importance of the RRM or RNA-binding ability of NONO-TFE3, yet does not specify how RNA contributes to transcription activation or which types of RNA are involved. Overall, the detailed molecular mechanism under the condensate of TFE3 oncofusions is not fully clear, and especially the contribution of RNA. To demonstrate these points, it needs more in vitro biochemical experiments. The manuscript does not meet the publication standards of Nature Communications.

Major points:

1. The CUT&RUN results indicated that the DNA motif specificity of NONO-TFE3 fusions remained largely consistent with that of TFE3 alone. However, as shown in Fig. 1H, a substantial proportion of genes bound in the CUT&Tag analysis did not overlap with those identified in RIP-seq, and the number of bound genes decreased significantly compared to Fig. 1J. This discrepancy warrants further explanation. Additionally, the authors should review the RNA-seq results for genes associated with NONO-TFE3 to verify if genes in the overlap between CUT&Tag and RIP-seq exhibit more pronounced upregulation than others, strengthening the overall conclusion. To enhance the rigor of these findings, RIP-seq should also be performed for TFE3 S321A.
2. While the overlap between CUT&Tag and RIP-seq data suggests colocalization of NONO-TFE3 with RNA, this does not establish a direct causal link between NONO-TFE3's RNA-binding ability and its genomic binding pattern in tRCC cells. To

further investigate this relationship, the authors should analyze the CUT&Tag and RIP-seq profiles of RNA-binding-deficient NONO-TFE3 mutants.

The abstract states that “our study establishes the causal role of RNA and RNA-binding proteins in facilitating oncofusion condensate formation to promote renal cancer progression.” However, the demonstration of causality here is debatable. Typically, causality can be investigated through *in vitro* assays; however, the *in vitro* assays in this manuscript, such as those in Fig. 6C and D, are insufficient for this purpose. Firstly, RNA is not present in these assays, making it difficult to examine RNA’s role directly. Secondly, while CTD recruitment may imply that these condensates are associated with gene transcription, this alone falls short of establishing causality.

3. In Page 6, the statement that “liquid-like condensate formation is driven by TFE3 fusion partners” is imprecise. While TFE3 fusion proteins form liquid droplets and TFE3 alone does not, this does not conclusively attribute condensate formation solely to fusion partners. Although NONO alone can form droplets, it remains possible that interactions between NONO and TFE3 contribute to condensate formation. Furthermore, while other TFE3 fusions also form condensates in cells, it is unclear whether these fusion partners can form droplets independently. This warrants further clarification and analysis.

4. The authors suggest that PSPC1 is critical for transcriptional regulation by NONO-TFE3, primarily based on their observed colocalization. However, this reasoning lacks specificity. A more detailed examination is needed to elucidate the molecular mechanisms underlying interactions among PSPC1, NONO-TFE3, and RNA. *In vitro* biochemical assays are important to conduct. Clarifying these interactions could provide a deeper understanding of how PSPC1 contributes to transcriptional regulation in the context of NONO-TFE3.

In this manuscript, the authors repeatedly emphasize the RNA-binding capacity of NONO-TFE3 but do not clarify the types of RNA present in the promoter regions. Additionally, as newly transcribed RNAs are known to feedback-regulate transcriptional initiation condensates, it is important to investigate this potential regulatory feedback mechanism.

5. The authors claimed that RNA-binding capacity is necessary for condensate formation. Although some experiments indirectly suggest that RNA is important, such as the 4A mutation and RNase fused chimera, the current manuscript lacks direct experimental evidence that RNA provides multivalence and participates in condensate formation. In fact, in Fig. 4i and Fig. 6 C-D, condensates are formed by NONO-TFE3 and Pol II CTD (or with PSPC1) even in the absence of RNA, indicating that protein-protein interactions are sufficient to provide multivalence. This leaves many questions unresolved: Is there really RNA in the condensates inside the cells? If there is RNA, is it necessary for condensation? Do the 4A mutation and RNase fused chimera disrupt the condensation by affecting protein binding to RNA or simply by affecting protein conformation like the CCD domain? The authors may need to provide further clarification on this detailed mechanism.

6. Figure 5 shows a significant reduction in transcription at NONO-TFE3 target genes, with no comparable effect observed at genes not regulated by NONO-TFE3 following PSPC1 depletion. Previous studies have indicated that PSPC1 plays a role in genome-wide transcription activation in ESCs (Shao, W., Bi, X., Pan, Y. et al. Phase separation of RNA-binding protein promotes polymerase binding and transcription. *Nat. Chem. Biol.* 18, 70–80 (2022)). Interestingly, PSPC1 appears to specifically interact with NONO-TFE3, selectively activating transcription of fusion target genes in tRCC. Further investigation is warranted to clarify how PSPC1 function differs in cells containing TFE3 fusions compared to other cell types.

7. The authors proposed that TFE3 oncofusion condensate can promote the formation of transcription hubs and enhance prooncogenic transcription. It is reasonable because Zuo, L. et al. 2019 (10.1038/s41467-021-21690-7) has also reported the FET fusion proteins can form condensate, recruit Pol II CTD and promote transcription. However, to provide direct evidence to verify the downstream process of TFE3 oncofusion condensate is correct, the authors can provide data to show whether disrupting the condensate would decrease prooncogenic transcription under the condition that both NONO-TFE3 and PSPC1 are endogenously expressed (for example, by disrupting condensation using the method shown in Fig. 7).

Minor points:

1. Page6, line 8, there should be “SLAM-seq” rather than “SALM-seq”.
2. The author needs to make the name of the model fusion proteins clearer in the manuscript, like Gal4DBD-NONO-TFE3 rather than NONO-TFE3.
3. Fig.4H, 4I, Fig.6D, the scale bars are needed in the panels.
4. Do all TFE3 oncofusions with RNA binding partner have similar structures? The author may add a schematic diagram to show the structures in the Supplementary Fig. 1.
5. In the analysis of Fig. 2, it may be better to add a control group of TFE3 with no fusion like that in Fig. 1. This addition can further demonstrate the importance of NONO-TFE3-RNA interactions on gene expression.
6. In Fig. S3C, the author provided a characterization parameter called “granularity”, but it is not mentioned in the main text or methods.

7. In the analysis of droplet assay in Fig. 6D, it may be better to use the participant ratio of pS5-CTD than the droplet area to show if the recruitment of pS5-CTD is promoted when PSPC1 was added.

8. To verify the effect of RNA on the condensation and Pol II recruitment, the authors need to add RNA with different lengths (or sequences) to the current droplet experiments.

Reviewer #4

(Remarks to the Author)

General comments

Oncofusion proteins may contribute to the formation of liquid-like condensates due to the structural and functional properties of the fusion itself. These condensates may play a role in cancer, for example by altering chromatin biology or influencing cellular signaling. In this study, Dr. Guo et al. investigate NONO-TFE3, the product of a chromosomal translocation placing the transcription factor TFE3 under the regulation of NONO, a nuclear RNA-binding protein. This chromosomal translocation is responsible for translocation-RCC (t-RCC), an aggressive type of kidney cancer. Previous research has shown that the intrinsically disordered coiled-coil domain in NONO facilitates the formation of NONO-TFE3 condensates (PMID: 33592266). Here, the authors use various omics approaches and a dTAG-based inducible system to control NONO-TFE3 levels in UOK109 cells derived from a patient carrying NONO-TFE3 fusion. They demonstrate that the RNA-binding ability provided by NONO is essential for condensate formation, which in turn promotes tRCC growth both in vitro and in vivo. Using a dual approach that combines TurboID-proximity proteomics and CRISPR screening, they identify PSPC1, a paraspeckle-associated RNA-binding protein, within NONO-TFE3 condensates. They further show that PSPC1 is essential for NONO-TFE3-driven tumor cell growth (in UOK109) but not for VHL-associated RCC (in 786-O cells). The NONO-TFE3 and PSPC1 co-condensates promote the recruitment of RNA polymerase II (RNAPII) to chromatin. Notably, by applying a chemogenetic tool, they show that dissolving these condensates significantly inhibits tumor growth. Overall, this study is well-designed and several data support the notion that NONO-dependent oncofusion condensates play a role in tRCC. However, the underlying mechanisms remain unclear. If the authors' claim is that increasing the association of RNAPII with chromatin enhances the transcriptional activity of TFE3 on its target genes, this must be demonstrated through gene expression analysis. Importantly, TFE3, along with other members of the MiT/TFE family (notably TFEB and MITF), are oncogenic even outside of oncofusion contexts, with their cancer-promoting roles strongly dependent on their transcriptional activity (PMID: 16001072, PMID: 26168401, PMID: 28619945, PMID: 36987696, PMID: 34779410, PMID: 38195686). TFE3 oncofusions may drive cancer progression by increasing TFE3's nuclear localization (which is otherwise regulated by mTORC1 in response to nutrient availability) and possibly by enhancing TFE3 stability. In this study, the authors compare the genomic binding sites of NONO-TFE3 with TFE3-S321A (a constitutively nuclear form of TFE3 that cannot be phosphorylated by mTORC1) and show that both share 85% of genomic binding sites in UOK109 cells. However, they observe that UOK109 cell growth under constitutively active TFE3 expression is similar to that seen in TFE3-silenced cells (Fig. 3I, Fig. S3E). These findings are surprising and suggest potential technical issues, such as inefficient expression of the TFE3-S321A construct. To fully explore the role of the NONO component, it would be ideal to generate UOK109 cells lacking the NONO segment but endogenously expressing TFE3-S321A and then compare transcriptional profiles and oncogenic activity in these cells relative to the parental line expressing NONO-TFE3. In addition I have some concerns related to the selected list of "primary targets" (see specific comments).

Specific comments

- Fig. 1J, K: How are UOK109 cells overexpressing TFE3-S321A or NONO-TFE3 generated? Is this achieved through knockout followed by stable transfection? Western blot data should be included to confirm comparable expression levels of the two constructs, and RNA-seq data should be provided for further comparison.
- Fig. 2F-J: The authors define "primary transcripts" as those that decrease after 4 hours of TFE3 degradation. It seems more likely that the first transcripts to decrease are simply the less stable ones. The true "primary targets" would more accurately be those that increase rapidly upon washout, though this point may be less critical given the chronic model of TFE3 activity being studied. Additionally, the comparisons in Fig. 2F and H are not entirely accurate. Cut & Tag or RIP-seq analysis following 4 hours of TFE3 degradation reflects changes due to reduced NONO-TFE3 binding to DNA/RNA, whereas SLAM-seq after 4 hours highlights downregulated transcripts based on stability. This approach does not fully align with the data being gathered. Genes bound by NONO-TFE3 but not yet downregulated may, in fact, be the ones most influenced by the RNA-binding capacity of the oncofusion. A more consistent comparison would involve DMSO versus dTAG at 24 hours across all datasets, focusing on transcripts significantly deregulated and enriched for NONO-TFE3 binding (as per Cut & Tag or traditional ChIP-seq). In addition, if the authors claim that NONO enhances RNAPII recruitment to TFE3-bound DNA, they should support this by comparing transcript levels in UOK109 KI cells expressing TFE3-S321A (without NONO).
- Fig. 2I: The model is not entirely clear. The authors propose that NONO-TFE3 influences transcriptional and post-transcriptional regulation of the same targets, resulting in enhanced expression of oncogenic targets, but only for "primary targets" identified after 4 hours of degradation. This claim lacks supporting evidence, and, as noted above, the selection of "primary transcripts" may be somewhat arbitrary.
- Fig. 2J: This comparison considers only a subset of selected "primary" genes. A good correlation should also be expected with the larger set of target genes downregulated after 24 hours. However, this correlation is expected, as it involves comparing genes upregulated in tRCC-cells relative to those upregulated in tRCC patient tumors.
- Fig. 3A: Was an additional NLS added to the mutant TFE3? This appears to be an error; please clarify this point.
- Fig. S3D: The levels of TFE3-CA are lower relative to other constructs, which could indicate technical issues affecting assessment of TFE3-S321A's oncogenic potential.

• Fig. S3E - Fig. 3I: It seems implausible that cells overexpressing TFE3-CA would mimic TFE3-silenced cells in terms of proliferation and tumor growth. Nuclear localization and transcriptional activity (RNA-seq) should be compared between NONO-TFE3 and TFE3-S321A, as well as between mutant constructs for the NONO fragment. To conclusively show that the NONO segment drives tumor growth, the authors should consider generating UOK109 KI cells lacking the NONO segment but mutated at Ser321.

Minor comments:

- Review figure legends to ensure they clearly describe dTAG-13 treatment details.
- Correct the term "ex vivo" throughout the paper, as it is used inappropriately to describe in vitro experiments.
- On page 4: Update the figure reference from Fig. S1G to the correct Fig. S1H where reported: "RIP-seq data also revealed the enrichment of similar gene sets"

Version 1:

Reviewer comments:

Reviewer #1

(Remarks to the Author)

The revised manuscript addressed all of previous critiques and now it is acceptable for publication.

Reviewer #2

(Remarks to the Author)

The authors provided answers to all my comments , and I have no further comments.

Reviewer #3

(Remarks to the Author)

The authors have addressed all my comments in great detail and have significantly improved their manuscript. I recommend the manuscript for publication.

Reviewer #4

(Remarks to the Author)

In this revised version of the manuscript, the authors have added data suggesting that nascent RNA is present within NONO-TFE3 condensates and provided functional assays to support a role for adjacent RNA in modulating the transcriptional activity of the fusion protein. However, the response to point 4.7, concerning the lack of oncogenic activity of the constitutively active TFE3-S321A mutant, remains inadequate.

First, the RNA-seq analysis (new Fig. S4M) includes only two biological replicates per group, which is not statistically sufficient. Even disregarding this major limitation, the data presented in the provided dataset show that well-established TFE3 target genes such as GPNMB, ATP6V1H, RRAGD, and TPP1 are not upregulated upon reintroduction of the TFE3-S321A construct. This strongly suggests technical issues with the construct itself, which likely explains why its re-expression fails to rescue cell growth and instead results in a phenotype comparable to TFE3 loss-of-function. This significantly weakens the conclusions drawn from this experiment.

A
t the end of their rebuttal, the authors also cite published data to argue that non-fusion forms of MiT/TFE transcription factors are insufficient to drive tumorigenesis. However, this interpretation is overly simplistic and partially incorrect. For instance, kidney cysts are commonly observed in tRCC patients (PMID: 34489456, PMID: 32699116), and are also present in additional murine models, not just the one cited (PMID: 27668431), but also in PRCC-TFE3 mice (PMID: 31043488). The relative abundance of cystic versus neoplastic lesions in mouse models likely depends on the timing and cell type specificity of TFEB/TFE3 induction. Furthermore, loss of FLCN, a tumor suppressor gene which leads to constitutive nuclear localization of TFEB and TFE3, results in robust tumorigenesis that is completely abolished by inactivation of either factor (PMID: 36987696). These studies clearly demonstrate that constitutive activation of MiT/TFE factors is sufficient to drive renal tumorigenesis, even in the absence of oncogenic fusions.

Therefore, this part of the study is not sufficiently robust and undermines the overall conclusions of the manuscript. It cannot be accepted in its current form.

Version 2:

Reviewer comments:

Reviewer #4

(Remarks to the Author)

The authors have addressed all my concerns, and the statement they added in the discussion section aligns well with my views. I have no further requests.

Point-to-Point Responses

We would like to express our gratitude to the editor and the four reviewers for their valuable insights and constructive feedback, which have significantly improved our manuscript. We have considered all the points raised and have conducted all the recommended experiments in this substantially revised manuscript. To ensure comprehensive responsiveness, we have extensively revised the manuscript to address concerns raised by all four reviewers. Modifications made in the main text and supplementary materials were highlighted using blue font for easy identification. For your convenience, we have provided a summarized overview of the new and revised figures in **Table A1** below.

Table A1. Major changes made throughout the revised manuscript.

Figure / panel No.	Original Figure No.	Response to Reviewer(s)	Remarks
1J,1K	1J, 1K	R2.4, R3.1, R4.1	Comparison of NONO-TFE3 and TFE3-S321 binding sites across the genome
S1A, S3E	New	R1.1, R2.1, R2.7	Unique transcriptomes of tRCC
S1G, S5F, 4J	New	R1.1	2 independent shRNAs for validation
S1C	New	R3.11	Domain structures of TFE3 oncofusions
2I, S2F, S3D	New	R3.4, R4.3	The regulatory feedback mechanism
S2G	New	R3.1	Expression of RIP-seq and CUT&Tag overlapped and non-overlapped genes
S4C bottom	S3C	R3.3	Fusion partners can form condensates
S4E-G	New	R3.5, R4.3	RNA binding to support condensate formation
S4H-M	New	R3.1, R3.2, R4.6	Causal link between RNA-binding and genomic binding of NONO-TFE3
6C-D, S6E	6C-D	R3.4, R3.14, R4.3	Biochemical assays to demonstrate the impact of RNA and PSPC1 on oncofusion condensates
S7A-B	New	R3.14	The impact of RNA length and sequence on condensate formation
2I, 7A-D, 7G	New	R3.4, R4.3	The impact of adjacent RNA on condensate formation and transcription
8G-I, S8H	New	R2.14, R3.7	Anti-tumor effects of disrupting condensates

Reviewer #1

We sincerely thank the reviewer for the supportive remarks that “*This study has been very well designed complemented with convincing data throughout the manuscript. In addition, this study also provides some proof-of-principle strategies on how to target oncofusion condensates through nanobody-based approaches. Overall, this is an excellent study.*”

1.1. For Fig.4, it was unclear why 786-O parental cells were used for screening. One would argue that another TFE3 fusion protein containing tRCC cell lines or 786-O expressing TFE3 fusion protein may serve better. Also, multiple shRNAs would need to be used to further solidify the conclusion. In addition, why was SFPQ required for the survival of 786-O, which lacks TFE3 fusion. This needs to be clarified.

Response: We sincerely appreciate the reviewer’s suggestions. Detailed explanations are listed below:

For Fig.4, it was unclear why 786-O parental cells were used for screening. One would argue that another TFE3 fusion protein containing tRCC cell lines or 786-O expressing TFE3 fusion protein may serve better.

Response: Below are the reasons we chose 786-O parental cells for CRISPR screening:

(1) Including a kidney cancer cell line that lacks TFE3 onco-fusions as a control would enable the identification of downstream targets specific to TFE3 onco-fusions (**Figure 4F**). These specific targets could

Figure 4F

Figure S4C

Figure S1A

Figure 4B

Figure 4F. Venn diagram showing the candidates number identified from the CRISPR screening with UOK109 and 786-O cell lines.

Figure S1A. Heatmap of genes overexpressed in tRCC with different TFE3 oncofusions (33 cases) as compared with other RCC subtypes or normal kidney (1251 cases) across all datasets.

Figure S4C. Representative images (left) and quantification (right) of droplet formation in HEK 293T cells transfected with the indicated TFE3 oncofusions in the presence or absence of RNA binding domains. Granularity, referring to the fluorescence signal fluctuation, is calculated as the standard deviation (S.D.) of the relative signal intensity along a line plot (normalized to the mean signal intensity of the entire line) at each pixel through an individual cell. (n = 8 cells from 3 independent biological replicates; one-way ANOVA with Tukey’s post-hoc test; *** p < 0.0001). Scale bar, 10 μm. Data are shown as mean ± SD.

Figure 4B. Volcano plot showing proteins preferentially interacting with NONO-TFE3 over NONO-TFE3-4A (logFC > 1 or < -1 and p < 0.05).

guide the development of personalized therapies capable of precise tumor targeting with minimal toxicity, providing new therapeutic options for tRCC patients who do not respond to current treatments. We avoided the use of 786-O cells expressing the TFE3 fusion protein because overexpression systems tend to introduce numerous artifacts, leading to false-positive or false-negative candidates and compromising the reliability of the screening.

(2) Our analysis using published RNA-seq data (PMID: 34986355, 34489456) showed distinct transcriptional signatures of tRCC compared with other renal cancer cases and normal kidney tissues. We reanalyzed published RNA-seq data collected from 33 tRCC cases and 1251 non-tRCC cases. Our analysis reveals that tRCC cases with TFE3 oncofusions exhibit remarkably similar transcriptomes regardless of fusion partners, which are distinctly different from those of non-tRCC cases (**NEW Figure S1A**). This data is aligned with the data collected in our manuscript that TFE3 oncofusion proteins exhibited similar biochemical features, such as RNA binding and liquid-like condensation capabilities (**Figure S4C**). Furthermore, our proteomic data showed that most fusion partners are part of a large interacting complex, suggesting a shared regulatory mechanism among these TFE3 fusion proteins (**Figure 4B**). These data indicate TFE3 oncofusion proteins might exhibit similar transcriptional regulatory mechanisms to support oncogenic transcription and mediate the pathogenesis of tRCC. Therefore, we use UOK109 cells as a representative tRCC cell line to identify the key genes essential for tRCC survival.

We completely agree with the reviewer that incorporating additional tRCC cell lines containing other TFE3 fusion proteins would be ideal to identify authentic and general regulators that are essential for tRCC cell survival. However, this seems to be beyond the scope of the current study. If additional resources are available, we will obtain additional patient-derived tRCC cell lines to further extend our study in the future.

Also, multiple shRNAs would need to be used to further solidify the conclusion.

Response: As requested, we used 2 independent shRNAs against NONO-TFE3, SFPQ-TFE3 and PSPC1 to exclude the off-target effects and obtained consistent results. New data are included in the manuscript as **NEW Figure S1G** and **NEW Figure 4J**.

Figure S1G

Figure 4J

Figure S1G. Representative images (left) and quantification (right) of the colony formation assay results using UOK145 cells transfected with the scrambled shRNA or 2 independent shRNAs against SFPQ. n = 3 independent biological replicates; one-way ANOVA with Tukey's post-hoc test, *** p<0.0001. Data are shown as mean ± SD.

Figure 4J. Representative images (left) and quantification (right) of clone formation assay results for UOK109 or 786-O cells expressing the indicated shRNAs. (n = 3 independent biological replicates; one-way ANOVA with Tukey's post-hoc test; ***p<0.0001). Data are shown as mean ± SD.

In addition, why was SFPQ required for the survival of 786-O, which lacks TFE3 fusion. This needs to be clarified.

Response: The requirement of SFPQ for the survival of 786-O cells, which lack TFE3 fusion, could be attributed to the following reasons:

(1) SFPQ knockout mice are embryonically lethal at an early stage (PMID: 29719248), in contrast to its closely related family members, NONO and PSPC1, whose knockout are tolerated during development, as these mice are born normal and show no apparent defects (PMID: 29358041, 28192372 and 29483655). This suggests that SFPQ may be involved in fundamental biological processes, such as RNA splicing, polyadenylation and stabilization (PMID: 27084935), which are essential for the survival of all cell types and cannot be compensated for by other proteins.

(2) SFPQ plays essential roles, such as in DNA damage repair (PMID: 27084935), across all types of kidney cancer; thus, its depletion leads to cell death in both cell lines.

We have added the following in the revised “Discussion” section: “*Notably, SFPQ, a close family member of NONO and PSPC1, was identified as essential for both TFE3 fusion and non-fusion kidney cancer. This dependency could be attributed to its fundamental roles in RNA splicing, polyadenylation and stabilization, and DNA damage repair (PMID: 27084935), which are essential for the survival of all cell types and cannot be compensated for by other proteins.*”

1.2. For Fig.5, is there a gene list or specific pathway that NONO-TFE3 controls that may regulate tRCC tumorigenesis and progression? This needs to be discussed.

Response: We have performed integrative analysis combining SLAM-seq, RIP-seq and CUT&Tag data to identify specific genes and pathways that NONO-TFE3 controls. As shown in **Fig 2E** and **Fig S3H**, gene ontology (GO) analysis indicated that NONO-TFE3 regulates genes involved in metabolic process, DNA replication, cell cycle, and DNA damage response, all of which can contribute to tumorigenesis and progression of tRCC. We also include a gene list containing all NONO-TFE3 regulated genes in **Table S2**.

Figure 2E

Figure S3H

Figure 2E. The Gene Ontology (GO) analysis of DEGs identified from the 4-h dTAG13 treatment group.

Figure S3H. The Gene Ontology (GO) analysis of differential transcriptional genes identified from 12- or 24-h dTAG-13 treatment groups.

Reviewer #2

In this paper, Zhou et al. revealed that RNA-binding capability of TFE3 fusion partners drives the formation of TFE3 condensates. And TFE3 oncofusions could co-condensate with RNAPII and PSPC1 to form the transcriptional hubs and support gene transcription, promoting the tRCC malignancy. Besides, Zhou et al. designed a chemically inducible split nanobody and observed a good anti-tumor effect both in vitro and in vivo. This paper demonstrated the mechanism of TFE3 fusion protein in the tRCC malignant and provided the treatment strategy in this field. However, I have several concerns about this manuscript.

2.1. In this paper, the author mainly used UOK109 (NONO-TFE3) cell lines to perform experiments, thus it is not appropriate to reach the conclusion, as tRCC has many fusion types.

Response: We thank the reviewer for raising this concern. Indeed, we acknowledge the limitations of our study, primarily due to the use of only two tRCC TFE3-rearranged cell lines with TFE3 fusion proteins (UOK109 with NONO-TFE3 and UOK145 with SFPQ-TFE3). However, we did evaluate the RNA-binding-

Figure S4C

Figure 4B

Figure S8F

Figure S8G

Figure S1A

Figure S3E

Figure S4C. Representative images (left) and quantification (right) of droplet formation in HEK 293T cells transfected with the indicated TFE3 oncofusions in the presence or absence of RNA binding domains. Granularity, referring to the fluorescence signal fluctuation, is calculated as the standard deviation (S.D.) of the relative signal intensity along a line plot (normalized to the mean signal intensity of the entire line) at each pixel through an individual cell. (n = 8 cells from 3 independent biological replicates; one-way ANOVA with Tukey's post-hoc test; *** p<0.0001). Scale bar, 10 μ m. Data are shown as mean \pm SD.

Figure S1A. Heatmap of genes overexpressed in tRCC with different TFE3 oncofusions (33 cases) as compared with other RCC subtypes or normal kidney (1251 cases) across all datasets.

Figure S8F. Representative images (left) and quantification (right) of droplet formation in UOK145 GFP cells (endogenous SFPQ-TFE3 replaced with GFP labeled SFPQ-TFE3 by knocking down SFPQ-TFE3 first and then re-expressing GFP-SFPQ-TFE3) transfected with the indicated constructs with or without rapamycin (Rapa). (n = 3 independent biological replicates; one-way ANOVA with Tukey's post-hoc test; *** p<0.0001). Scale bar, 10 μ m. Data are shown as mean \pm SD.

Figure S8G. Representative images (left) and quantification (right) of clone formation assay results using UOK145 GFP cells as described in (F) with or without rapamycin (Rapa) treatment. (n = 3 independent biological replicates; one-way ANOVA with Tukey's post-hoc test; *** p<0.0001). Data are shown as mean \pm SD.

Figure 4B. Volcano plot showing proteins preferentially interacting with NONO-TFE3 over NONO-TFE3-4A (logFC > 1 or < -1 and p < 0.05).

Figure S3E. Gene set enrichment analysis of NONO-TFE3 target genes identified from SLAM-seq in UOK109 KI cells and genes up-regulated (logFC > 3 and FDR < 0.05) in tRCC patients with various fusion partners (left) or NONO alone (right).

dependent condensation features of some other TFE3 oncofusion proteins in HEK293T cells by expressing cDNAs encoding these TFE3 oncofusion proteins (**Figure S4C**). While this feature may not apply to all TFE3 oncofusion proteins, the RNA-binding-dependent fusion properties might be relevant to some fusion cases, as evaluated in **Figure S8F-G**. Nevertheless, we would like to highlight that tRCC tumors exhibit distinct transcriptional features compared to non-tRCC cases (**NEW Figure S1A**). Furthermore, regardless of the fusion partners of TFE3, tRCC cases show similar transcriptional profiles (**NEW Figure S1A**). We analyzed publicly available RNA-seq data from patient samples, comprising 33 tRCC cases and 1,251 non-tRCC cases (PMID: 34986355, 34489456). As described above, tRCC cases with different TFE3 oncofusions exhibit remarkably similar transcriptomes, which are different from those of non-tRCC cases (**NEW Figure S1A**). Additionally, the target genes we identified using UOK109 are highly enriched in patients harboring different TFE3 or TFEB fusion partners (**NEW Figure S3E**). Moreover, our proteomic analysis revealed that most fusion partners are part of a large interacting complex with TFE3 oncofusion protein, further suggesting a shared transcriptional regulatory mechanism among these TFE3 fusion proteins to support the pathogenesis of tRCCs (**Figure 4B**).

We completely agree with the reviewer that it is not appropriate to draw a general conclusion based on a limited number of cell lines with TFE3 fusions. However, we hope the reviewer appreciates our efforts to identify general molecular features or potential therapeutic targets shared among most tRCC cases, with the goal of developing effective treatment for this devastating disease. Further validation experiments in tRCCs with different fusion events will be necessary to strengthen these discoveries in the future. To avoid overstating our findings, we have toned down our overall conclusion and also added the following statement to the Discussion section: *“It is important to note that, given the existence of multiple fusion partners for TFE3, the mechanism we identified with NONO-TFE3 may represent only some of the key characteristics of these oncofusions. We cannot rule out the possibility of other unique mechanisms specific to different fusion types, which warrant further investigation.”*

2.2. On Page 4, for the dTAG-13 treatment, how long the protein of NONO-TFE3 would be depleted, 24h? It could also be compared with the shNONO or shTFE3.

Response: As shown in **Figure 1B**, we observed a rapid reduction in the NONO-TFE3 protein level at 4 hours after dTAG-13 treatment (40-50% reduction). An almost-to-total depletion was achieved when cells were treated for 24 hours. As suggested by Reviewer 2, we also include shRNAs against NONO as a comparison, which induced ~70% protein reduction (**Figure B1 for reviewer only**).

2.3. On page 4, RIP-seq data also revealed the enrichment of similar gene sets (Figure S1G, Table S1). Should this figure legend be Figure S1H?

Response: We thank the review for pointing out this typo. This typo has been corrected in our updated manuscript.

2.4. On page 5, the reported TFE3-S321A was introduced in UOK109 and conducted the CUT&Tag experiments, does the endogenous NONO-TFE3 interfere the results? Does the

Figure 1B. Representative immunoblotting showing time-dependent degradation of the endogenous NONO-TFE3 with dTAG13 (500 nM) at indicated time points.
Figure for reviewer only B1. Immunoblotting showing endogenous NONO-TFE3 level using dTAG-13 or 2 independent shRNA against NONO.

Figure 1J and K. Venn diagram (J) and heatmap binding profiles of NONO-TFE3 or TFE3-S321A CUT&Tag signals (K) in dTAG-13 treated UOK109 KI cells transfected with Flag tagged NLS-TFE3 S321A. n = 2 independent biological replicates.

TFE3-S321A transfect the other cells without endogenous NONO-TFE3 such as dTAG-treated UOK109KI, or 786-O?

Response: We thank the reviewer for raising this concern. As the reviewer mentioned, we cannot rule out the possibility that endogenous NONO-TFE3 might interfere with the binding of overexpressed TFE3-S321A. Additionally, overexpression systems can introduce artifacts, potentially leading to the identification of false binding targets. To address these concerns, we followed the reviewer's advice by transfecting dTAG-treated UOK109 KI cells with TFE3-S321A and re-performing the CUT&Tag assay. The new results confirmed our original conclusion that NONO-TFE3 exhibits enhanced genomic binding at the original TFE3 binding sites (**NEW Figure 1J-K**).

2.5. On page 5, confirming comparable expression levels of TFE3-S321A and NONO-TFE3, what does this mean? Were the protein expression levels of TFE3 comparable?

Response: We apologize for the confusion. To compare the CUT&Tag signal intensities of TFE3-S321A and NONO-TFE3, we overexpressed these proteins in UOK109 cells. To ensure a robust comparison, we carefully matched the expression levels of the two proteins within the cells. However, as raised in Question 2.4, we echo reviewer's concern regarding potential interference from endogenous NONO-TFE3. We then expressed TFE3-S321A in dTAG-treated UOK109 KI cells to avoid the inference from the endogenous NONO-TFE3 (**NEW Figure 1J-K**). UOK109 KI cells were treated with dTAG for 24 hours to eliminate endogenous NONO-TFE3. Subsequently, TFE3-S321A was expressed at levels comparable to endogenous NONO-TFE3. CUT&Tag assays were then conducted using these two cell lines, ensuring comparable expressions of TFE3-S321A and NONO-TFE3 for the following experiments.

2.6. On page 6, the author compared the SLAM-seq between the DMSO (solvent control) and 4-h dTAG-treatment groups. Did the author analyze the shorter dTAG-treatment groups (2h), which may provide more sensitive genes?

Figure for reviewer only B2. Principal component analysis (PCA) of SLAM-seq results at the indicated timepoints (left) and Venn diagram showing the overlapping genes identified from SLAM-seq (dTAG-13 treatment for 2 and 4 hours) (right).

Response: Since the protein levels of NONO-TFE3 remain at a similar level between 2 hrs and 4 hrs dTAG treatment (**Figure 1B**), we do not expect major difference between these 2 time-points. Nevertheless, to address the reviewer's concern, we performed the SLAM-seq in cells treated with dTAG for 2 hrs. As we expected, the transcriptomes of the 2-hour and 4-hour dTAG treatment groups exhibit high similarity, with over 96% of genes overlapping between the two conditions. Related data is included in **Figure B2 for**

reviewer only.

2.7. On page 6, for Figure 2J, are the enrichments of NONO-TFE3 regulated genes in different fusion of tRCC patients different, or mainly upregulated in the tRCC patients with NONO-TFE3 fusion?

Response: Figure 2J showed the enrichment of NONO-TFE3-regulated genes (identified in UOK109 cells) across different fusion types in tRCC patients, including NONO-TFE3 (3 cases), SFPQ-TFE3 (3 cases), ASPSCR1-TFE3 (8 cases), PRCC-TFE3 (6 cases), RBM10-TFE3 (1 case), MALTA1-TFEB (3 cases), CTCL1-TFEB (1 case), SFPQ-TFEB (1 case), ACTG1-MITF (1 case). As explained in response to Question 2.1, tRCC patient samples exhibited similar transcriptional signatures regardless of TFE3 fusion partners

(**Figure S1A**), suggesting shared transcriptional regulatory mechanisms in tRCC. We further carried out an enrichment analysis by comparing the signature genes identified in UOK109 cell line (with NONO-TFE3) with the transcriptomic data in patients with NONO-TFE3 fusions and observed a significant enrichment between these 2 datasets (**NEW Figure S3E**), suggesting that UOK109 cell is a reliable pre-clinical model to recapitulate the molecular features of tRCCs.

Figure S1A

Figure S1A. Heatmap of genes overexpressed in tRCC with different TFE3 oncofusions (33 cases) as compared with other RCC subtypes or normal kidney (1251 cases) across all datasets.

Figure S3E. Gene set enrichment analysis of NONO-TFE3 target genes identified from SLAM-seq in UOK109 KI cells and genes up-regulated ($\log_{2}FC > 3$ and $FDR < 0.05$) in tRCC patients with various fusion partners (left) or NONO alone (right).

Figure S3E

2.8. On Page 6, similar condensation patterns were also observed with at least five additional tRCC-related TFE3 fusion proteins. Is the condensation pattern observed in the ASPSCR1-TEF3? As the ASPSCR1-TEF3 was the most common and had a poor prognosis, this fusion is better to be validated.

Figure for reviewer only B3. Representative image of GFP fused ASPSCR1 overexpressed in 786-O cells. Scale bar, 10 μ m.

Response: As the reviewer suggested, we carefully examined the condensation-forming capability of the ASPSCR1-TEF3 fusion protein. In our experimental settings, we did not observe clear droplet formation when expressing the full-length ASPSCR1-TEF3 fusion protein in 786-O cells (**Figure B3 for reviewer only**). However, this observation does not entirely rule out the possibility that ASPSCR1-TEF3 may exhibit condensation properties under different conditions. There are several potential reasons for the undetected ASPSCR1-TEF3 condensates: (1) The size of condensates can vary significantly between different proteins, and the resolution required to detect ASPSCR1-TEF3 droplets may exceed the capabilities of our microscope. (2) The 786-O cells used in our experiments may not provide a suitable cellular environment for ASPSCR1-TEF3 to form condensates.

Some proteins require specific interacting chaperone proteins to facilitate condensate formation, and 786-O cells might lack the necessary chaperones. Further studies by introducing GFP at the endogenous ASPSCR1-TEF3 in patient derived tRCC cell lines would be necessary to evaluate the condensation features of ASPSCR1-TEF3 in the future.

Figure for reviewer only B4. Representative imaging of GFP labelled NONO-TFE3 retaining exon 1-9 or 1-11 of NONO (left) and quantifications of droplet numbers each cell (right). $n = 8$ independent replicates. Two-sided unpaired Student's t-test; *** $p < 0.0001$). Scale bar, 10 μ m. Data are shown as mean \pm SD.

2.9. On Page 7, the author explored the domain of NONO partner that supports condensation. The previous study (Nat Commun. 2021 Sep 6;12(1):5262) has reported two types of NONO-TFE3 fusion, the NONO part retains exons 1-9 or exons 1-11, whether the condensation of NONO-TFE3 type (retains exons 1-11) is similar to the type (retains exons 1-9)?

Response: We have investigated the condensation property of another NONO-TFE3 fusion type (retaining exons 1-11 of NONO) as pointed out by the reviewer. As shown in **Figure B4 for reviewers only**, NONO-TFE3 fusion with exon 1-11 exhibited a similar condensation property to the model we used, as both contain the intact condensate-driving RRM and CC domains.

2.10. On page 7, deletion of RNA binding domains in other fusion partners, e.g., U2AF2, KHSRP, MATR3, RBM10 and SFPQ. Similarly, whether the ASPSCR1-TFE3 was analyzed?

Response: We did not observe well-defined RNA binding domains (such as RRM and KH domains) in the fusion part of ASPSCR1. However, we cannot entirely rule out the RNA binding capability of ASPSCR1 as certain amino acid sequences can function as RNA binding motif and mediate the protein-RNA interaction (PMID: 37402367). We hope the reviewer agrees that characterizing the RNA-binding motif of ASPSCR1-TFE3 is beyond the scope of the current study.

2.11. On page 9, for the candidates identified from both TurboID and CRISPR screens, please provide detailed information.

Response: As reviewer requested, we provided a full list of candidates identified from TurboID and CRISPR screens in **Table S3** and **Table S4**.

2.12. In page 9, did the author observe the combined effect of dTAG-13+shPSPC1 in the vivo and vitro?

Response: As reviewer suggested, we performed in vitro clone formation assay to examine if any synergistic effect could be observed in the dTAG-13+shPSPC1 group. We did not find dramatic synergy in this combined treatment as single treatment is strong enough to inhibit the growth of UOK109 cells (**Figure B5 for reviewer only**)

Figure for reviewer only B5. Representative images (left) and quantification (right) of the colony formation assay results using UOK109 KI cells treated with conditions as indicated. n= 3 independent replicates. Two-sided unpaired Student's t-test; *** p<0.0001). Data are shown as mean \pm SD.

2.13. On page 10, why selected HK2 and RRM2 as examples?

Response: The reasons we selected HK2 and RRM2 as examples are as follows:

(1) These two genes are ranked as top target genes of NONO-TFE3 based on following observations: CUT&Tag data showed strong enrichment of NONO-TFE3, PSPC1 and RNAPII binding at the promoters of these two genes; the expression of these two genes are significantly reduced in NONO-TFE3 or PSPC1 knockout conditions; the RIP-seq and SLAM-seq results showed that the nascent transcripts derived from HK2 and RRM2 are directly associated with NONO-TFE3.

(2) Previous studies supported oncogenic function of HK2 and RRM2. Hexokinase 2 (HK2), a pivotal enzyme in glucose metabolism, is often overexpressed in numerous cancers and is closely associated with tumor initiation and progression. It plays a crucial role in regulating cell cycle progression, promoting lactate production, and facilitating autophagy (PMID: 23911236, 28915575). Ribonucleotide reductase regulatory subunit M2 (RRM2) plays a critical role in cancer development and progression due to its involvement in DNA synthesis and repair. As a key component of ribonucleotide reductase (RNR), RRM2 regulates the conversion of ribonucleotides into deoxyribonucleotides, providing the building blocks required for DNA replication and repair. In cancers, RRM2 is often overexpressed, and this dysregulation contributes to

several tumor-promoting processes, including proliferation, genome instability, therapeutic resistance and cancer stemness (PMID: 33411689, 37588202).

(3) Targeting HK2 and RRM2 using small molecule inhibitors showed potential anti-tumor effects. For RRM2, inhibitors such as COH29, triapine and other small molecules have been developed to block its activity, impairing DNA synthesis and inducing cancer cell death. These inhibitors are being explored in preclinical and clinical settings for their potential to enhance the efficacy of chemotherapies and overcome resistance in various cancers (PMID: 33227712). For HK2, inhibitors like 3-Bromopyruvate (3-BP), 2-deoxyglucose (2-DG) and other targeted molecules disrupt glycolysis and energy production in cancer cells, leading to reduced tumor growth. Some HK2 inhibitors are undergoing clinical trials to assess their therapeutic potential, particularly in combination with other treatments (PMID: 35609742).

Since TFE3 oncofusion proteins play an essential role in transcriptional regulation and might not be optimal therapeutic targets, the selection of HK2 and RRM2 provides examples of alternative therapeutic targets for tRCC treatment. The availability of small molecule inhibitors targeting HK2 and RRM2 offers high translational potential for treatment.

2.14. On page 12, does author observe the distribution or toxic effect (such as liver and kidney function) of nanobody in vivo? And compared with the antitumor effect dTAG?

Response: In our nanobody-based in vivo testing system, we used rapamycin to induce the FKBP-cpFRB heterodimerization and brought the split LaG parts into proximity to form an intact functional LaG nanobody. Rapamycin is an FDA-approved drug that is applicable for the treatment of multiple diseases, with an acceptable toxicity profile. To further prove the tolerance of this treatment, we monitored the liver and

Figure for reviewer only B6. Serum concentrations of Alanine Aminotransferase (ALT) and Creatinine in mice treated with DMSO or Rapamycin (Rapa). n = 5 mice. Two-sided unpaired Student's t-test; *** p<0.0001. Data are shown as mean ± SD.

Figure 8D. Representative images (left) and quantification (right) of clone formation assay results for UOK109 KI cells transduced with the indicated constructs with or without rapamycin (Rapa) treatment. (n = 3 independent biological replicates; one-way ANOVA with Tukey's post-hoc test; *** p<0.0001). Data are shown as mean ± SD.

Figure 8G and H. Principal component analysis (G) and heatmap (H) of RNA-seq results with UOK109 KI cells treated with dTAG-13 or rapamycin (Rapa). n = 2 independent biological replicates.

Figure 8I. Venn diagram showing overlapping differentially expressed genes (DEGs) identified from RNA-seq in UOK109 KI cells treated with DMSO, dTAG-13 or Rapamycin (Rapa) for 4 hours.

Figure S8H. Pearson correlation analysis of RNA-seq data from UOK109 KI cells treated with indicated conditions. n = 2 independent biological replicates.

kidney function by measuring ALT and Creatinine concentration in blood respectively. We did not observe dramatic differences between the control and drug-treated groups. The related data are shown in **Figure B6 for reviewer only**. Although dTAG-13 has demonstrated high efficacy for leukemia cells *in vivo* (PMID: 29581585), its performance in solid tumors has been less effective. At least in our lab, we were unable to achieve efficient NONO-TFE3 degradation in xenografts formed by UOK109 cells through intraperitoneal (i.p.) injection in mice. This limitation is likely attributed to the poor tissue penetration of dTAG-13, which may hinder its ability to reach target proteins effectively in solid tumor environments. This is why we used shRNA to knock down NONO-TFE3 in our *in vivo* experiments (**Figure 3I**). For the *in vitro* clone formation assay, dTAG-13-induced degradation of NONO-TFE3 exhibits a similar inhibitory effect on tumor growth as rapamycin-induced disruption of condensates via nanobody (**Figure 8D**). Additionally, our newly performed RNA-seq analysis revealed that dTAG-13-induced degradation led to gene expression changes comparable to those observed with nanobody-induced condensate disruption (**NEW Figure 8G-I, NEW Figure S8H**).

Collectively, these data reinforce our findings that the condensation property of the onco-fusions is crucial for their transcriptional activity.

2.15. The figure legends could be clarified more clearly, such as the representative images in Figure 3 and S3.

Response: We apologize for the confusion in our figure legends. Due to the word limitations imposed by publication requirements, we initially simplified the figure legends to comply with the criteria. However, as suggested by the reviewer, we have added further explanations or include details in Method section and made every effort to ensure the captions are clear and informative for readers.

2.16. The tRCC was renamed as TFE3-rearranged RCC in 2022 WHO Renal Cell Carcinoma classification, it should be modified.

Response: We thank the reviewer for pointing out this issue. Based on 2022 WHO Renal Cell Carcinoma classification (PMID: 38765391), the TFE3-rearranged and TFEB-altered RCC categories were separated, and the MIT category no longer exists. In addition, TFEB-altered RCC is again subdivided into TFEB-rearranged and TFEB-amplified RCC. In our manuscript, tRCC refers to “translocation renal cell carcinoma”, which includes rearrangements of both TFEB and TFE3.

Reviewer #3

Guo et al. explored the importance of NONO in the TFE3 oncofusions for tRCC malignancy and its interaction with PSPC1 and Pol II via several approaches like sequencing, cell imaging and animal studies. They reported a possible mechanism that is TFE3 oncofusions form condensates with RNAPII and other RNA-binding proteins to promote tumor cell proliferation. Furthermore, they propose a potential therapy method for tRCC based on the above mechanism to inhibit tumor growth by dissolving oncofusion condensates, which produces positive results at the cellular level and in mouse experiments.

While numerous studies have demonstrated that fusion oncoproteins can form condensates to recruit RNA Polymerase II and regulate gene transcription *in vivo* and *in vitro*, this manuscript uniquely highlights the roles of RNA and the RNA-binding protein PSPC1 in facilitating oncoprotein condensate formation. However, previous work has shown that PSPC1 broadly contributes to genome-wide transcription activation in ESCs (Shao, W., Bi, X., Pan, Y. et al. Phase separation of RNA-binding protein promotes polymerase binding and transcription. *Nat. Chem. Biol.* 18, 70–80 (2022)). The authors should clarify what is distinct in PSPC1's function or mode of action within this context compared to existing studies. Regarding the role of RNA, the evidence underscores the importance of the RRM or RNA-binding ability of NONO-TFE3 yet does not specify how RNA contributes to transcription activation, or which types of RNA are involved. Overall, the detailed molecular mechanism under the condensate of TFE3 oncofusions is not fully clear, and especially the contribution of RNA. To demonstrate these points, it needs more *in vitro* biochemical experiments. The manuscript does not meet the publication standards of Nature Communications.

Response: We sincerely appreciate the reviewer's suggestions. Additional *in vitro* experiments have been performed to address the concerns, which are explained in detail within every single question.

Major points:

3.1. The CUT&RUN results indicated that the DNA motif specificity of NONO-TFE3 fusions remained largely consistent with that of TFE3 alone. However, as shown in Fig. 1H, a substantial proportion of genes bound in the CUT&Tag analysis did not overlap with those identified in RIP-seq, and the number of bound genes decreased significantly compared to Fig. 1J. This discrepancy warrants further explanation. Additionally, the authors should review the RNA-seq results for genes associated with NONO-TFE3 to verify if genes in the overlap between CUT&Tag and RIP-seq exhibit more pronounced upregulation than others, strengthening the overall conclusion. To enhance the rigor of these findings, RIP-seq should also be performed for TFE3 S321A.

Response: We thank the reviewer for raising these questions. The detailed explanations are listed as follows:

The CUT&RUN results indicated that the DNA motif specificity of NONO-TFE3 fusions remained largely consistent with that of TFE3 alone. However, as shown in Fig. 1H, a substantial proportion of genes bound in the CUT&Tag analysis did not overlap with those identified in RIP-seq.

Response: CUT&Tag experiment is to identify the genomic association of NONO-TFE3, while RIP-seq is to identify RNAs that are directly or indirectly associated with NONO-TFE3. The rationale to compare CUT&Tag and RIP-seq data is because NONO-TFE3 contains both RNA (NONO) and DNA (TFE3) binding components.

DNA binding: Since NONO does not have its own DNA binding motifs within the fusion part, it is unlikely to have significant impact to alter genomic distribution of TFE3 when fused with TFE3 C-terminus containing

defined DNA binding domain. Indeed, based on our CUT&Tag data (**Figure 1K**), majority of NONO-TFE3 binding sites (~90%) are overlapped with TFE3-S321A genomic binding sites. The fusion partners, such as NONO, could enhance the transcriptional activity at certain TFE3 binding sites due to its condensation features.

RNA binding: On the other hand, NONO is an RNA binding protein. The RIP-seq data reflects the RNAs associated with NONO-TFE3, which might be due to NONO as the fusion partner. Since NONO is a generic RNA binding protein and is able to interact with various RNAs, we do not expect that NONO would exclusively bind to RNAs directly derived from NONO-TFE3 binding sites. However, based on our analysis, we observed that NONO-TFE3 with high RNA association (not necessarily derived from NONO-TFE3 binding sites) showed enhanced DNA binding, which suggested that NONO-mediated RNA binding supports the genomic association of NONO-TFE3, which aligned with the main focus in our manuscript.

Moreover, a "transcription factor binding site" is a specific DNA sequence where a transcription factor protein can bind, but it does not necessarily mean that transcription factor directly targets the gene containing that binding site for transcription. Many transcription factors can bind to multiple sites across the genome, some of which may not have a functional impact on gene expression, meaning not all genes with a binding site are considered "target genes" of the transcription factor (PMID: 21295369, 32424124, 26857150, 24888900). In our case, the primary targets identified from SLAM-seq are all within the overlapping genes of CUT&Tag and RIP-seq, indicating that both chromatin binding and RNA binding of NONO-TFE3 are critical for the expression of its target genes. Genes exhibiting only CUT&Tag or RIP-seq signals are less likely to be the primary target genes of NONO-TFE3, as they showed less gene expression changes after dTAG-13 treatment compared with the overlapped ones (**NEW Figure S2G**). The functional significance of these regions remains unclear. We speculate that they may contribute to chromatin structure maintenance; however, further investigations are needed to validate this hypothesis, which is beyond the scope of this study.

And the number of bound genes decreased significantly compared to Fig. 1J. This discrepancy warrants further explanation.

Response: The CUT&Tag performed in Figure 1H used the antibodies against endogenous Flag-NONO-TFE3 in UOK109 KI cells, whereas the CUT&Tag shown in original Figure 1J was performed in UOK109 WT cells overexpressing NONO-TFE3 or NLS-TFE3-S321A. These data were collected in different settings and could not be directly compared side-by-side. To overcome this limitation, we transfected dTAG-treated UOK109 KI cells with cDNA encoding Flag-TFE3-S321A under the control of a weak promoter (PGK) to

maintain its expression similar as endogenous NONO-TFE3 (**NEW Figure S4J**), followed by the CUT&Tag assay. We obtained ~5,294 genes showing binding signals in dTAG-treated UOK109 KI cells expressing Flag-TFE3-S321A, which aligned with the gene numbers (~5,011) identified from CUT&Tag using the antibody against endogenous Flag-NONO-TFE3 in UOK109 KI cells. Furthermore, we observed that TFE3-S321A binds to genomic regions similar to those targeted by NONO-TFE3, but

Figure S4J. Immunoblotting showing the protein expression levels of NONO-TFE3, TFE3-S321A and TFE3-C in dTAG-13 treated UOK109 KI stable cells.
Figure 1J and K. Venn diagram (J) and heatmap binding profiles of NONO-TFE3 or TFE3-S321A CUT&Tag signals (K) in dTAG-13 treated UOK109 KI cells transfected with Flag tagged NLS-TFE3 S321A. n = 2 independent biological replicates.

with much lower enrichment, consistent with findings made from the overexpression system. The updated data are included in **NEW Figure 1J-K**.

Figure S2G

Figure S2G. Quantification of expression level (left) and fold changes after dTAG-13 treatment (right) of the genes exhibiting overlapped CUT&Tag and RIP-seq signals or CUT&Tag signals alone (non-overlapped). The box plots indicated the median (center line), the third and first quartiles (box limits) and 1.5x interquartile range (IQR) above and below the box (whiskers). (n = 2 independent biological replicates; two-sided Wilcoxon test; $p < 0.05$). TPM, transcripts per kilobase million.

Additionally, the authors should review the RNA-seq results for genes associated with NONO-TFE3 to verify if genes in the overlap between CUT&Tag and RIP-seq exhibit more pronounced upregulation than others, strengthening the overall conclusion.

Response: We re-analyzed the RNA-seq results and confirmed that genes overlapping between CUT&Tag and RIP-seq (red bar) exhibit higher expression levels and undergo more pronounced changes in expression after dTAG-13 treatment, when compared to the non-overlapped group (green bar). The corresponding data are presented in the **NEW Figure S2G**.

To enhance the rigor of these findings, RIP-seq should also be performed for TFE3 S321A.

Response: We conducted RIP-seq for TFE3 S321A and found no significant RNA enrichment, as the IP group exhibited RNA binding intensities similar to those of the input group. The corresponding data are shown in **NEW Figure S4H**.

Figure S4H. Metagenome analysis of CUT&Tag (left, centered on called WT peaks from -5 to 5 kb adjacent to TSS) and normalized RIP-seq intensities (right) of NONO-TFE3 binding genes from -2 kb TSS to +2 kb transcriptional end site (TES) for NONO-TFE3 WT and 4A mutant. TFE3 and input were also included in RIP-seq analysis as control.

3.2. While the overlap between CUT&Tag and RIP-seq data suggests colocalization of NONO-TFE3 with RNA, this does not establish a direct causal link between NONO-TFE3's RNA-binding ability and its genomic binding pattern in tRCC cells. To further investigate this relationship, the authors should analyze the CUT&Tag and RIP-seq profiles of RNA-binding-deficient NONO-TFE3 mutants.

Response: To address this concern, we performed CUT&Tag and RIP-seq for the RNA-binding-deficient NONO-TFE3-4A mutant. As shown in the metagenome analysis, the 4A mutant exhibited a significantly reduced RNA-binding capacity (as reflected from RIP-seq data; right) and decreased genomic binding (as shown in CUT&Tag data; left). These findings indicate that the RNA binding feature plays a crucial role in mediating the genomic binding of NONO-TFE3. The corresponding data are included in **NEW Figure S4H**.

The abstract states that “our study establishes the causal role of RNA and RNA-binding proteins in facilitating oncofusion condensation to promote renal cancer progression.” However, the demonstration of causality here is debatable. Typically, causality can be investigated through in vitro assays; however, the in vitro assays in this manuscript, such as those in Fig. 6C and D, are insufficient for this purpose. Firstly, RNA is not present in these assays, making it difficult to examine RNA’s role directly. Secondly, while CTD recruitment may imply that these condensates are associated with gene transcription, this alone falls short of establishing causality.

Response: We thank the reviewer for raising this concern. To demonstrate the causality, we have performed additional *in vitro* assays (as detailed in our response to Question 3.4 below), which address the role PSPC1 and RNA in facilitating the partition of RNAPII CTD into NONO-TFE3 condensates and promoting efficient transcription.

3.3. In Page 6, the statement that “liquid-like condensate formation is driven by TFE3 fusion partners” is imprecise. While TFE3 fusion proteins form liquid droplets and TFE3 alone does not, this does not conclusively attribute condensate formation solely to fusion partners. Although NONO alone can form droplets, it remains possible that interactions between NONO and TFE3 contribute to condensate formation. Furthermore, while other TFE3 fusions also form condensates in cells, it is unclear whether these fusion partners can form droplets independently. This warrants further clarification and analysis.

Response: As pointed out by the reviewer, we cannot rule out the possibility that interactions between NONO and TFE3 may also play a role in condensate formation, rather than attributing it exclusively to the fusion partners. To be more precise, we have rephrased it to “*the fusion partners are one of the important components contributing to liquid-like condensate formation of TFE3 oncofusion events.*” Furthermore, we performed additional experiments to monitor the condensation features of other fusion partners by

Figure S4C. Representative images of full length TFE3 fusion partners fused with mCherry. Scale bar, 10 μ m.

expressing cDNAs encoding selected fusion partners (**NEW Figure S4C**) fused to mCherry in HEK293T cells. Our data supported the conclusion that these selected fusion partners could form droplets independently. The related data are included in **NEW Figure S4C**.

3.4. The authors suggest that PSPC1 is critical for transcriptional regulation by NONO-TFE3, primarily based on their observed colocalization. However, this reasoning lacks specificity. A more detailed examination is needed to elucidate the molecular mechanisms underlying interactions among PSPC1, NONO-TFE3, and RNA. *In vitro* biochemical assays are important to conduct. Clarifying these interactions could provide a deeper understanding of how PSPC1 contributes to transcriptional regulation in the context of NONO-TFE3.

Response: We have conducted the following *in vitro* biochemical assays to elucidate the molecular mechanisms underlying interactions among PSPC1, NONO-TFE3, and RNA:

(1) Co-condensation assay. We first examined the effects of PSPC1 and RNA on the co-condensation of NONO-TFE3 and RNAPII CTD. As shown in **NEW Figure S7A** and **NEW Figure 6C**, recombinant RNAPII CTD failed to form condensates on its own but could be incorporated into NONO-TFE3 condensates. PSPC1 could co-condense with NONO-TFE3 and RNAPII CTD to form larger condensates than those containing only NONO-TFE3 and RNAPII CTD. We then titrated different amounts of RNA (10, 100, 1000 ng/ μ L) into these condensates. A low amount of RNA (10 ng/ μ L) had a mild impact on the condensate size of NONO-TFE3 and RNAPII CTD, while it significantly increased the size of condensates containing PSPC1, NONO-TFE3, and RNAPII CTD. Increased RNA (100 ng/ μ L) could disrupt the condensates containing NONO-TFE3 and RNAPII CTD but had a minor impact on condensates containing PSPC1, NONO-TFE3, and RNAPII CTD. Further increased RNA (1000 ng/ μ L) would disrupt the condensate formation of PSPC1, NONO-TFE3, and RNAPII CTD (**NEW Figure 6C-D**). These data are consistent with previous findings that charge balance between RNA and protein is crucial for the behavior of transcriptional condensates (PMIDs: 28556382, 29569441, and 33333019). The presence of PSPC1 effectively prevents the excessive RNA-induced disruption of condensates containing RNAPII CTD and NONO-TFE3, suggesting that PSPC1 possibly stabilizes the condensates by neutralizing negatively charged RNA.

(2) *In vitro* sedimentation assay. We further validated our observations from the co-condensation assay through droplet sedimentation (**NEW Figure 6E**). The *in vitro* formed condensates were centrifuged, and the proteins in the condensates and in the solution were separated. This method helps evaluate the involvement of RNAPII within the condensates in the presence of PSPC1 and/or RNA. Consistent with the co-condensation assay, a greater amount of RNAPII CTD was retained within the condensates containing PSPC1/NONO-TFE3/RNAPII CTD, compared to the condensates only containing NONO-TFE3 and RNAPII CTD. Low concentrations of RNA (10 ng/μL) further enhanced RNAPII CTD incorporation efficiency.

(3) *In vitro* kinase assay. Phosphorylation of RNAPII CTD is required for its transcriptional activities. Our Gal4DBD-mediated luciferase data suggested that the fusion partner, i.e., NONO, promotes the transcriptional activity of TFE3, and PSPC1 could further enhance the transcriptional activity of NONO-TFE3. Therefore, we tested whether PSPC1 and RNA (10 ng/μL) would boost the phosphorylation of RNAPII CTD using an *in vitro* kinase assay. We mixed RNAPII CTD, NONO-TFE3, and recombinant CTD

Figure 6C. Representative images of *in vitro* droplets formed by mCh fused NONO-TFE3, BFP fused Pol2 CTD and GFP-fused PSC1 with indicated concentrations of RNA. GFP alone was used as control. Scale bar, 10 μm.

Figure S7A. *In vitro* droplet formation using recombinant BFP labelled RNAPII CTD proteins (CTD-BFP). Sale bar = 10 μm.

Figure 6D. Quantifications of droplet area (top) and co-efficiency of *in vitro* droplet formation performed in (C). (n = 30 droplets from 3 independent biological replicates; two-sided unpaired Student's t-test; *** p<0.0001). Data are shown as mean ± SD.

Figure 6E. Schematic illustration of the droplets sedimentation assay (top) and immunoblotting of CTD incorporation in droplets (bottom).

Figure 7A. *In vitro* kinase assay using recombinant proteins as indicated. Phosphorated CTD (p-CTD) and other components were determined by immunoblotting with corresponding antibodies.

Figure 7B. Schematic diagram of the reconstituted *in vitro* transcription assay.

Figure 7C. Images of droplets formed with in the *in vitro* transcription reaction. Hoechst was used to indicate the DNA component within the droplets. Scale bar, 10 μm.

Figure 7D. Quantifications of *in vitro* transcription reaction activity (normalized to RNAPII only) with indicated recombinant proteins. (n = 6 independent biological replicates; two-sided unpaired Student's t-test; *** p<0.001). Data are shown as mean ± SD.

kinases, cyclin-dependent kinase 7 (CDK7), together with ATP, in the presence or absence of PSPC1 and/or RNA. The western blot analysis revealed markedly enhanced CTD phosphorylation in the presence of PSPC1 or RNA compared with the conditions without. Furthermore, the addition of both PSPC1 and RNA led to the highest phosphorylation levels of RNAPII CTD compared with other groups (**NEW Figure 7A**). These results, along with findings from the co-condensation assay, strongly indicate that enhanced molecular interactions within phase-separated condensates enable more efficient biochemical reactions by locally concentrating key components.

(4) In vitro transcription assay. We checked the effects of PSPC1 and RNA in a fully defined *in vitro* transcription system with a DNA template containing a heteroduplex bubble that has been widely used as a nucleic acid scaffold (PMID: 34916619). RNAPII can bind to single-stranded DNA within the bubble without the help of general transcription factors. We then added NONO-TFE3 in the presence or absence of PSPC1 together with RNAPII CTD in this assay. The transcriptional activity was measured by qRT-PCR using primers against RNAs derived from the template (**NEW Figure 7B-D**). We observed that NONO-TFE3 enhanced transcriptional output, while PSPC1 + NONO-TFE3 showed more than 3-fold increase in transcriptional output compared with the control group. Furthermore, this enhanced transcriptional activity was not observed in the NONO-TFE3-4A group with or without PSPC1. These results further support our hypothesis that low concentrations of RNA produced in these systems work together with NONO-TFE3 and PSPC1 to support RNAPII-mediated transcription.

Taking these results together, NONO-TFE3 oncofusion enables RNAPII to form co-condensates. The presence of newly transcribed low amounts of RNA could further retain RNAPII within the co-condensates, possibly due to the electrostatic force derived from the high negative charge density of RNA. On the other hand, PSPC1 might not only mitigate the charge repulsion from excessive RNA, which may otherwise expel RNAPII CTD, but also utilize RNA to facilitate condensate formation for RNAPII engagement. This, in turn, efficiently compartmentalizes RNAPII CTD, enhancing its phosphorylation by CDK to promote transcription. Thus, the interplay between PSPC1 and RNA in promoting RNAPII condensate formation plays a crucial role in regulating the transcriptional activity of NONO-TFE3. The related data have been updated in the corresponding figures as mentioned above.

In this manuscript, the authors repeatedly emphasize the RNA-binding capacity of NONO-TFE3 but do not clarify the types of RNA present in the promoter regions. Additionally, as newly transcribed RNAs are known to feedback-regulate transcriptional initiation condensates, it is important to investigate this potential regulatory feedback mechanism.

Response: To clarify the types of RNAs that bind to NONO-TFE3 at promoter regions, we analyzed the RIP-seq data and found that protein-coding mRNAs are the predominant RNA species interacting with NONO-TFE3, followed by lncRNAs and miRNAs. Notably, we cannot rule out the presence of enhancer RNAs, as the RIP-seq conditions may not be optimal for detecting these short-lived and low-abundance RNA species. We have included this data in **NEW Figure S2F**.

We also followed the reviewer's great suggestion to investigate the potential regulatory feedback mechanism of RNA. Using a tailored reporter system designed to assess feedback regulatory dynamics (PMID: 33333019), we examined how artificially modulating feedback RNA levels influences NONO-TFE3 transcriptional activity. Briefly, we incorporated a TFE3 DNA-binding sequence (GTCACGTGAC, 6x) upstream of the coding region of luciferase. The transcriptional activity of NONO-TFE3 was reflected by measuring the luciferase expression. To modulate the RNA levels surrounding the locus, the DNA sequences encoding RNAs (200 nucleotides), together with doxycycline (Dox)-inducible promoters, were cloned into the adjacent coding regions of luciferase. This plasmid was co-transfected with NONO-TFE3

Figure S2F
NONO-TFE3 interacting RNAs

Figure S3D
Figure S2F. Category annotation of NONO-TFE3 interacting RNAs identified from RIP-seq. **Figure 2I.** Scheme depicting the reporter system (top) where local RNA expression near a luciferase reporter gene can be induced by Doxycycline (Dox). Transcriptional activity of NONO-TFE3 were monitored by luciferase intensity (normalized to 0 ng/mL Dox, bottom). n = 6 independent biological replicates, one-way ANOVA with Tukey's post-hoc test, *** p < 0.0001. Data were shown as mean ± SD.

Figure S3D. qRT-PCR analysis of relative RNA expression level (normalized to 0 ng/mL Dox) induced by indicated doses of Doxycycline (Dox). n = 6 independent biological replicates, one-way ANOVA with Tukey's post-hoc test, *** p < 0.0001. Data were shown as mean ± SD.

Figure 2I
into HEK293T cells, followed by treatment with escalating Dox doses (0, 10, 100, 1000 ng/mL) to induce different levels of RNA expression surrounding TFE3 binding sites. The gradually increased RNA expression in response to the escalation of Dox treatment was measured by qRT-PCR. The NONO-TFE3-mediated transcriptional activity was measured by luciferase activity via luminescence (**NEW Figure 2I, top**). Our data aligns with previous observations (PMID: 33333019) that increasing short feedback RNA levels initially enhances reporter expression but suppresses it at excessive concentrations (**NEW Figure 2I, bottom and NEW Figure S3D**). These results suggest that transcription operates as a non-equilibrium process with dynamic RNA-mediated feedback. In this model, low levels of short RNAs produced during transcription initiation enhance NONO-TFE3 target gene expression, whereas high levels of RNAs generated during elongation could exert inhibitory effects (PMID: 33333019).

3.5. The authors claimed that RNA-binding capacity is necessary for condensate formation. Although some experiments indirectly suggest that RNA is important, such as the 4A mutation and RNase fused chimera, the current manuscript lacks direct experimental evidence that RNA provides multivalence and participates in condensate formation. In fact, in Fig. 4i and Fig. 6 C-D, condensates are formed by NONO-TFE3 and Pol II CTD (or with PSPC1) even in the absence of RNA, indicating that protein-protein interactions are sufficient to provide multivalence. This leaves many questions unresolved: Is there really RNA in the condensates inside the cells? If there is RNA, is it necessary for condensation? Do the 4A mutation and RNase fused chimera disrupt the condensation by affecting protein binding to RNA or simply by affecting protein conformation like the CCD domain? The authors may need to provide further clarification on this detailed mechanism.

Response: To prove that RNAs are indeed involved in condensation formation, additional *in vitro* and *in vivo* experiments were performed:

(1) To demonstrate that nascent RNA is an integral component of NONO-TFE3 condensates, we employed an RNA labeling approach that permits the visualization of newly synthesized RNA within cells temporally and spatially (PMID: 18840688). This strategy involves the incorporation of 5-ethynyl uridine (EU), an alkyne-modified nucleoside, into nascent RNA, followed by “click chemistry” with an azide-conjugated fluorescent dye. As illustrated in **NEW Figure S4E**, we detected pronounced colocalization between nascent RNA and NONO-TFE3 condensates, highlighting the ability of NONO-TFE3 to associate with newly synthesized transcripts.

(2) To determine whether the loss of condensation in the 4A mutant and RNaseA-fused chimera is due to structural changes, we used AlphaFold to predict the protein structures of NONO-TFE3 WT, 4A mutant, and RNaseA-fused chimera. The predictions revealed no significant conformational differences among the three variants, hinting that the inability to form condensates arises from compromised RNA binding rather than structural changes (**Figure for reviewer C1**). The mutation of 4A within the two RRM domains of the DBHS family proteins (SFPQ, NONO, PSPC1) is a well-established approach to abolish RNA binding, as reported

in previous studies (PMID: 29358041, 29483655, 39358506). These mutations are commonly used to specifically investigate RNA-binding-dependent functions without altering the protein's overall structure or stability. Additionally, we introduced a catalytically inactive version of RNaseA (H12A, H119A) fused with NONO-TFE3 and observed that it retained the ability to form condensates. This finding further supports that the disruption of condensate formation is caused by the loss of RNA binding rather than alterations in protein conformation. The related data have been incorporated into **NEW Figure 4F-G**.

(3) As described in the response to Question 3.4, we conducted an in vitro droplet formation assay with and without RNA to demonstrate the contribution of RNA in forming multivalent condensate. Consistent with previous findings (PMID: 33333019), we observed a dose-dependent effect of RNA on NONO-TFE3/PSPC1/RNAPII condensate size: low RNA concentrations promoted condensate formation, while high RNA concentrations led to condensate dissolution. Our findings align with the concept of a non-equilibrium feedback control mechanism, where low RNA levels facilitate condensate formation through electrostatic interactions, while higher RNA levels drive condensate dissolution (PMID: 33333019, 34916619).

Figure S4E

Figure for reviewer only C1

Figure S4F

Figure S4G

Figure S4E. Immunofluorescence of NONO-TFE3 and 5-ethynyl uridine (EU) labelled RNA in UOK109 cells. Cells without EU adding in culture medium were used as control. Scale bar, 10 μ m.

Figure for reviewer only C1. Side views of the AlphaFold predicted structures of NONO-TFE3 WT, 4A mutant and RNase A-fused chimera (RNase A-NT).

Figure S4F. Representative images (left) and quantification (right) of the droplet numbers in UOK109 cells expressing NONO-TFE3 fused with wildtype RNase A (RNase A-NT) and catalytic dead RNase A (RNase A (CD)-NT). $n = 8$ cells from 3 independent biological replicates; two-sided unpaired Student's t -test; *** $p < 0.0001$. Scale bar, 10 μ m. Data are shown as mean \pm SD.

Figure S4G. Immunoblotting showing the equal expression of NONO-TFE3 fused with wildtype RNase A (RNase A-NT) and catalytic dead RNase A (RNase A (CD)-NT). GAPDH were used as loading control.

3.6. Figure 5 shows a significant reduction in transcription at NONO-TFE3 target genes, with no comparable effect observed at genes not regulated by NONO-TFE3 following PSPC1 depletion. Previous studies have indicated that PSPC1 plays a role in genome-wide transcription activation in ESCs (Shao, W., Bi, X., Pan, Y. et al. Phase separation of RNA-binding protein promotes polymerase binding and transcription. *Nat. Chem. Biol.* 18, 70–80 (2022)). Interestingly, PSPC1 appears to specifically interact with NONO-TFE3, selectively activating transcription of fusion target genes in tRCC. Further investigation is warranted to clarify how PSPC1 function differs in cells containing TFE3 fusions compared to other cell types.

Response: We appreciate the reviewer for raising this point. After carefully re-analyzing our data, we provided some potential explanations:

Based on our own CUT&Tag and RNA-seq data shown in **Figure 5B**, only 40% of PSPC1 binding regions overlapped with NONO-TFE3 binding sites, while only 18% of PSPC1-regulated genes overlapped with NONO-TFE3 regulated genes. On the other hand, more than 90% of NONO-TFE3 binding sites overlapped with PSPC1, and more than 60% of NONO-TFE3 regulated genes are regulated by PSPC1. These data suggest that NONO-TFE3 mediated transcriptional regulation heavily relies on PSPC1, while PSPC1 has a broader impact on transcriptional regulation. Since the survival of UOK109 cells is heavily dependent on NONO-TFE3, the genes co-regulated by NONO-TFE3 and PSPC1 are essential to support its survival, while other PSPC1-regulated genes might not be essential for UOK109 cell survival even if they show altered expression upon PSPC1 depletion.

We further classified the global genes into 3 clusters based on the RNA-seq data. Cluster 1 includes genes regulated by both NONO-TFE3 and PSPC1 (the overlapped 500 genes in **Figure 5B**). PSPC1 depletion significantly reduced the Pol II binding at these genes. Cluster 2 consists of genes only regulated by PSPC1 (2,173 genes in **Figure 5B**). We still observed a compromised occupancy of RNAPII at these genes after PSPC1 depletion, supporting the role of PSPC1 in transcription. This is consistent with the notion of the publication that reviewer mentioned. Cluster 3 includes genes that exhibit RNAPII binding but are not regulated by either NONO-TFE3 or PSPC1 (10,899 (total expressing genes with RNAPII binding)-500-286-2173=8,930 genes). These genes did not show an obvious reduction in RNAPII binding after PSPC1 depletion. In our manuscript, we define Cluster 2 and 3 as genes not regulated by NONO-TFE3. The inclusion of both clusters in the analysis may mask the RNAPII reduction intensities due to the overall signal dilution, leading to the conclusion that PSPC1 depletion does not cause a dramatic difference in RNAPII binding for NONO-TFE3 non-regulated genes (**Figure C2 for reviewer only**).

Figure 5B

Figure for reviewer only C2

Figure 5B. (Left) Venn diagram showing the overlapping genes occupied by NONO-TFE3 and PSPC1 in UOK109 KI cells. (Right) Venn diagram showing overlapping differentially expressed genes (DEGs) identified from RNA-seq in UOK109 KI cells treated with DMSO or dTAG-13 for 24 h (NONO-TFE3 regulated genes) or UOK109 KI cell with and without PSPC1 knocking down (KD) (PSPC1 regulated genes). **Figure for reviewer only C2.** Metagene binding profiles of RNA Pol II-total in control (shScramble, black line) and PSPC1 knocking down (shPSPC1, red line) UOK109 KI cells at indicated gene clusters.

In addition, the timing of RNAPII binding analysis after PSPC1 depletion may be a factor contributing to the discrepancies between the results by Shao W et al. (PMID: 34916619 as mentioned by Reviewer 3) and ours. In their degron system, RNAPII phosphorylation levels decreased 4–6 hours after PSPC1 degradation induced by IAA treatment but recovered to the baseline after 24 hours IAA treatment (Figure 5a of their paper). This suggests the presence of a compensatory mechanism that maintains global RNAPII activity at a steady-state level. However, ChIP-seq for RNAPII was only conducted at the 6-hour time point but not 24 hours of IAA treatment (Figure. 5c of their paper). In contrast, we employed shRNAs to knock down PSPC1, which required more than 24 hours to achieve effective depletion, a timeframe during which RNAPII activity would have already recovered according to their findings.

At last, NONO, PSPC1, and SFPQ belong to the Drosophila behavior/human splicing (DBHS) protein family, which functions as a molecular scaffold to mediate protein-protein and protein-nucleic acid interactions (PMID: 29719248, 29358041, 28192372, 27084935 and 29483655). Based on previous research, these three proteins form homo- or hetero-dimers with different combinations (NONO/PSPC1, NONO/SFPQ, PSPC1/SFPQ) to achieve their biological functions, including transcriptional regulation, RNA splicing, and DNA damage response. In the data collected from mESCs as pointed out by the reviewer, although PSPC1 plays an important role in genome-wide transcriptional activation, deletion of PSPC1 has a minor impact on mESC survival or differentiation (PMID: 29483655), possibly due to the compensatory effects from NONO and SFPQ. On the other hand, NONO is occupied by TFE3 in UOK109 tRCC cells; therefore, these cells are more sensitive to the depletion of DBHS family members, such as PSPC1. This speculation is further supported by the data observed in 786-O RCC cells without the NONO-TFE3

oncofusion event. As shown in **Figure 4J**, deletion of PSPC1 has no impact on the survival of 786-O RCC cells (without TFE3 oncofusion event), further supporting the unique function of PSPC1 in regulating TFE3 oncofusion tRCC. Although the deletion of PSPC1 might have a broad impact on transcriptional output, PSPC1 might be dispensable in cell types without TFE3 fusion events, possibly due to compensation from other DBHS members.

3.7. The authors proposed that TFE3 oncofusion condensate can promote the formation of transcription hubs and enhance prooncogenic transcription. It is reasonable because Zuo, L. et al. 2019 (10.1038/s41467-021-21690-7) has also reported the FET fusion proteins can form condensate, recruit Pol II CTD and promote transcription. However, to provide direct evidence to verify the downstream process of TFE3 oncofusion condensate is correct, the authors can provide data to show whether disrupting the condensate would decrease prooncogenic transcription under the condition that both NONO-TFE3 and PSPC1 are endogenously expressed (for example, by disrupting condensation using the method shown in Fig. 7).

Response: Following the reviewer’s suggestions, we performed RNA-seq on UOK109 cells expressing CB-LaG-MBP with rapamycin treatment. As demonstrated in the PCA plot and heatmap, rapamycin-induced disruption of NONO-TFE3 condensation largely resembled the transcriptomic changes observed with dTAG-13 treatment (**NEW Figure 8G-I, NEW Figure S8H**). These results further support our conclusion that the condensate-forming ability of NONO-TFE3 is essential for its transcriptional activity.

Figure 8G and H. Principal component analysis (G) and heatmap (H) of RNA-seq results with UOK109 KI cells treated with dTAG-13 or rapamycin (Rapa). n = 2 independent biological replicates.

Figure 8I. Venn diagram showing overlapping differentially expressed genes (DEGs) identified from RNA-seq in UOK109 KI cells treated with DMSO, dTAG-13 or Rapamycin (Rapa) for 4 hours.

Figure S8H. Pearson correlation analysis of RNA-seq data from UOK109 KI cells treated with indicated conditions. n= 2 independent biological replicates.

Minor points:

3.8. Page6, line 8, there should be “SLAM-seq” rather than “SALM-seq”.

Response: The typo has been corrected.

3.9. The author needs to make the name of the model fusion proteins clearer in the manuscript, like Gal4DBD-NONO-TFE3 rather than NONO-TFE3.

Response: The name of NONO-TFE3 fused with different tags have been updated throughout the manuscript.

3.10. Fig.4H, 4I, Fig.6D, the scale bars are needed in the panels.

Response: Scale bars are added.

3.11. Do all TFE3 oncofusions with RNA binding partner have similar structures? The author may add a schematic diagram to show the structures in the Supplementary Fig. 1.

Response: As requested by the reviewer, we have added a schematic diagram illustrating the structures of TFE3 fusions with RNA-binding partners (**NEW Figure S1C**). The RNA recognition motif (RRM) and K homology (KH) domains are highlighted as the RNA-interacting regions within these onco-fusions.

Figure S1C

Figure S1C. Scheme showing the domain structures of RBP fused TFE3 onco-fusions

5. In the analysis of Fig. 2, it may be better to add a control group of TFE3 with no fusion like that in Fig. 1. This addition can further demonstrate the importance of NONO-TFE3-RNA interactions on gene expression.

Figure S4L

Figure S4M

Figure S4L-M. PCA plot (L) and heatmap (M) of RNA-seq with dTAG-13 treated UOK109 KI cells stably expressing indicated NONO-TFE3 truncations. NT: NONO-TFE3. n = 2 independent biological replicates.

Response: We appreciate the reviewer's suggestion. RNA-seq on UOK109 cells expressing all NONO-TFE3 truncates and TFE3 mutants were performed (**NEW Figure S4L-M**). As shown in the PCA plot and heatmap, full-length NONO-TFE3 largely restored gene expression. However, NONO-TFE3 truncations deficient in RNA binding or condensation formation and TFE3 mutants (TFE3-S321A and TFE3-C (the part that fused with NONO)) showed impaired transcriptional activity and failed to restore the transcriptome to a state similar to that of UOK109 KI cells prior to dTAG-13 treatment. These data support our conclusion that interactions between NONO-TFE3 and RNA are essential for the regulation of its target gene expression.

3.12. In Fig. S3C, the author provided a characterization parameter called "granularity", but it is not mentioned in the main text or methods.

Response: We apologize for the missing definition of granularity, a condensation characterization parameter adopted from a previous publication (PMID: 34526716). Granularity, referring to the fluorescence signal fluctuation, is calculated as the standard deviation (S.D.) of the relative signal intensity along a line plot (normalized to the mean signal intensity of the entire line) at each pixel through an individual cell. This parameter was utilized to compare the condensate-forming capability of different proteins. To make the definition clear to the readers, we have drawn a line in the representative images and provided detailed explanations in the corresponding caption.

3.13. In the analysis of droplet assay in Fig. 6D, it may be better to use the participant ratio of pS5-CTD than the droplet area to show if the recruitment of pS5-CTD is promoted when PSPC1 was added.

Response: Following the reviewer's suggestions, we include the analysis of participant ratio (co-efficiency) in **NEW Figure 6D (bottom panel)**.

3.14. To verify the effect of RNA on the condensation and Pol II recruitment, the authors need to add RNA with different lengths (or sequences) to the current droplet experiments.

Response: To address this concern, we treated the *in vitro* droplets formed by mCherry fused NONO-TFE3, BFP fused RNAP II CTD and GFP fused PSPC1 with antisense and sense RNA strand (200 nucleotides from 5' of RRM2 mRNA), the antisense RNA exhibited the same quantitative effects as the sense strand, demonstrating that charge balance, rather than RNA sequence, governs the condensate formation of the NONO-TFE3 complex. The related data are included in **NEW Figure S7B**.

Figure S7B. Representative images (left) and quantification (right) of *in vitro* droplets formed by mCh fused NONO-TFE3, BFP fused Pol2 CTD and GFP-fused PSPC1 with antisense or sense strand RNA (200bp from the transcriptional start site of RRM2 gene).

Reviewer #4

General comments

Oncofusion proteins may contribute to the formation of liquid-like condensates due to the structural and functional properties of the fusion itself. These condensates may play a role in cancer, for example by altering chromatin biology or influencing cellular signaling. In this study, Dr. Guo et al. investigate NONO-TFE3, the product of a chromosomal translocation placing the transcription factor TFE3 under the regulation of NONO, a nuclear RNA-binding protein. This chromosomal translocation is responsible for translocation-RCC (t-RCC), an aggressive type of kidney cancer. Previous research has shown that the intrinsically disordered coiled-coil domain in NONO facilitates the formation of NONO-TFE3 condensates (PMID: 33592266). Here, the authors use various omics approaches and a dTAG-based inducible system to control NONO-TFE3 levels in UOK109 cells derived from a patient carrying NONO-TFE3 fusion. They demonstrate that the RNA-binding ability provided by NONO is essential for condensate formation, which in turn promotes tRCC growth both in vitro and in vivo.

Using a dual approach that combines TurboID-proximity proteomics and CRISPR screening, they identify PSPC1, a paraspeckle-associated RNA-binding protein, within NONO-TFE3 condensates. They further show that PSPC1 is essential for NONO-TFE3-driven tumor cell growth (in UOK109) but not for VHL-associated RCC (in 786-O cells). The NONO-TFE3 and PSPC1 co-condensates promote the recruitment of RNA polymerase II (RNAPII) to chromatin. Notably, by applying a chemogenetic tool, they show that dissolving these condensates significantly inhibits tumor growth.

Overall, this study is well-designed and several data support the notion that NONO-dependent oncofusion condensates play a role in tRCC. However, the underlying mechanisms remain unclear. If the authors' claim is that increasing the association of RNAPII with chromatin enhances the transcriptional activity of TFE3 on its target genes, this must be demonstrated through gene expression analysis. Importantly, TFE3, along with other members of the MiT/TFE family (notably TFEB and MITF), are oncogenic even outside of oncofusion contexts, with their cancer-promoting roles strongly dependent on their transcriptional activity (PMID: 16001072, PMID: 26168401, PMID: 28619945, PMID: 36987696, PMID: 34779410, PMID: 38195686). TFE3 oncofusions may drive cancer progression by increasing TFE3's nuclear localization (which is otherwise regulated by mTORC1 in response to nutrient availability) and possibly by enhancing TFE3 stability. In this study, the authors compare the genomic binding sites of NONO-TFE3 with TFE3-S321A (a constitutively nuclear form of TFE3 that cannot be phosphorylated by mTORC1) and show that both share 85% of genomic binding sites in UOK109 cells. However, they observe that UOK109 cell growth under constitutively active TFE3 expression is similar to that seen in TFE3-silenced cells (Fig. 3I, Fig. S3E). These findings are surprising and suggest potential technical issues, such as inefficient expression of the TFE3-S321A construct. To fully explore the role of the NONO component, it would be ideal to generate UOK109 cells lacking the NONO segment but endogenously expressing TFE3-S321A and then compare transcriptional profiles and oncogenic activity in these cells relative to the parental line expressing NONO-TFE3.

In addition, I have some concerns related to the selected list of "primary targets" (see specific comments).

Specific comments

4.1. Fig. 1J, K: How are UOK109 cells overexpressing TFE3-S321A or NONO-TFE3 generated? Is this achieved through knockout followed by stable transfection? Western blot data should be included to confirm

comparable expression levels of the two constructs, and RNA-seq data should be provided for further comparison.

Figure for reviewer only D1

Figure for reviewer only D1. Immunoblotting showing the protein expression levels of NONO-TFE3, TFE3-S321A and TFE3-C in dTAG-13 treated UOK109 KI cells.

Figure 1J and K. Venn diagram (J) and heatmap binding profiles of NONO-TFE3 or TFE3-S321A CUT&Tag signals (K) in dTAG-13 treated UOK109 KI cells transfected with Flag tagged NLS-TFE3 S321A. n = 2 independent biological replicates.

Figure 1J

Figure 1K

Response: We apologize for the missing information about these cells. In original Fig 1J and K, we used UOK109 WT cells overexpressing NONO-TFE3 and TFE3 S321A to perform the CUT&Tag assay. In this overexpression system, we cannot rule out the possibility that endogenous NONO-TFE3 might interfere with the binding of overexpressed NONO-TFE3 and TFE3-S321A. Additionally, overexpression can introduce

artifacts, potentially leading to the identification of false binding targets. To overcome this limitation and address the similar concerns raised by Reviewer 2, we followed the reviewer's advice by transfecting NONO-TFE3 knockout UOK109 cells with TFE3-S321A. However, we were unable to proceed by knocking out NONO-TFE3 first and then transfecting the knockout cells with TFE3, as attempts to generate viable single clones with NONO-TFE3 knockout using CRISPR-Cas9 were unsuccessful. This aligns with our observation that the deletion of NONO-TFE3 leads to cell death in UOK109 cells. Alternatively, we transfected dTAG-13-treated UOK109 KI cells with TFE3-S321A, which effectively achieved the desired outcome. Briefly, UOK109 KI cells were transfected with plasmid encoding Flag-NLS-TFE3-S321A while simultaneously treated with dTAG-13. After 24-hour treatment, the transfected Flag-NLS-TFE3-S321A construct was successfully expressed, and the endogenous NONO-TFE3 was effectively depleted. The expression level of Flag-NLS-TFE3-S321A was comparable to the endogenous NONO-TFE3 in this transient transfection settings, as evaluated by western blot analysis (**Figure D1 for reviewer only**). Cells were then harvested and subjected to CUT&Tag assay to assess the chromatin binding profile of the transfected TFE3-S321A. The new CUT&Tag profiling of NONO-TFE3 and TFE3-S321A revealed that they share ~90% of genomic binding sites. However, NONO-TFE3 exhibited enhanced genomic binding compared to TFE3-S321A, which is consistent with our previous data obtained from the overexpression system, i.e., NONO-TFE3 remains at the majority TFE3 genomic binding sites with stronger chromatin association features. The updated data are shown in **NEW Figure 1J-K**.

In parallel, RNA-seq was conducted to examine the gene expression changes as suggested by the reviewer. A comprehensive analysis is presented in the response to Question 4.7 below.

4.2. Fig. 2F-J: The authors define "primary transcripts" as those that decrease after 4 hours of TFE3 degradation. It seems more likely that the first transcripts to decrease are simply the less stable ones. The true "primary targets" would more accurately be those that increase rapidly upon washout, though this point may be less critical given the chronic CUT model of TFE3 activity being studied. Additionally, the comparisons in Fig. 2F and H are not entirely accurate. Cut & Tag or RIP-seq analysis following 4 hours of TFE3 degradation reflects changes due to reduced NONO-TFE3 binding to DNA/RNA, whereas SLAM-seq after 4 hours highlights downregulated transcripts based on stability. This approach does not fully align with the data being gathered. Genes bound by NONO-TFE3 but not yet downregulated may, in fact, be the ones most influenced by the RNA-binding capacity of the oncofusion. A more consistent comparison would

involve DMSO versus dTAG at 24 hours across all datasets, focusing on transcripts significantly deregulated and enriched for NONO-TFE3 binding (as per Cut & Tag or traditional ChIP-seq).

Response: SLAM-seq kit (LEXOGEN) includes two modules: Anabolic module Kit (Cat#061) and Catabolic module Kit (Cat#062), which serve different purposes. As shown in **Figure D2 for reviewer only**, the anabolic module, which we used in the manuscript, is specifically optimized to detect RNA synthesis rates over short (pulse) S4U labeling durations in response to stimuli, e.g., a fast-acting drug candidate or the acute deletion of a specific gene. This approach is widely used to identify primary targets of transcriptional factors or epigenetic regulators (PMID: 29622725, 36455613, 36859550). As illustrated in the schematic workflow, at t_0 , modified nucleotides (S4U) are added, which label newly synthesized RNA (nascent, in green) while existing RNA (in black) is unlabeled. At t_x , RNA synthesis is stopped by cell lysis and RNA isolation. Sampling at different intervals allows for measurement of transcript synthesis rates. Importantly, it has been demonstrated that RNA stability does not impact the conclusions drawn using this Anabolic Module (PMID: 28945705, 29622725). This feature makes it highly reliable for our purpose of analyzing the primary targets of NONO-TFE3.

Figure for reviewer only D2. Scheme depicting the workflows of 2 modules of SLAM-seq Kit.

Taking 4-hour dTAG-13 treatment as an example, UOK109 KI cells were divided into 3 groups: (1). Control group: treated with DMSO for 4 hours; (2). Experimental group: treated with dTAG-13 for 4 hours; (3). Background group: treated with dTAG-13 for 4 hours but without subsequent S4U labeling. After 4 hours treatment of DMSO or dTAG-13, S4U was added to the control and experimental groups for an additional 2-hour nascent RNA labelling. Since the 4-hour dTAG-13 treatment induced 50% reduction of NONO-TFE3 protein, the incorporation of S4U to the nascent transcripts of its target genes should proportionally decrease compared with the DMSO-treated control group. Then, the S4U labelled nascent transcripts were chemically alkylated with iodoacetamide to introduce a signature modification in

the RNA sequence and result in T to C mutation after reverse transcription and PCR. For the analysis, transcripts with T to C mutations were selected (using Background group to minimize the nonspecific noises) and those that were differently expressed between control and experimental group are classified as early response genes after acute depletion of NONO-TFE3. In this labeling strategy, the decreased transcripts identified are attributed to reduced synthesis of nascent RNA rather than the instability of these RNA molecules.

By using a shorter dTAG-13 treatment to capture the early transcriptional response and integrating NONO-TFE3 binding profiles (CUT&Tag), this approach enhances the accuracy of identifying direct targets or immediate targets (primary target genes) of NONO-TFE3. Prolonged drug treatment could lead to the detection of secondary targets resulting from downstream transcriptional changes. Notably, the expression of genes affected by the 4-hour dTAG-13 treatment was largely restored after dTAG-13 washout, further reinforcing that these genes are the direct or immediate targets (primary target genes) of NONO-TFE3.

In contrast, the Catabolic module allows the detection of RNA degradation rates (RNA stability) over time in response to a stimulus. This module uses a long-time S4U labeling step to allow RNA metabolism to reach an approximate steady-state level. Then, the exchange of S4U for unlabeled uridine in the cell culture media stops RNA labeling. Sampling occurs over a time course after unlabeled uridine is added. As illustrated in the scheme, cells are first cultured in S4U-containing media for a specified period (Δt), which

may extend for up to 24 hours, to establish approximate steady-state labeling of the RNA. Then, at time point t_0 , the culture media is replaced with the one containing unlabeled uridine (U), which replaces S4U in the cells and terminates the labeling of newly synthesized RNA. From this moment, only the RNA synthesized prior to the media change retains the label (green), while all newly synthesized RNA after this point remains unlabeled (black). Sampling at different intervals (t_x) allows for measurement of transcript degradation rates by determining the proportion of labeled RNA.

In addition, if the authors claim that NONO enhances RNAPII recruitment to TFE3-bound DNA, they should support this by comparing transcript levels in UOK109 KI cells expressing TFE3-S321A (without NONO).

A comprehensive RNA-seq analysis is presented in the response to Question 4.7 below.

4.3. Fig. 2I: The model is not entirely clear. The authors propose that NONO-TFE3 influences transcriptional and post-transcriptional regulation of the same targets, resulting in enhanced expression of oncogenic targets, but only for “primary targets” identified after 4 hours of degradation. This claim lacks supporting evidence, and as noted above, the selection of "primary transcripts" may be somewhat arbitrary.

Response: As we stated above, we used **Anabolic module** to perform the SLAM-seq, which is indeed used to identify the “primary/direct/immediate targets” of NONO-TFE3. To further clarify the proposed model, additional experiments were performed:

(1) Nascent RNA labeling: To demonstrate that nascent RNA is an integral component of NONO-TFE3 condensates, we employed an RNA labeling approach that permits the visualization of newly synthesized RNA within cells temporally and spatially (PMID: 18840688). This strategy involves the incorporation of 5-ethynyl uridine (EU), an alkyne-modified nucleoside, into nascent RNA, followed by “click chemistry” with an azide-conjugated fluorescent dye. As illustrated in **NEW Figure S4G**, we detected pronounced colocalization between nascent RNA and NONO-TFE3 condensates, highlighting the ability of NONO-TFE3 to associate with newly synthesized transcripts.

(2) A reporter system to evaluate feedback regulatory dynamics: Leveraging a previously published reporter system (PMID: 33333019), we explored how artificially manipulating the levels of feedback RNAs influences the transcriptional activity of NONO-TFE3. Briefly, we incorporated a specific TFE3 DNA binding sequence (GTCACGTGAC, 6x) upstream of the promoter of the gene encoding luciferase to use the expression level of luciferase as a readout to indicate the transcriptional activities of NONO-TFE3. In parallel, a DNA sequence encoding RNAs (200 nucleotides from 5' of RRM2 cDNA, which was identified from our RIP-seq results) was cloned into the locus proximal to the gene encoding luciferase under the control of NONO-TFE3 binding. The expression of this RNA is designed to be under the control of doxycycline (Dox) (**NEW Figure 2I, top**). We then co-transfected this reporter plasmid, together with a plasmid encoding NONO-TFE3, followed by treatment with escalating doses of doxycycline. Different amounts of RNA derived from this reporter system were produced in response to the escalating doses of doxycycline treatment. The impact of adjacent RNA levels on the transcriptional activities of NONO-TFE3 was reflected by the expression of luciferase as measured by luminescence (**NEW Figure 2I, bottom**). As shown in **NEW Figure 2I** and **NEW Figure S3D**, we observed that increasing levels of feedback RNAs initially promoted reporter expression but subsequently suppressed it when RNA levels became excessively high. This data is consistent with our in vitro co-condensation and sedimentation assays (**NEW Figure 6C-E**) and previous publications (PMID: 28556382, 29569441, and 33333019). The results presented here aligned with the current understanding that transcription is a non-equilibrium process mediated by electrostatic forces that provides dynamic feedback through its RNA products (PMID: 27161661, 30950394 and 33333019). The mechanism underlying this phenomenon is nonequilibrium regulation of complex coacervation, a type of

liquid–liquid phase separation mediated by electrostatic interactions between oppositely charged polyelectrolytes. In the context of transcription, one of the polyelectrolytes is negatively charged RNA, and the others are transcriptional molecules with intrinsic disordered regions (IDRs) containing a net positive charge. Low levels of RNA can enhance condensate formation because of favorable interactions between oppositely charged species. The addition of RNA beyond the point where the opposite charges compensate each other, results in an excess of negatively charged RNA molecules that repel each other. Thus, high RNA levels can cause condensate dissolution (PMID: 28556382, 29569441).

(3) *In vitro* transcriptional assay: We checked the effects of RNA and PSPC1 in a fully defined *in vitro* transcription system with a DNA template containing a heteroduplex bubble that has been widely used as a nucleic acid scaffold (PMID: 34916619). RNAPII can bind to single-stranded DNA within the bubble without the help of general transcription factors. We then added NONO-TFE3 in the presence or absence of PSPC1 together with RNAPII in this assay. The transcriptional activity was measured by qRT-PCR using primers against the RNAs derived from the template (**NEW Figure 7B and 7D**). We observed that NONO-TFE3 enhances transcriptional output, while PSPC1 + NONO-TFE3 showed more than 3-fold enhanced transcriptional output compared with the control. Furthermore, this enhanced transcriptional activity was not observed in the NONO-TFE3-4A mutation with or without PSPC1. These results further support our

Figure S4G. Immunofluorescence of NONO-TFE3 and 5-ethynyl uridine (EU) labelled RNA in UOK109 cells. Cells without EU adding in culture medium were used as control. Scale bar, 10 μ m.

Figure 2I. Scheme depicting the reporter system (top) where local RNA expression near a luciferase reporter gene can be induced by Doxycycline (Dox). Transcriptional activity of NONO-TFE3 were monitored by luciferase intensity (normalized to 0 ng/mL Dox, bottom). $n = 6$ independent biological replicates, one-way ANOVA with Tukey's post-hoc test, *** $p < 0.0001$. Data were shown as mean \pm SD.

Figure 3D. qRT-PCR analysis of relative RNA expression level (normalized to 0 ng/mL Dox) induced by indicated doses of Doxycycline (Dox). $n = 6$ independent biological replicates, one-way ANOVA with Tukey's post-hoc test, *** $p < 0.0001$. Data were shown as mean \pm SD.

Figure 6C. Representative images of *in vitro* droplets formed by mCh fused NONO-TFE3, BFP fused Pol2 CTD and GFP-fused PSPC1 with indicated concentrations of RNA. GFP alone was used as control. Scale bar, 10 μ m.

Figure 6D. Quantifications of droplet area (top) and co-efficiency of *in vitro* droplet formation performed in (C). ($n = 30$ droplets from 3 independent biological replicates; two-sided unpaired Student's t-test; *** $p < 0.0001$). Data are shown as mean \pm SD.

Figure 6E. Schematic illustration of the droplets sedimentation assay (top) and immunoblotting of CTD incorporation in droplets (bottom).

hypothesis that low concentrations of RNA produced in these systems work together with NONO-TFE3 and PSPC1 to support RNAPII-mediated transcription.

Figure 7B

Figure 7D

Figure 7B. Schematic diagram of the reconstituted *in vitro* transcription assay.

Figure 7D. Quantifications of *in vitro* transcription reaction activity (normalized to RNAPII only) with indicated recombinant proteins. (n = 6 independent biological replicates; two-sided unpaired Student's t-test; *** p<0.001). Data are shown as mean ± SD.

Collectively, these three sets of experiments suggest the existence of nascent RNA in NONO-TFE3 condensates and the dynamic regulation of adjacent RNA on the transcriptional activity of NONO-TFE3. We also realize the limitations of using the reporter system since it cannot authentically mimic the regulation of RNA production on the transcriptional activities of NONO-TFE3 at the same locus. We hope the reviewer understands the technical challenges. At least the reporter system provided compelling evidence to illustrate the impact of adjacent RNA levels on regulating the transcriptional activities of NONO-TFE3. We have also provided an additional

discussion on this point in the revised "Discussion" section.

4.4. Fig. 2J: This comparison considers only a subset of selected "primary" genes. A good correlation should also be expected with the larger set of target genes downregulated after 24 hours. However, this

Figure for reviewer only D3. Gene set enrichment analysis of NONO-TFE3 target genes identified from SLAM-seq in UOK109 KI cells (24-hour dTAG-13 treatment) and genes up-regulated (logFC > 3 and FDR < 0.05) in tRCC.

correlation is expected, as it involves comparing genes upregulated in tRCC-cells relative to those upregulated in tRCC patient tumors.

Response: As suggested by the reviewer, we further examined the correlation of NONO-TFE3-regulated genes with patient samples by analyzing those downregulated after 24-hour dTAG-13 treatment. Indeed, as reviewer expected, there are strong correlations between DEGs identified from the UOK109 KI cells treated with dTAG-13 for 24 hrs and gene upregulated in tRCC patient tumors. These data suggested that regardless of how "primary" or "secondary" genes are defined, the transcriptional networks governed by NONO-TFE3 in UOK109 cells hold significant clinical relevance. The related data is included in **Figure D3 for reviewer only**.

4.5. Fig. 3A: Was an additional NLS added to the mutant TFE3? This appears to be an error; please clarify this point.

Response: Yes, an additional NLS sequence was introduced into the mutant TFE3 due to the inadequate nuclear localization of TFE3 S321A. Fluorescence imaging revealed that, in comparison to WT TFE3, which is primarily localized in the cytosol, TFE3 S321A exhibits partial nuclear localization (70-80%) but does not achieve an entire nuclear localization as observed in NONO-TFE3 (100%) (check details in Question 4.7). To ensure full nuclear localization of TFE3 S321A comparable to NONO-TFE3, an additional NLS sequence was added to drive its total nuclear translocation.

4.6. Fig. S3D: The levels of TFE3-CA are lower relative to other constructs, which could indicate technical issues affecting assessment of TFE3-S321A's oncogenic potential.

Figure for reviewer only D1

Figure S4J

Figure for reviewer only D1. Immunoblotting showing the protein expression levels of NONO-TFE3, TFE3-S321A and TFE3-C in dTAG-13 treated UOK109 KI cells 24 hours after transfection.

Figure S4J. Immunoblotting showing the protein expression levels of NONO-TFE3, TFE3-S321A and TFE3-C in dTAG-13 treated UOK109 KI stable cells.

Response: We appreciate that the reviewer brought up this critical concern. While transient transfection allows us to achieve comparable expression levels of NLS-TFE3-S321A and NONO-TFE3 (**Figure D1 for reviewer only**), we observed that UOK109 KI cells stably expressing TFE3-S321A exhibit reduced protein levels compared to endogenous NONO-TFE3 following dTAG-13 treatment (**NEW Figure S4J**). Based on the previous publication, the relatively lower expression level of TFE3-S321A is unlikely due to the technical issue. Previous study (PMID: 36608670) identified an evolutionarily conserved phosphorylation-dependent degron sequence (E46-

D52) in the N-terminus of TFE3 which is essential for its ubiquitination and subsequent proteasomal degradation by the E3 ligase CUL1 ^{β -TrCP1/2}. Mutation or deletion of this degron sequence significantly enhances TFE3 protein stability. In the case of NONO-TFE3, the degron-containing N-terminal is replaced by NONO, avoiding E3 ligase mediated degradation. Consequently, NONO-TFE3 exhibits enhanced protein stability and higher levels compared to TFE3-S321A, which retains the degron sequence at the N-terminus.

To overcome this limitation, we followed reviewer's advice by generating the UOK109 KI cells expressing the C-terminal truncate of TFE3 (TFE3-C, the part that fused with NONO that lacks the degron). After dTAG-13 treatment, both transiently transfected cells and the stable cell line exhibited comparable protein levels of NONO-TFE3 and TFE3-C (**NEW Figure S4J and Figure D1 for reviewer only**). Furthermore, TFE3-C is exclusively located in the nucleus. Similar as TFE3-S321A, TFE3-C is unable to rescue the oncogenic signatures observed in UOK109 KI cells with the intact NONO-TFE3. Detailed experimental procedure and explanation are described in the response to Question 4.7.

4.7. Fig. S3E - Fig. 3I: It seems implausible that cells overexpressing TFE3-CA would mimic TFE3-silenced cells in terms of proliferation and tumor growth. Nuclear localization and transcriptional activity (RNA-seq) should be compared between NONO-TFE3 and TFE3-S321A, as well as between mutant constructs for the NONO fragment. To conclusively show that the NONO segment drives tumor growth, the authors should consider generating UOK109 KI cells lacking the NONO segment but mutated at Ser321.

Response: As described in our response to Question 4.1, these stable cell lines expressing NONO-TFE3 truncations and TFE3-S321A were established in dTAG-13-treated UOK109 KI cells. Briefly, the lentivirus encoding NONO-TFE3 truncations, TFE3-S321A, or TFE3-C was transduced into UOK109 KI cells to generate stable cell lines, followed by dTAG-13 treatment to deplete endogenous NONO-TFE3 in UOK109 cells. The colony formation assay, as listed in **NEW Figure S4I**, was performed using these cell lines.

The following additional experiments were performed as requested by the reviewer:

Nuclear localization: The nuclear localization of these truncations was measured and quantified using fluorescence imaging analysis since GFP was fused with NONO-TFE3 truncations, TFE3-S321A, or TFE3-C (**Figures for reviewer only D4 and D5**). The majority (>90%) of WT TFE3 is located in cytoplasm, which is consistent with previous publications (PMID: 27298091, 24448649). More than 80% of TFE3-S321A is

located in the nuclei. Furthermore, NONO-TFE3, its truncations, and TFE3-C are all exclusively located within the nuclei.

Figure for reviewer only D4

Figure for reviewer only D5

Figure S4L

Figure S4I

Figure S4M

Figure for reviewer only D4 and D5. Representative images (D2) and quantification (D3) of the nuclear localizations of TFE3 and NONO-TFE3 truncations (GFP labelled). Hoechst was used for nuclear staining. (n = 3 independent biological replicates). Scale bar, 10 μm. Data are shown as mean ± SD.

Figure S4I. Representative images (top) and quantification (bottom) of colony formation assay results using UOK109 KI cells treated with dTAG13 (500nM) and rescued with expression of indicated NONO-TFE3 truncation variants. (n = 3 independent biological replicates; one-way ANOVA with Tukey's post-hoc test, *** p<0.0001). Data are shown as mean ± SD.

Figure S4L-M. PCA plot (L) and heatmap (M) of RNA-seq with dTAG-13 treated UOK109 KI cells stably expressing indicated NONO-TFE3 truncations. NT: NONO-TFE3. n = 2 independent biological replicates.

RNA-seq: RNA-seq was performed in the cells mentioned above expressing TFE3 variants (**NEW Figure S4L-M**). As shown in the PCA plot and heatmap, full-length NONO-TFE3 largely restored gene expression. However, NONO-TFE3 truncations deficient in RNA binding or condensation formation and TFE3 mutants (TFE3-S321A and TFE3-C) showed impaired transcriptional activity and failed to restore the transcriptome to a state similar to that of UOK109 KI cells prior to dTAG-13 treatment. These data clearly demonstrate that restoring the nuclear function of TFE3 (S321A or C-terminus) alone cannot support the oncogenic transcription of UOK109 cells. The fusion partner, i.e., NONO in this case, is required to mediate the oncogenic transcription of TFE3 and support tRCC cell survival.

Our entire manuscript aims to explain why the fusion partner is required to support TFE3, but not the overexpression of TFE3, to achieve the oncogenic function. Below is the summary based on our observations.

(1) The replacement of the N-terminus of TFE3 with fusion partners, such as NONO, enhances protein stability due to the lack of a degron sequence at the N-terminus of TFE3. This suggests that, at least in this tRCC cell line, a more stable and higher abundance of TFE3-related form is required to initiate and sustain oncogenic transcription.

(2) The fusion partner enables the interactions between TFE3 oncofusion with RNA, PSPC1, RNAPII, and other components to form nuclear condensates, thereby supporting oncogenic transcription and tRCC growth. This is further evident as TFE3-C fails to restore efficient gene expression, despite having protein levels comparable to NONO-TFE3.

(3) Although the increased expression of MiT family transcription factors (such as TFE3 or TFEB) has been reported to be involved in various cancers, transgenic mouse models with TFEB (a close family member sharing similar target genes with TFE3) overexpression in kidney only developed cysts and microscopic neoplastic lesions, which did not fully recapitulate the features observed in both TFEB and TFE3 oncofusion-induced human tRCC (PMID: 27668431). This model suggested that, at least in the context of tRCC, non-fusion forms of MiT family transcription factors, even with overexpression, are not sufficiently potent to drive full tumorigenesis. In stark contrast, transgenic mice with TFE3 oncofusion protein develop aggressive renal cancer that recapitulates the morphological, histological, and molecular features observed in human tRCC (PMID: 38386415). These data strongly suggest that a fusion partner is required for TFE3 to mediate oncogenic transcription and contribute to the malignant transformation of tRCC, while simply upregulating TFE family members might not be sufficient to mediate the pathogenesis of tRCC.

Minor comments:

8. Review figure legends to ensure they clearly describe dTAG-13 treatment details.

Response: In response to the reviewer's request, we have included more detailed explanations of the procedures used to treat cells with dTAG-13 in the revised "Method" section to ensure greater clarity and understanding.

9. Correct the term "ex vivo" throughout the paper, as it is used inappropriately to describe in vitro experiments.

Response: We thank the review for pointing out this oversight. We have rephrased "ex vivo" to "in vitro"

10. On page 4: Update the figure reference from Fig. S1G to the correct Fig. S1H where reported: "RIP-seq data also revealed the enrichment of similar gene sets"

Response: We have corrected this accordingly in our revised manuscript.

Point-to-Point Responses to Reviewer #4

4.1. In this revised version of the manuscript, the authors have added data suggesting that nascent RNA is present within NONO-TFE3 condensates and provided functional assays to support a role for adjacent RNA in modulating the transcriptional activity of the fusion protein. However, the response to point 4.7, concerning the lack of oncogenic activity of the constitutively active TFE3-S321A mutant, remains inadequate.

Response: We thank the reviewer for raising concerns regarding the oncogenic potential of constitutively active TFE3-S321A in tRCCs. We would like to clarify that our data does not contradict the oncogenic role of the nuclear form of TFE3, such as TFE3-S321A, in renal cancer in general. However, our data suggests that TFE3-S321A may not be sufficient to sustain oncogenic transcription specifically in tRCC, where cell survival relies on TFE3 oncofusion proteins. Below, we provide several lines of evidence and a point-by-point response to clarify this point.

4.2. First, the RNA-seq analysis (new Fig. S4M) includes only two biological replicates per group, which is not statistically sufficient. Even disregarding this major limitation, the data presented in the provided dataset show that well-established TFE3 target genes such as GPNMB, ATP6V1H, RRAGD, and TPP1 are not upregulated upon reintroduction of the TFE3-S321A construct. This strongly suggests technical issues with the construct itself, which likely explains why its re-expression fails to rescue cell growth and instead results in a phenotype comparable to TFE3 loss-of-function. This significantly weakens the conclusions drawn from this experiment.

Response: We thank the reviewer for highlighting the limitation of using two biological replicates for RNA-seq. We agree that additional replicates would enhance statistical power. To address this limitation, we ensured relatively high sequencing depth (average $\sim 21 \times 10^6$ reads / sample) and rigorous quality control, and we observed a high degree of concordance between two replicates (**Figure S4L and Figure A for reviewer only, left panel**). To further confirm the RNA-seq result, additional real-time quantitative PCR (qRT-PCR) is performed to validate key RNA-seq findings, including the mRNA expression levels of *GPNMB*, *TPP1*, *RRAGD*, and *ATP6V1H* as mentioned by the reviewer ($n = 6$ independent biological replicates), thus providing additional support to our conclusions (**Figure A for reviewer only, middle and right panels**). Furthermore, these results are in line with our CUT&Tag data, which indicate that the

Figure A for reviewer only

Figure A for reviewer only. (left): Pearson correlation analysis of 2 independent biological replicates of RNA-seq samples with indicated treatments (2-tailed Student t-test). (middle) : TPM (transcripts per million) values from RNA-seq showing the expression levels of TFE3/TFEB target genes in UOK109 KI cells under the indicated treatments. (right): Real-time qPCR validation on the relative expression level (normalized to Actin expressions) of TFE3/TFEB target genes in UOK109 KI cells with indicated treatment. ($n = 6$ independent biological replicates; one-way ANOVA with Tukey's post-hoc test; *** $p<0.0001$, * $p<0.05$). Data are shown as mean \pm SD.

chromatin-binding strength of TFE3-S321A is lower than that of NONO-TFE3 (**Figure 1K and Figure B for Reviewer only**).

Figure 1K

Figure B for reviewer only

Figure 1K. Heatmap binding profiles of NONO-TFE3 or TFE3-S321A CUT&Tag signals in dTAG-13 treated UOK109 KI cells transfected with Flag tagged NLS-TFE3 S321A.

Figure B for reviewer only. Genome browser views of representative loci in UOK109 showing CUT&Tag binding signals of NONO-TFE3 and TFE3-S321A.

It is worth noting that TFE3-S321A was introduced into UOK109 cells, which harbor the NONO-TFE3 fusion protein and are adapted to NONO-TFE3 fusion-mediated oncogenic transcription. In our experimental setting, we compared the transcriptional outputs between NONO-TFE3 and TFE3-S321A in dTAG-13-treated cells, in which endogenous NONO-TFE3 was degraded. Under these conditions, TFE3 target genes may have already been upregulated by NONO-TFE3 in UOK109 cells compared to cells lacking the TFE3 fusion protein. Our data suggests that the introduction of TFE3-S321A cannot sustain the transcription of NONO-TFE3 target genes at comparable levels; however, our data does not suggest that TFE3-S321A fails to activate its target genes.

To further confirm this, we analyzed published RNA-seq data from HEK293T cells transiently overexpressing TFE3 or TFE3 fusion constructs (PMID: 34986355). As shown in **Figure C for reviewer only**, the introduction of TFE3 alone induced transcriptional activation compared with parental cells, while TFE3 fusion proteins led to more enhanced transcriptional activation of TFE3/TFE3 target genes (right panel).

This finding agrees with the primary conclusion of our study. As demonstrated in our study, replacement of the TFE3 N-terminus with fusion partners such as NONO facilitates interactions between the TFE3 oncofusion and various nuclear components, including RNA, PSPC1, and RNA polymerase II. These interactions promote the formation of transcriptional condensates, which enhance chromatin binding of the oncofusions and drive robust and sustained oncogenic transcription to support tRCC growth.

Figure C for reviewer only

Figure C for reviewer only. (left): Global expression levels of canonical TFE3/TFE3 target genes in HEK293T cells overexpressing the indicated constructs (PMID: 34986355). (The box plots indicated the median (center line), the third and first quartiles (box limits) and 1.5X interquartile range (IQR) above and below the box (whiskers). Two-sided Wilcoxon test; $p < 0.05$). (right): Paired matched dot blot showing expression changes of individual canonical TFE3/TFE3 target genes in HEK293T cells overexpressing TFE3 or TFE3 fusion (Two-sided Wilcoxon test; $p < 0.05$).

Of note, a recent study (PMID: 40240607) has challenged the long-standing model that transcription factors (TFs) regulate gene expression primarily through direct binding to regulatory elements upstream of their target genes. The study revealed a surprisingly limited overlap between TF binding sites and functional gene regulation. For most TFs, binding is often a poor predictor of whether expression of that gene responds to loss of the bound TF. These findings suggest that the ability of a TF to regulate gene expression *is highly context-dependent*, influenced by the chromatin landscape, genomic architecture, and cell-type-specific factors, *rather than being solely dictated by promoter occupancy*. In our case, although TFE3-S321A occupies chromatin regions similar to those bound by NONO-TFE3, it does not induce comparable levels of target gene expression. This discrepancy likely reflects the fact that UOK109 cells may have undergone extensive chromatin remodeling toward a tumor-permissive state, which requires the assembly of transcriptional condensates for robust oncogenic gene activation. Our TurboID proximity labeling and CRISPR-based functional screening revealed that NONO-TFE3 interacts extensively with nuclear proteins and RNAs, facilitating condensate formation that enhances chromatin engagement and transcriptional output. In contrast, TFE3-S321A lacks the capacity to form such condensates, and therefore, may not be able to fully rescue the transcriptional program driven by NONO-TFE3.

In transcription factors, both the DNA-binding domain (DBD) and activation domain (AD) are essential for their function. The DBD enables recognition and binding to specific DNA sequences within regulatory regions of target genes, while the AD facilitates interactions with cofactors, chromatin remodelers, and components of the transcriptional machinery to modulate gene expression. In the case of TFE3-C (the C-terminal fragment of TFE3 fused to NONO), despite maintaining protein levels comparable to the full-length NONO-TFE3 fusion, it failed to restore robust gene expression. This is likely due to the absence of the N-terminal AD, emphasizing that DNA binding alone is insufficient. Instead, functional transcription factors require coordinated interactions with other cofactors, including proteins and RNAs, to fully activate gene expression.

Collectively, we hope that the reviewer agrees that although TFE3-S321A is able to induce transcriptional activation, TFE3 oncofusion proteins, e.g., NONO-TFE3, exhibit more pronounced transcriptional activity compared with TFE3-S321A. In tRCCs, a stronger activation of TFE3 target genes or unique transcriptional programs driven by TFE3 oncofusions may be necessary to sustain tumor growth (more detailed evidence is provided in the next section – see our response to Comment 4.3).

4.3. At the end of their rebuttal, the authors also cite published data to argue that non-fusion forms of MiT/TFE transcription factors are insufficient to drive tumorigenesis. However, this interpretation is overly simplistic and partially incorrect. For instance, kidney cysts are commonly observed in tRCC patients (PMID: 34489456, PMID: 32699116), and are also present in additional murine models, not just the one cited (PMID: 27668431), but also in PRCC-TFE3 mice (PMID: 31043488). The relative abundance of cystic versus neoplastic lesions in mouse models likely depends on the timing and cell type specificity of TFEB/TFE3 induction. Furthermore, loss of FLCN, a tumor suppressor gene which leads to constitutive nuclear localization of TFEB and TFE3, results in robust tumorigenesis that is completely abolished by inactivation of either factor (PMID: 36987696). These studies clearly demonstrate that constitutive activation of MiT/TFE factors is sufficient to drive renal tumorigenesis, even in the absence of oncogenic fusions. Therefore, this part of the study is not sufficiently robust and undermines the overall conclusions of the manuscript. It cannot be accepted in its current form.

Response: We appreciate the reviewer's comments highlighting those previous studies: (1) transgenic mice with TFEB overexpression and (2) loss of the tumor suppressor gene FLCN, which leads to constitutive nuclear localization of TFEB and TFE3, was shown to induce tumorigenesis. We fully

acknowledge the significant value of these models in demonstrating the tumorigenic roles of TFEB/TFE3. However, we would also like to point that the biological contexts, including phenotype and molecular feature, of these models might differ from that of the translocation renal cell carcinoma (tRCC), which may account for the distinct outcomes observed in our TFE3-S321A rescue experiments. Below are the detailed comparisons between these models.

Phenotype difference: Based on previous publications (PMID: 27668431, 38386415), the TFEB transgenic mouse model was established by overexpressing TFEB in cadherin-16–expressing cells using the KSP-Cre driver, while the tRCC mouse model was generated by introducing the ASPSCR1-TFE3 transgene under the control of Sglt2-Cre. Comparing the reported phenotypes from these two models, kidney-specific TFEB overexpression leads to relatively small tumors characterized by prolonged latency, whereas the ASPSCR1-TFE3 transgene induces tRCC with large, macroscopic tumors that exhibit complete penetrance and short latency.

Distinct molecular features: To compare the molecular features of TFEB/TFE3-mediated renal cancer with those of TFE3 fusion–mediated tRCC, we performed a systematic analysis of published RNA-seq data collected from genetically modified mouse models and human samples from these conditions. To minimize potential batch effects in RNA-seq analyses across samples generated by different labs, we employed a standardized analytical pipeline with stringent criteria to identify differentially expressed genes (DEGs). Additionally, we applied well-established algorithms designed to correct for batch-associated variability (PMID: 30333627). When appropriate, we relied on fold-change values rather than absolute RPKMs to mitigate residual batch influence on data interpretation presented below.

Transcriptomic profiling of kidney samples from the TFEB transgenic mouse model and Sglt2-Cre; ASPSCR1-TFE3 transgene mice revealed distinct molecular signatures (PMID: 27668431, 38386415). RNA-seq analysis identified 4% (P0) or 10% (P14) overlapping differentially expressed genes (DEGs) between these two models, which *underscores the divergent molecular features between the models* (relative to control kidneys; fold changes >1 and FDR <0.05) (**Figure S9A**) although both transgenic mouse models developed renal cancers. We then further compared the expression levels of the canonical TFEB/TFE3 target genes using these RNA-seq data. We found that the expression levels of TFEB/TFE3

Figure S9A

Figure S9B

Figure S9. (A) Venn diagram showing the overlapping genes identified from RNA-seq of indicated mouse models at timepoint P0 and P14. **(B)** Quantification of canonical TFEB/TFE3 targeted gene expression levels in indicated mouse model. (The box plots indicated the median (center line), the third and first quartiles (box limits) and 1.5x interquartile range (IQR) above and below the box (whiskers). Two-sided Wilcoxon test; $p < 0.05$). TPM, transcripts per kilobase million.

target genes (PMID: 32978159,38386415, 36987696) are notably higher in the tRCC mouse model compared to the TFEB transgenic model. This indicates a more strongly activated TFEB/TFE3-driven transcriptional program in tRCC, potentially reflecting the enhanced transcriptional activity conferred by oncofusion proteins relative to their wild-type counterparts (**Figure S9B**).

Similarly, distinct transcriptional signatures are also observed in mouse models with Birt-Hogg-Dubé (BHD) syndrome and tRCC mouse model (see data below). In the Birt-Hogg-Dubé (BHD) syndrome mouse model, conditional knockout of FLCN results in constitutive nuclear localization and transcriptional activation of TFEB and TFE3, ultimately leading to the development of renal cystic phenotypes (PMID: 36987696). While this constitutive nuclear localization resembles the behavior of TFE3 fusion proteins observed in translocation renal cell carcinoma (tRCC), our transcriptomic analysis reveals that there are notable differences between these two disease contexts.

1. Patient samples from BHD and tRCC cases exhibit distinct transcriptomic features based on publicly available RNA-seq data (PMID: 38386415, 36987696). The significantly upregulated genes identified in renal tumors from BHD patients showed minor overlapping with the upregulated genes that are found in tRCC patients, highlighting the distinct molecular landscapes of these two tumor types (**Figure D for Reviewer only**) (relative to control kidneys; fold changes >2 and FDR <0.05).

Figure D for reviewer only

Figure E for reviewer only

Figure D for reviewer only. Venn diagram showing the overlapping genes identified from RNA-seq of indicated patient sample cohorts. Data are available in PMID: 38386415, 36987696.

Figure E for reviewer only. (left): Venn diagram showing the overlapping TFEB/TFE3 targeted genes in DEGs identified from the left Venn diagram. (right): Quantification of canonical TFEB/TFE3 targeted gene expression levels in indicated patient samples. (The box plots indicated the median (center line), the third and first quartiles (box limits) and 1.5x interquartile range (IQR) above and below the box (whiskers). Two-sided Wilcoxon test; p<0.05).

2. TFEB/TFE3 targeted genes enriched in renal tumors from BHD patients are not similarly enriched in renal tumors from tRCC patients. Among the significantly upregulated genes, 229 out of 2,402 in BHD patient tumors and 50 out of 1340 in tRCC patient tumors are validated TFEB/TFE3 target genes (PMID: 32978159,38386415, 36987696). Only 20 overlapping genes are identified. Moreover, the expression levels of canonical TFEB/TFE3 target genes are relatively higher in tRCC patient samples compared to those from BHD patients (**Figure E for Reviewer only**). These findings suggest that while TFEB/TFE3 transcriptional programs are active in both contexts, their relative contributions to the transcriptomic landscape may differ in two disease settings.

Figure F for reviewer only

Figure G for reviewer only

Figure F for reviewer only. Venn diagram showing the overlapping genes identified from RNA-seq of indicated mouse models. Data are available in PMID: 8386415, 36987696.

Figure G for reviewer only. Quantification of canonical TFEB/TFE3 targeted gene expression levels in indicated mouse model. (The box plots indicated the median (center line), the third and first quartiles (box limits) and 1.5x interquartile range (IQR) above and below the box (whiskers). Two-sided Wilcoxon test; p<0.05).

3. Transcriptomic profiling of tumor samples from BHD and tRCC mouse models demonstrates markedly different gene expression patterns (PMID: 38386415, 36987696). Mirroring the transcriptomic differences observed in patient samples, RNA-seq profiling of tumors from BHD and tRCC mouse models also revealed distinct gene expression landscapes (**Figure F for Reviewer only**). Notably, the tRCC model exhibited a more robust activation of canonical TFEB/TFE3 target genes (**Figure G for Reviewer only**), further reinforcing the conclusion that TFE3 oncofusions may drive a more potent transcriptional program compared to TFEB/TFE3.

4. TFEB/TFE3 target gene expression is more robustly activated in the tRCC cell line than in BHD-derived cell lines. RNA-seq analysis (PMID: 38386415, 36987696) of UOK109 (a tRCC cell line) and UOK257 (a BHD cell line) revealed fewer than 8% overlapping DEGs following depletion of NONO-TFE3 and TFEB/TFE3 respectively (**Figure H for Reviewer only**). Moreover, depletion of NONO-TFE3 in UOK109 cells resulted in more pronounced expression changes in these TFEB/TFE3 target genes (PMID: 32978159,38386415, 36987696) than depletion of TFE3 in the UOK257 cells (**Figure I for Reviewer only**).

Altogether, these data suggest that, despite both giving rise to kidney tumors, TFEB/TFE3 and TFE3 fusions might engage divergent transcriptional programs and likely represent biologically distinct entities.

The mechanisms by which TFE3 oncofusions drive kidney tumorigenesis remain incompletely understood. Most existing studies, including our current work, have primarily focused on the direct DNA-binding targets of these onco-fusions and the corresponding downstream transcriptional programs. However, it is plausible that the non-DNA binding interactions, such as proteins or RNAs that can associate with the fusion partners of TFE3 (which are absent for TFE3-S321A), may also contribute to tRCC development and progression through mechanisms that remain to be elucidated. Although our findings support a model in which cis-regulatory feedback, mediating by NONO-TFE3 binding to its own target gene transcripts, enhances transcriptional outputs, we also identified a substantial number of RNA species enriched in RIP-seq that lack the corresponding promoter-associated CUT&Tag signals. This observation raises the possibility that NONO-TFE3 may regulate a subset of its RNA targets through mechanisms that are independent of direct chromatin engagement, potentially involving post-transcriptional regulation or RNA-mediated condensate dynamics. Another compelling mechanism worth considering is the reorganization of three-dimensional chromatin architecture, as transcriptional condensates have been shown to influence higher-order genome topology, thereby facilitating oncogenic gene expression programs and contributing to tumorigenesis (PMID: 34526716, 33417833, 35852417). These hypotheses may also account for the inability of TFE3-S321A to rescue gene expression to the same transcriptional level as NONO-TFE3 and to support cell growth in UOK109 cells. However, further investigation is required to fully elucidate these mechanisms, which is beyond the scope of the current study.

To reflect the reviewer's concerns, we have included the following statements in the "Discussion" section of our revised manuscript:

Figure H for reviewer only

Figure H for reviewer only. Venn diagram showing the overlapping genes identified from RNA-seq of indicated cell lines following depletion of NONO-TFE3 (UOK109) and TFEB/TFE3 (UOK257). Data are available in PMID: 38386415, 36987696.

Figure I for reviewer only

Figure I for reviewer only. Quantification of canonical TFEB/TFE3 targeted gene expression fold changes in indicated cell lines following depletion of NONO-TFE3 (UOK109) and TFEB/TFE3 (UOK257). (The box plots indicated the median (center line), the third and first quartiles (box limits) and 1.5x interquartile range (IQR) above and below the box (whiskers). Two-sided Wilcoxon test; $p < 0.05$).

“These findings align with the insufficient rescue phenotype observed in UOK109-KI cells expressing TFE3-S321A under dTAG-13 treatment. It is worth noting that this observation does not conflict with the established oncogenic role of nuclear TFEB/TFE3 activation in renal cancer. Previous studies have reported that kidney-specific overexpression of TFEB in genetically modified mouse models leads to malignant transformation, supporting the oncogenic role of nuclear activation of TFE transcription factors. We compared published RNA-seq data from genetically modified mouse models with either TFE3 overexpression or ASPSCR1-TFE3 transgene expression (Figure S9). Although both models developed renal cancer, they exhibited distinct transcriptional landscapes, with less than 15% overlap in differentially expressed genes between the two models. We further compared the expression levels of canonical TFEB/TFE3 target genes using these RNA-seq datasets and found that their expression was notably higher in the tRCC mouse model compared to the TFEB transgenic model. This indicates a more strongly activated TFEB/TFE3-driven transcriptional program in tRCC, potentially reflecting the increased transcriptional activity conferred by the oncofusion proteins compared to their wild-type counterparts. Collectively, these data suggest that while nuclear activation of TFE family members has an oncogenic function in renal cancer generally, TFE3 oncofusion proteins are specifically required to mediate malignant transformation in tRCC.”